# *OsMADS17* simultaneously increases grain number and grain weight in rice

Yuanjie Li[1,2,3], Sheng Wu[1,3], Yongyu Huang[1], Xin Ma[1], Lubin Tan [1], Fengxia Liu[1], Qiming Lv [2], Zuofeng Zhu [1], Meixia Hu[2], Yongcai Fu[1], Kun Zhang[1], Ping Gu[1], Daoxin Xie[3], Hongying Sun[1] ✉ & Chuanqing Sun[1,2] ✉

During the processes of rice domestication and improvement, a trade-off effect between grain number and grain weight was a major obstacle for increasing yield. Here, we identify a critical gene *COG1*, encoding the transcription factor OsMADS17, with a 65-bp deletion in the 5′ untranslated region (5′ UTR) presented in cultivated rice increasing grain number and grain weight simultaneously through decreasing mRNA translation efficiency. OsMADS17 controls grain yield by regulating multiple genes and that the interaction with one of them, *OsAP2-39*, has been characterized. Besides, the expression of *OsMADS17* is regulated by OsMADS1 directly. It indicates that *OsMADS1-OsMADS17-OsAP2-39* participates in the regulatory network controlling grain yield, and downregulation of *OsMADS17* or *OsAP2-39* expression can further improve grain yield by simultaneously increasing grain number and grain weight. Our findings provide insights into understanding the molecular basis co-regulating rice yield-related traits, and offer a strategy for breeding higher-yielding rice varieties.

Crop domestication and improvement are outstanding events in human history. During the process of the wild rice (*Oryza rufipogon* Griff.) domesticated into cultivated rice (*O. sativa* L.), remarkable morphological transitions have occurred, such as superior plant architecture accompanied by decreased tiller number and tiller angle and higher grain yield resulted from increased both grain number and grain weight[1–14] (Fig. 1a, b). After domestication, increasing grain yield remained to be the prime target in the process of rice improvement. Over the past decades, a number of key genes regulating grain number[15–25] and grain weight[26–42] had been characterized, which largely contributed to the significant achievements in genetic improvement of rice grain yield. However, most of these genes only regulated one of the traits, or even negatively regulated the other one, which were known as the trade-off effect between grain number and grain weight, and it restrained the greater achievements for improving rice grain yield. Therefore, how to overcome the trade-off effect between grain number and grain weight and promoting them

simultaneously is a challenging task left to be resolved in modern breeding programs.

Genomic sequence variants arising in the gene-coding and regulatory regions are important genetic bases of trait variation. In terms of gene expression, these variants involve complex procedures, crucially including transcription and translation. Previous studies devoted most of their attention to changes in transcript levels for deciphering the mechanism of rice yield-related trait variation[4–7,20–23,32–39]. However, translation, including all the post-transcriptional processes, is more rarely mentioned, and sequence polymorphisms in the 5′ untranslated region (5′ UTR) for regulating rice grain yield through variation in translation efficiency without changes in transcript level or protein function are scarcely reported.

In the current study, we identify a wild rice introgression line 8IL73 with less and smaller grains compared with the recipient parent, a *japonica* variety, C418. Through genetic analysis and map-based cloning, we identify a quantitative trait locus *CONTROL OF GRAIN*

[1]Department of Plant Genetics and Breeding, China Agricultural University, Beijing 100193, China. [2]State Key Laboratory of Hybrid Rice, Hunan Hybrid Rice Research Center, Hunan Academy of Agricultural Sciences, Changsha 410125, China. [3]MOE Key Laboratory of Bioinformatics, Department of Biological Sciences and Biotechnology, Tsinghua University, Beijing 100084, China. ✉e-mail: hysun@cau.edu.cn; suncq@cau.edu.cn

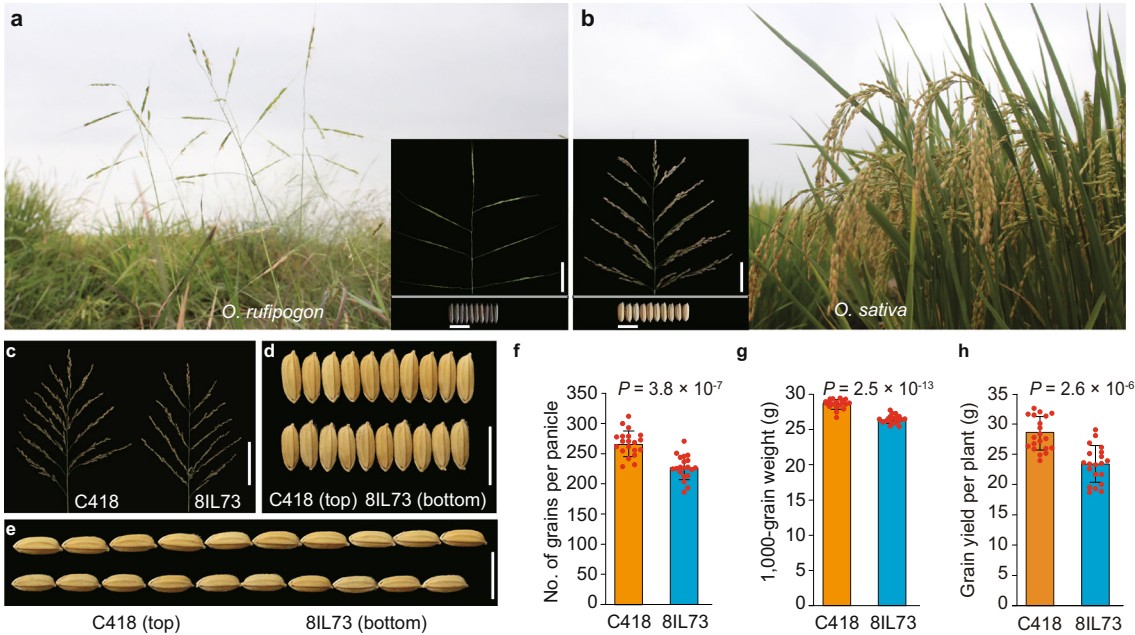

**Fig. 1 | Cultivated rice (*Oryza sativa*) generated more grain number and larger grain size than common wild rice (*O. rufipogon*). a**, **b** Phenotype of *O. rufipogon* (**a**) and *O. sativa* (**b**). Panicles and grains are shown in the lower corner. Scale bars, 5 cm for panicles, 1 cm for grains. **c**–**e** Main panicles (**c**), grain width (**d**), and grain length (**e**) of C418 and 8IL73 plants. Scale bars, 10 cm (**c**), 1 cm (**d**, **e**).

**f**–**h** Comparison of grain number per main panicle (**f**), 1000-grain weight (**g**), and grain yield per plant (**h**) between C418 and 8IL73. Data were means ± standard deviation (s.d.) (*n* = 20 plants), comparisons are made by two-tailed Student's *t*-test. Source data underlying Fig. 1f–h are provided as a Source Data file.

*YIELD 1* (*COG1*) encoding the transcript factor OsMADS17 simultaneously increasing both grain number and grain size through decreased translation efficiency caused by a 65-bp deletion in the 5′ UTR. *OsMADS17* combined with the upstream gene *OsMADS1* and the downstream gene *OsAP2-39* participates in the regulatory network regulating rice grain yield, and the favorable variants of *OsMADS1*, *OsMADS17*, and *OsAP2-39* can increase grain yield in rice improvement. Furthermore, the downregulation of *OsMADS17* or *OsAP2-39* expression in high-yielding rice varieties increases grain yield by further improving grain number and grain weight simultaneously. Our findings will provide not only insights into molecular mechanisms for increasing grain yield in rice, but also identify strategies for co-improvement of grain number and grain size.

## Results

### *COG1* controls grain number and grain weight in rice

We found that a wild rice introgression line 8IL73, which harbors seven chromosomal segments from DXCWR, the donor common wild rice (*O. rufipogon* Griff.) parent from Dongxiang, Jiangxi Province, China (Supplementary Fig. 1), had less grains per main panicle (−14.9%), lower individual grain weight (−7.6%), and lower grain yield per plant (−17.8%) compared with the recipient parent, an elite *japonica* variety C418 (Fig. 1c–h and Supplementary Fig. 2). To identify the genetic factors responsible for co-regulating grain number and grain weight in DXCWR, we carried out a genetic linkage analysis using 203 F$_2$ individuals derived from the cross between 8IL73 and C418, and detected a major quantitative trait locus (QTL) for grain number and grain weight located on the long arm of chromosome 4 (Supplementary Table 1 and Supplementary Fig. 3), which was named *CONTROL OF GRAIN YIELD 1* (*COG1*). Through evaluating grain number per main panicle of each recombinant identified from 6123 F$_2$ individuals, we fine-mapped *COG1* to a 19.6-kb interval, which contained only one predicted gene (*LOC_Os04g49150*), encoding a MADS-box transcription factor OsMADS17 (Fig. 2a). Sequencing the transcript region of *LOC_Os04g49150* revealed that there were ten single-nucleotide

polymorphisms (SNPs) and six insertions/deletions (indels) between 8IL73 and C418 (Supplementary Fig. 4). Among these variations, a synonymous mutation (SNP 1) and a 65-bp insertion/deletion (Indel 1) existed in the coding region and the 5′ UTR, respectively (Supplementary Fig. 4). Reverse transcription quantitative PCR (RT-qPCR) analysis showed that the transcript levels of *LOC_Os04g49150* were not significantly different between 8IL73 and C418 in developing young panicles (Supplementary Fig. 5a).

As sequence comparisons and transcript level analysis did not provide enough evidence to estimate the functional allele of *LOC_Os04g49150*, we generated two reciprocal transgenic plants, C418$^{CTP}$ and 8IL73$^{CTP}$. The C418$^{CTP}$ transgenic plants, which were generated by introducing the *LOC_Os04g49150* gene from 8IL73 into C418 plants, showed a similar *OsMADS17* expression level, but fewer grains per panicle, a lower grain weight, and grain yield than that of the C418 control plants (Fig. 2b–h and Supplementary Fig. 6). On the other hand, the 8IL73$^{CTP}$ transgenic plants, which were generated by introducing the *LOC_Os04g49150* gene from C418 into 8IL73 plants, displayed no significant difference in grain yield-related traits compared with 8IL73 (Supplementary Fig. 7). To further verify the function of *LOC_Os04g49150*, we generated a *OsMADS17* specific RNA interference (RNAi) vector (Supplementary Fig. 8a). Transgenic 8IL73 plants expressing the RNAi vector (8IL73$^{RNAi-OsMADS17}$) exhibited a significant decrease in *OsMADS17* expression levels and a significant increase in grain number, grain size, and grain yield compared with the controls (Fig. 2i–o and Supplementary Fig. 8b–j). Taken together, these results demonstrated that *LOC_Os04g49150* was responsible for the *COG1* gene, which functions as a simultaneous co-regulator for grain number and individual grain weight in rice.

### A 65-bp deletion in the 5′ UTR of OsMADS17 increases grain yield

To identify the causal variation in the *OsMADS17* gene between the two parental accessions, we used two genomic chimeric constructs to generate reciprocal, complementary transgenic plants C418$^{CTP-WC}$ (involving the 2.5-kb *OsMADS17* promoter from 8IL73 fused with the

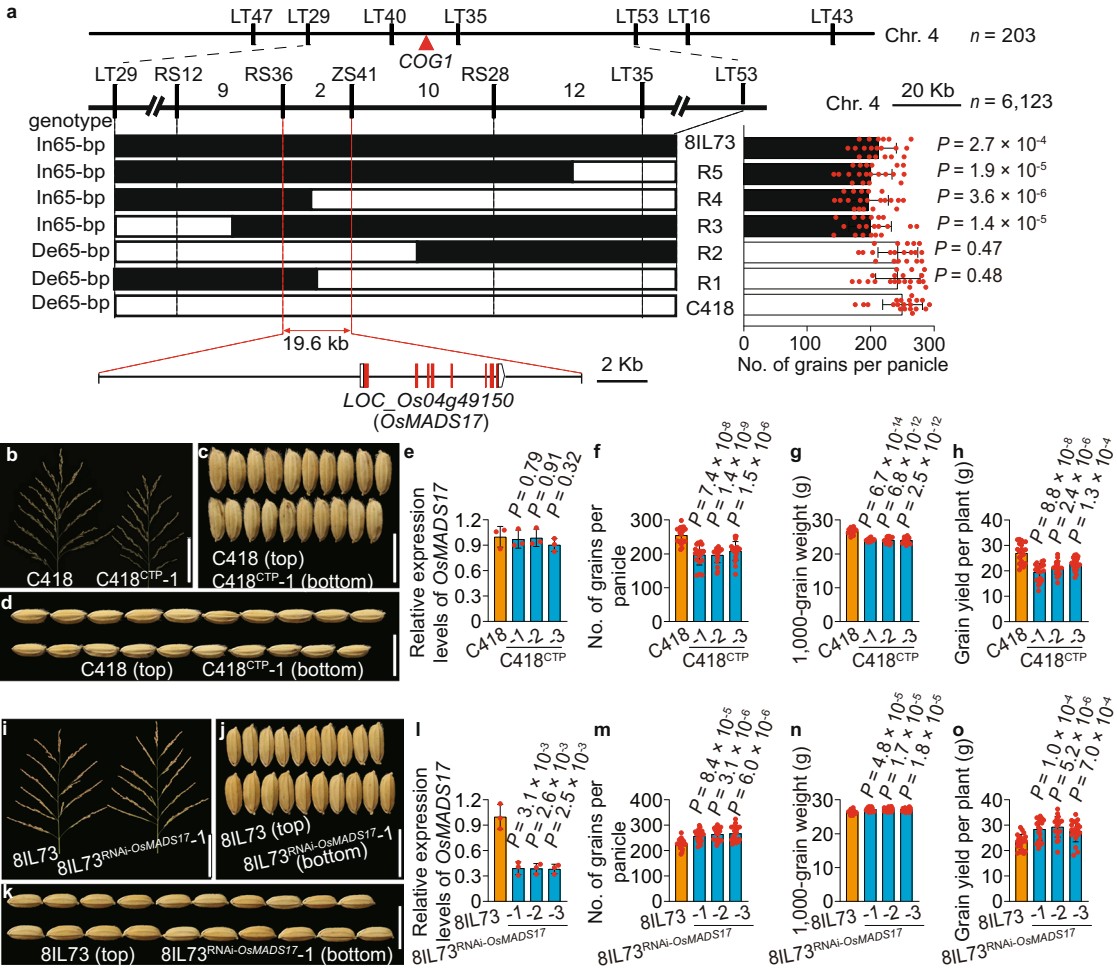

**Fig. 2 | Map-based cloning and gene function identification of *COG1*. a** *COG1* was mapped onto the long arm of chromosome 4 in the 19.6-kb region. **b–d** Main panicles (**b**), grain width (**c**), and grain length (**d**) of C418 and C418^CTP plants. **e** Relative expression levels of *OsMADS17* in C418 and C418^CTP plants ($n = 3$ replicates). **f–h** Comparison of grain number per main panicle (**f**), 1000-grain weight (**g**), and grain yield per plant (**h**) between C418 and C418^CTP plants ($n = 20$ plants).

**i–k** Same as (**b–d**) for 8IL73 and 8IL73^RNAi-*OsMADS17* plants. **l** Relative expression levels of *OsMADS17* in 8IL73 and 8IL73^RNAi-*OsMADS17* plants ($n = 3$ replicates). **m–o** Same as (**f–h**) for 8IL73 and 8IL73^RNAi-*OsMADS17* plants ($n = 20$ plants). Scale bars, 10 cm (**b**, **i**), 1 cm (**c**, **d**, **j**, **k**). Data were means ± s.d., comparisons are made by two-tailed Student's *t*-test. Source data underlying Fig. 2a, e–h, l–o are provided as a Source Data file.

*OsMADS17* transcript region from C418 in a C418 background) and C418^CTP-CW (involving the 2.5-kb *OsMADS17* promoter from C418 fused with the *OsMADS17* transcript region from 8IL73 in a C418 background) (Supplementary Fig. 9a). RT-qPCR and phenotypic analysis showed that no significant change in *OsMADS17* expression levels appeared between C418 and either of the transgenic plants, C418^CTP-CW or C418^CTP-WC, but the C418^CTP-CW transgenic plants had significantly fewer grains, significantly smaller grain size and lower grain yield than the C418 control, whereas the C418^CTP-WC transgenic plants showed a similar phenotype to that of the C418 control (Fig. 3a–g, Supplementary Fig. 9b–j, and Supplementary Data 1). These results indicated that variants in the transcript region of *OsMADS17* were responsible for the differences in grain number and individual grain weight between C418 and 8IL73.

To reveal the expression pattern of *OsMADS17*, we carried out RT-qPCR and RNA in-situ hybridization analysis, which showed that *OsMADS17* was predominantly expressed in the developing panicle and in panicle branch primordia, but that there was no significant difference in *OsMADS17* transcript levels between 8IL73 and C418 (Supplementary Fig. 5a, c–e). As no differences were detected in neither the transcript expression levels of the *OsMADS17* gene nor the amino acid sequence of OsMADS17 between 8IL73 and C418, we carried out western blot assays using the specific antibody Anti-OsMADS17 (Fig. 3h)

and found that C418 contained lower OsMADS17 protein levels than that of 8IL73 in young panicles and other tissues, such as root, tiller base, and node (Fig. 3i and Supplementary Fig. 5b). Further analysis for OsMADS17 protein levels in transgenic plants showed that C418^CTP and C418^CTP-CW contained higher OsMADS17 protein levels than in C418 plants (Fig. 3j, k), although no significant change in OsMADS17 protein levels were apparent between C418^CTP-WC and C418 plants (Fig. 3k). Correspondingly, the 8IL73^RNAi-*OsMADS17* transgenic plants had lower OsMADS17 protein accumulation than did the control (Fig. 3l). Therefore, we speculated that the 65-bp indel (for convenience, the 65-bp insertion is hereafter termed In65-bp, and the deletion of 65-bp is termed De65-bp) is probably involved in regulating the process of *OsMADS17* translation and causing the function change of *OsMADS17*.

To test this hypothesis, we conducted transient expression assays, using the dual-luciferase reporter system. The constructs with 5′ UTR^De65-bp of *OsMADS17* generated lower LUC/REN activity levels and lower translation efficiencies (TE), but no significant change in *LUC/REN* mRNA levels compared with 5′ UTR^In65-bp (Fig. 3n). Further analysis indicated that the fragments F3 (22-bp of the 5′ region of the 65-bp indel), F5 (32-bp at the end of 5′ region in the 65-bp, including the fragment F3), and the full length of 65-bp sequence promoted TE significantly compared with the other fragments within the 65-bp sequence. These results confirmed that the In65-bp in the 5′ UTR of the

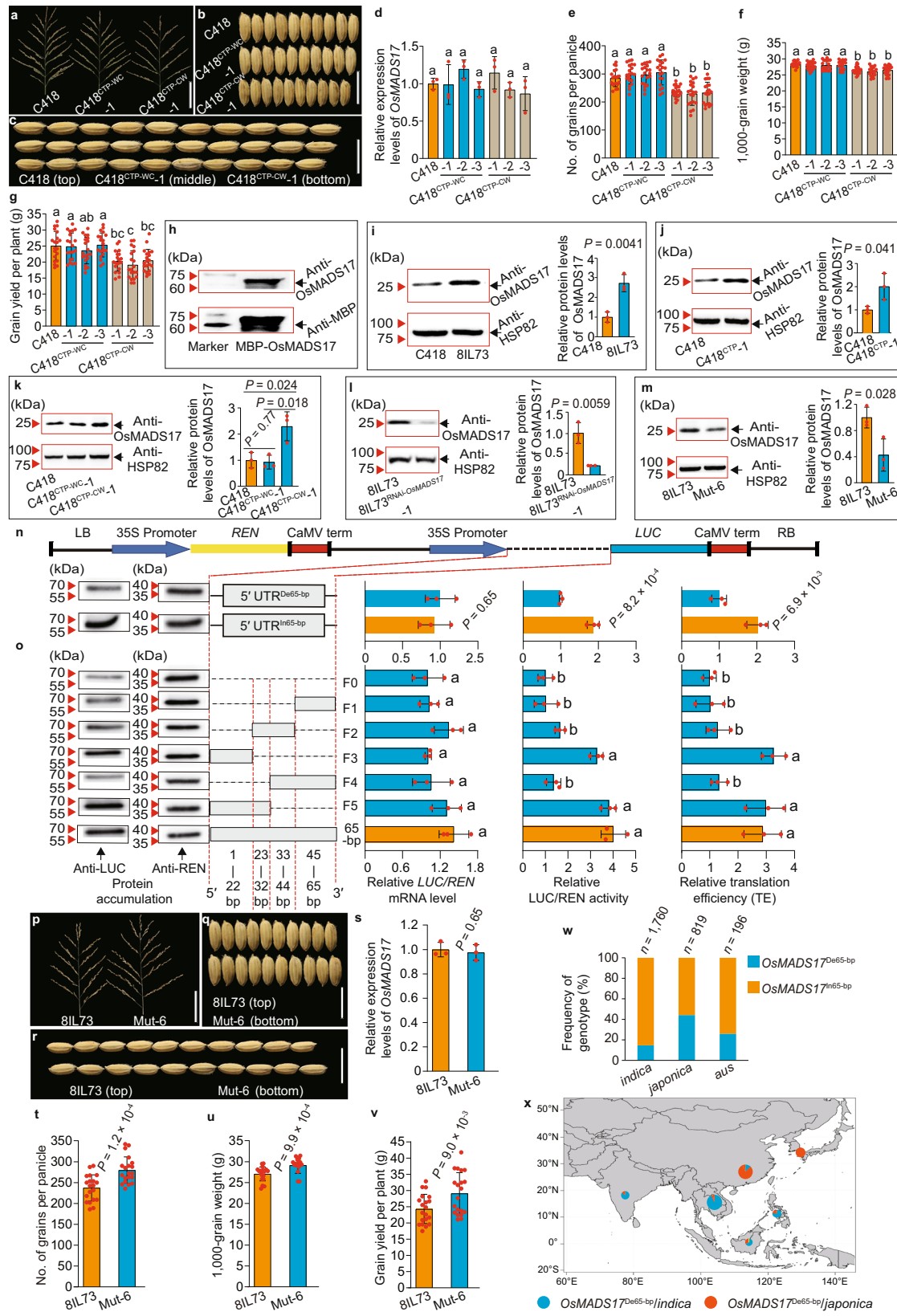

*OsMADS17* gene promoted TE significantly, and that the 22-bp sequence at the end of the 5′ region of the 65-bp was crucial for increasing translation (Fig. 3o and Supplementary Data. 2).

To further assess the function of the 65-bp sequence for increasing TE of the *OsMADS17* gene in rice plants, we selected the F3 fragment, which was located at the interval of PAM1 (GGG) and PAM2

(TGG) as the target for genome editing by the CRISPR/Cas9 system in 8IL73 plants (Supplementary Fig. 10a). Among the six types of transgenic plants, only the Mut-6 plants with the 34-bp deletion containing the F3 fragment exhibited significantly increased grain number, grain weight, and grain yield compared to 8IL73, accompanied by decreased TE and accumulation levels of OsMADS17 protein, but no significant

**Fig. 3 | A 65-bp deletion in the 5′ UTR of *OsMADS17* increases grain yield by attenuating translation efficiency. a–c** Main panicles (**a**), grain width (**b**), and grain length (**c**) of C418, C418^CTP-WC, and C418^CTP-CW plants. **d** Relative expression levels of *OsMADS17* in C418, C418^CTP-WC, and C418^CTP-CW plants (*n* = 3 replicates). **e–g** Comparison of grain number per main panicle (**e**), 1000-grain weight (**f**), and grain yield per plant (**g**) among C418, C418^CTP-WC, and C418^CTP-CW plants (*n* = 20 plants). **h** Determination of specificity of Anti-OsMADS17 antibody. Purified recombinant MBP-OsMADS17 proteins were detected by anti-OsMADS17 and anti-MBP antibody (*n* = 3 replicates). **i–m** Comparison of OsMADS17 protein levels in young panicles of C418 and 8IL73 (**i**), C418 and C418^CTP (**j**), C418, C418^CTP-WC, and C418^CTP-CW (**k**), 8IL73 and 8IL73^RNAi-*OsMADS17* (**l**), and 8IL73 and Mut-6 plants (**m**). HSP82 was used as the loading control (*n* = 3 replicates). **n, o** Dual-LUC reporter system for translation efficiency (TE) analysis. The 5′ UTR of *OsMADS17* with De65-bp

attenuated TE (**n**), and the fragment F3 with 22-bp at the end of the 5′ region was critical for the 65-bp fragment increasing TE (**o**) (*n* = 3 replicates). **p–r** Same as (**a**) to (**c**) for 8IL73 and Mut-6 plants. **s** Relative expression levels of *OsMADS17* in 8IL73 and Mut-6 plants (*n* = 3 replicates). **t–v** Same as (**e–g**) for 8IL73 and Mut-6 plants (*n* = 20 plants). **w** The frequency of genotypes *OsMADS17*^De65-bp and *OsMADS17*^In65-bp in cultivated rice. **x** The geographical distribution of cultivars with genotype *OsMADS17*^De65-bp in areas of Asia. Scale bars, 10 cm (**a**, **p**), 1 cm (**b**, **c**, **q**, **r**). In **d–g**, **o**, the statistical significance was determined by one-way ANOVA with Tukey's multiple comparisons test, different letters represent significant differences (*P* < 0.05), and the exact *P* values were provided in Supplementary Data 1, 2. In **i–n**, **s–v**, comparisons are made by two-tailed Student's *t*-test. Data were means ± s.d. Source data underlying Fig. 3d–o, s–x are provided as a Source Data file.

differences in mRNA levels (Fig. 3m, p–v and Supplementary Fig. 10b–j). Taken together, the De65-bp in the 5′ UTR of the *OsMADS17* gene significantly attenuated TE and resulted in increased grain number, grain weight, and grain yield.

To explore the distribution of the 65-bp indel in rice populations, we screened 82 accessions of wild rice and 169 varieties of cultivated rice (including 79 *indica* and 90 *japonica* varieties) for the 65-bp indel in the 5′ UTR of *OsMADS17* and found that all 82 accessions of wild rice harbored the 65-bp insertion (Supplementary Data 3), whereas 111 of the cultivars contained the In65-bp, with the rest of 58 cultivars carrying the De65-bp; no other allele was found (Supplementary Fig. 11a, b). We further analyzed the distribution of the 65-bp indel in 2775 cultivated rice accessions from the 3 K RG (RFGB v2.0 database, https://www.rmbreeding.cn/). The distribution study showed that the frequency of cultivars with the genotype *OsMADS17*^De65-bp in *japonica*, *indica*, and *aus* rice was 44.2, 14.8, and 26.0%, respectively (Fig. 3w), with the *japonica* varieties with *OsMADS17*^De65-bp being distributed mainly in East Asia, whereas the *indica* varieties with *OsMADS17*^De65-bp were mainly distributed in Southeast Asia (Fig. 3x). These results confirmed that the 65-bp deletion in the 5′ UTR of the *OsMADS17* gene originated from the mutation in cultivated rice.

To investigate the relationship between the 65-bp indel and grain number or grain weight, we analyzed 169 cultivated rice accessions with respect to the phenotype and the genotype at the *OsMADS17* locus. Analysis showed that the 58 cultivars with the genotype *OsMADS17*^De65-bp displayed significantly higher individual grain weight than the 111 cultivars with the genotype *OsMADS17*^In65-bp, whereas no significant difference in grain number was observed between the two groups (Supplementary Fig. 11a, b). Further analysis of 1591 cultivars from 3 K RG showed that 403 cultivars containing the genotype *OsMADS17*^De65-bp exhibited significantly higher grain weight than the 1188 cultivars with the genotype *OsMADS17*^In65-bp (Supplementary Fig. 11c). These results indicated that the 65-bp deletion was associated with larger grains in cultivated rice.

## OsMADS17 targets *OsAP2-39* to regulate grain yield

In order to identify the downstream genes regulated by OsMADS17, we first performed RNA-sequencing analysis using young panicles from C418, 8IL73, and 8IL73^RNAi-*OsMADS17* plants. As a result, 536 genes were shown to be significantly differentially expressed between C418 and 8IL73, whereas 953 differentially expressed genes (DEGs) were detected between 8IL73^RNAi-*OsMADS17* and 8IL73 (Supplementary Fig. 12a). We identified 355 overlapping DEGs between the two comparative groups for analysis with respect to the Gene Ontology (GO) and Kyoto Encyclopedia of Genes and Genomes (KEGG) databases. It showed that *OsMADS17* participated in multiple regulatory pathways, such as regulation of transcription, gene expression, and plant hormone signal transduction (Supplementary Fig. 12b–d). Among the 355 overlapping DEGs, we found 11 genes of the *APETALA2* (*AP2*) family (Fig. 4a), which play critical roles during the processes of floral organ development. Among the 11 genes, *OsAP2-39* (*LOC_Os04g52090*) expression was

proved to be negatively correlated with rice grain number and yield in a previous study[43], and RT-qPCR analysis also further confirmed significant differences in *OsAP2-39* expression in C418, 8IL73, and 8IL73^RNAi-*OsMADS17* plants (Fig. 4b). To explore the interaction between *OsMADS17* and *OsAP2-39*, we analyzed the promoter region of *OsAP2-39*, finding that CArG motifs located at ~2-kb upstream of the start codon (ATG) were putative binding sites (PBS) for MADS-box transcription factors. Yeast one-hybrid (Y1H) assays and Electrophoretic Mobility Shift Assays (EMSA) proved that OsMADS17 could bind to the PBS, and the transient expression system in rice protoplasts showed that OsMADS17 promoted the activation of the 2.4-kb promoter of *OsAP2-39* (Fig. 4c–e). These results confirmed that OsMADS17 regulated the expression levels of *OsAP2-39* positively by interacting physically with the promoter region.

To determine whether the *OsAP2-39* gene was responsible for rice grain yield regulation, we constructed the *OsAP2-39* overexpressed (OE-*OsAP2-39*) and *OsAP2-39* specific RNA interference (RNAi-*OsAP2-39*) vectors (Supplementary Fig. 13a) to generate C418^OE-*OsAP2-39* and C418^RNAi-*OsAP2-39* transgenic plants, respectively. The C418^OE-*OsAP2-39* overexpressed plants produced fewer grains, smaller grain size, and lower grain yield (Fig. 4f–i and Supplementary Fig. 13b–k), whereas the C418^RNAi-*OsAP2-39* RNAi plants resulted in an increase in grain number, grain size, and grain yield (Fig. 4j–m and Supplementary Fig. 13l–u) compared with C418 plants. To explore the relationship between the grain yield-related traits and nucleotide polymorphisms at the *OsAP2-39* locus, we searched the 3 K RG database with an association test between 1000-grain weight (TGW) and sequence variations. Among all of the variants in the 3-kb promoter region, one SNP (S2, T > C, 2256-bp upstream of the start codon (ATG)) showed the strongest signal correlating with TGW (Fig. 4n). Haplotype analysis also showed that cultivars with the *OsAP2-39*^T haplotype exhibited heavier TGW than those with *OsAP2-39*^C (Fig. 4o and Supplementary Fig. 14a, b). In addition, RT-qPCR analysis in young panicles with a length of 0–0.5 cm revealed that cultivars with *OsAP2-39*^T showed lower expression levels of *OsAP2-39* than did cultivars with *OsAP2-39*^C (Supplementary Fig. 14c). These results indicated that *OsAP2-39* regulated rice grain number, grain weight, and grain yield negatively.

To gain an insight into the distribution of the *OsAP2-39*^T/C genotype in different rice populations, we screened the 82 accessions of wild rice, finding all of them contained the genotype *OsAP2-39*^C (Supplementary Data 3). In the 2,775 cultivated rice accessions from rice 3 K RG, the frequency of cultivars with *OsAP2-39*^T in *japonica*, *indica*, and *aus* rice was 11.6, 6.6, and 4.1%, respectively (Fig. 4p). In addition, the combination of genotype *OsAP2-39*^T with genotype *OsMADS17*^De65-bp occurred in cultivated rice at a much lower frequency than those having only *OsAP2-39*^T or *OsMADS17*^De65-bp (Fig. 4q). Furthermore, the *OsAP2-39*^T genotype mainly appeared in East Asia and Southeast Asia in both *indica* and *japonica* rice populations, whereas the combination of *OsAP2-39*^T with *OsMADS17*^De65-bp appeared only in *indica* rice, being mainly distributed in Southeast Asia (Fig. 4r). Taken together, *OsAP2-39*^T is a rare mutation in cultivated rice, and the favorable variation of

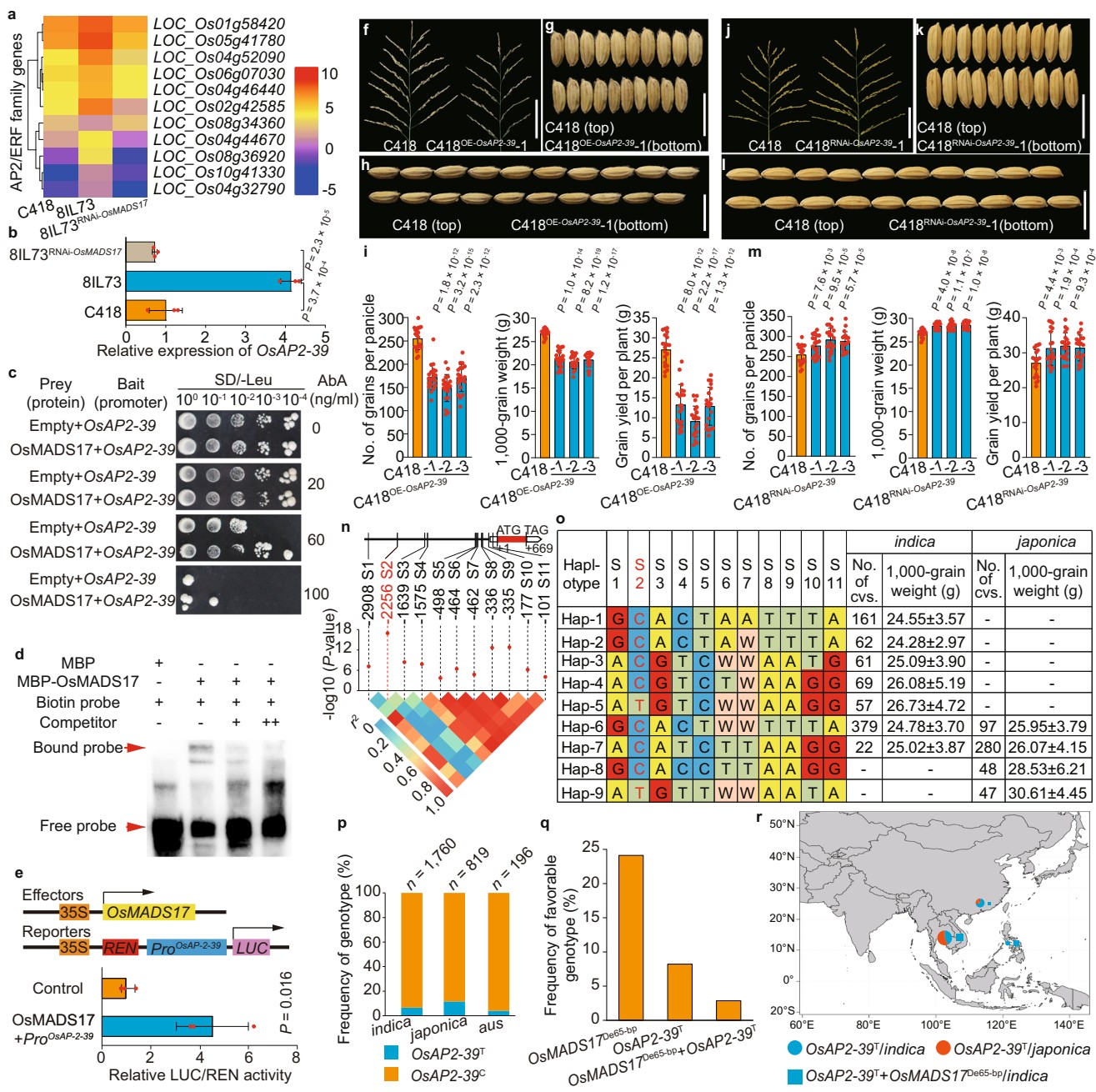

**Fig. 4 | OsAP2-39 was controlled by OsMADS17 directly and regulated grain yield negatively. a** Differentially expressed AP2/ERF family genes revealed by the RNA-sequencing analysis. **b** Validation for the expression levels of *OsAP2-39* by RT-qPCR (*n* = 3 replicates). **c**, **d** Yeast one-hybrid (**c**) and EMSA assays (**d**) proved OsMADS17 bound to the promoter region of *OsAP2-39* (*n* = 3 replicates). **e** Dual-*LUC* reporter assays indicated OsMADS17 promoted activation of the promoter of *OsAP2-39*. Control, cotransfected with reporter construct and an empty effector construct (set to 1) (*n* = 3 replicates). **f**–**h** Main panicles (**f**), grain width (**g**), and grain length (**h**) of C418 and C418^OE-*OsAP2-39* plants. **i** Comparison of grain number per main panicle, 1000-grain weight, and grain yield per plant between C418 and C418^OE-*OsAP2-39* plants (*n* = 20 plants). **j**–**m** Same as (**f**–**i**) for C418 and C418^RNAi-*OsAP2-39* plants (*n* = 20

plants). **n** Association test between 1000-grain weight and variants (with minor allele frequency >0.05) in the 3-kb promoter region of *OsAP2-39*, and the variants were accepted by the standard of *P* < 0.001. **o** Cultivars with haplotype *OsAP2-39*^T (with genotype T at S2 position) showed bigger grain size than the others. No. of cvs., number of cultivars. **p** The frequency of genotypes *OsAP2-39*^T and *OsAP2-39*^C in cultivated rice. **q** The frequency of favorable genotypes in cultivated rice.
**r** Geographical distribution of favorable genotypes in Asia areas. Scale bars, 10 cm (**f**, **j**), 1 cm (**g**, **h**, **k**, **l**). Data for **b**, **e**, **i**, **m** are means ± s.d., comparisons are made by two-tailed Student's *t*-test. Source data underlying Fig. 4a–e, i, m–r are provided as a Source Data file.

*OsAP2-39*^T and *OsMADS17*^De-65bp combined together only existed in *indica* rice.

### *OsMADS17* expression is regulated directly by OsMADS1

To identify the upstream genes of *OsMADS17*, we scanned the promoter region and found two CArG motifs, one of which could probably

interact with OsMADS1[44], which is a MADS-box transcription factor involved in grain length regulation[30,31]. Y1H (Fig. 5a) and EMSA (Fig. 5b) assays confirmed that OsMADS1 could bind to the promoter region of *OsMADS17*, and the transient expression system in rice protoplasts showed that OsMADS1 promoted the activation of the promoter of *OsMADS17* (Fig. 5c). In addition, downregulated expression of

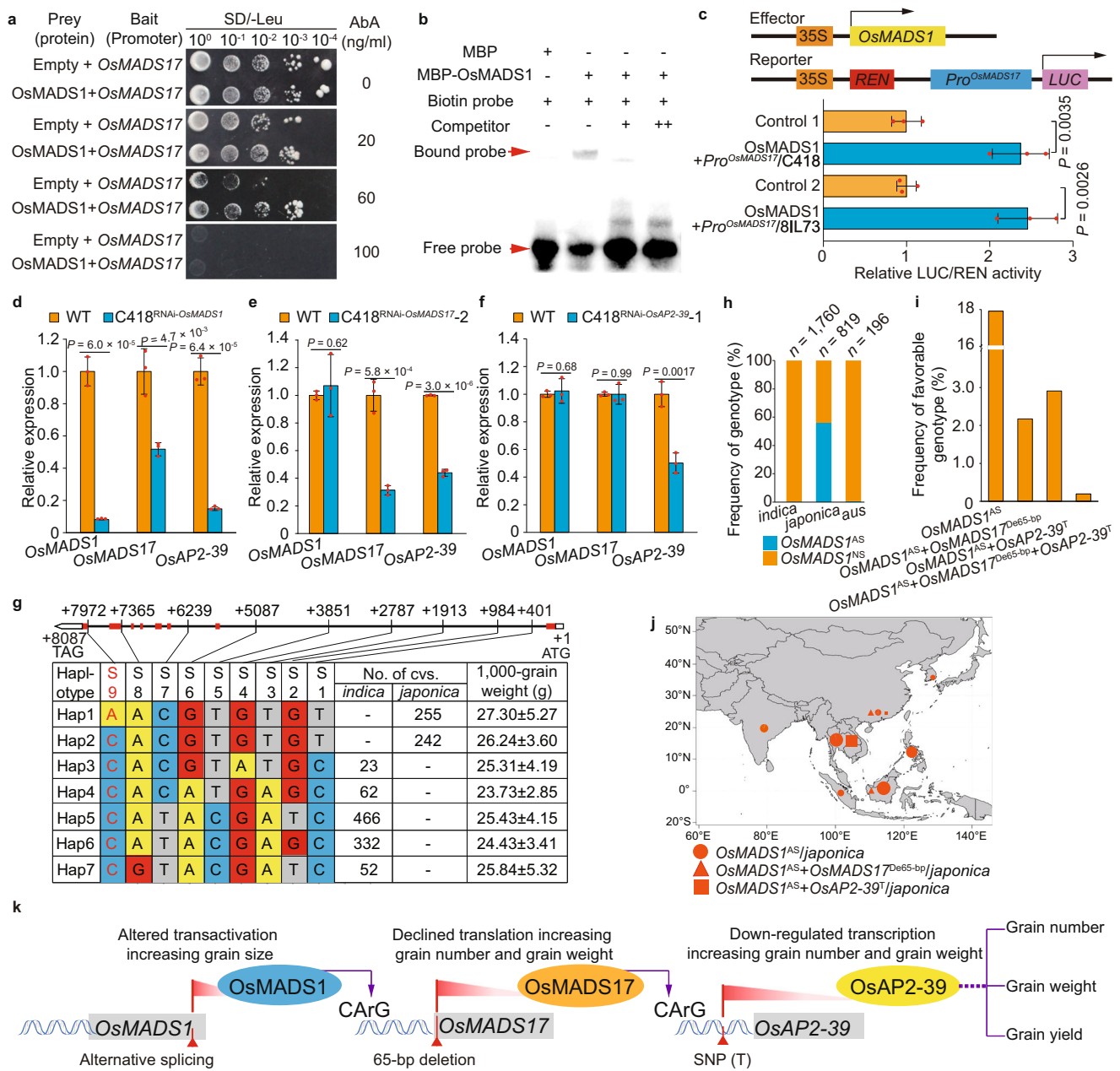

**Fig. 5 | *OsMADS17* expression was regulated by OsMADS1 directly and positively. a, b** Yeast one-hybrid (**a**) and EMSA (**b**) assays proved that OsMADS1 is bound to the promoter region of *OsMADS17*. **c** Dual-*LUC* reporter assays indicated that OsMADS1 promoted the activation of the promoter of *OsMADS17* in both C418 and 8IL73 plants. Control, cotransfected with reporter construct and an empty effector construct (set to 1) (*n* = 3 replicates). **d−f** Relative expression levels of *OsMADS1*, *OsMADS17*, and *OsAP2-39* in young panicles (0−0.5 cm) of C418, C418^RNAi-*OsMADS1*, C418^RNAi-*OsMADS17*, and C418^RNAi-*OsAP2-39* plants (*n* = 3 replicates).

**g** Cultivars with haplotype *OsMADS1*^AS (with genotype A at S9 position) showed bigger grain size than the others. No. of cvs., number of cultivars. **h** The frequency of genotypes *OsMADS1*^AS and *OsMADS1*^NS in cultivated rice accessions. **i** The frequency of different combinations of favorable genotypes in cultivars. **j** Geographical distribution of favorable genotypes in Asia areas. **k** *OsMADS1*, *OsMADS17*, and *OsAP2-39* participated in the regulatory network controlling rice grain yield. Data for **c**−**f** are means ± s.d. and comparisons are made by two-tailed Student's *t*-test. Source data underlying Fig. 5a–j are provided as a Source Data file.

*OsMADS1* was accompanied by decreased expression of *OsMADS17* (Fig. 5d). This evidence indicated that OsMADS1 regulated the expression of *OsMADS17* both positively and directly.

To identify the mechanism of *OsMADS1* in controlling grain yield, we generated RNAi transgenic plants of *OsMADS1* in the C418 background (C418^RNAi-*OsMADS1*). Analysis showed that the C418^RNAi-*OsMADS1* plants generated significantly longer grains, but there was no significant difference in grain number compared with the control (Supplementary Fig. 15). As the C418^RNAi-*OsMADS1* plants formed abnormal grains with hulls which were not closed, they showed a decrease in TGW and yield compared with C418 (Supplementary Fig. 15). Recently, it has been

reported that alternative splicing (AS) of the *OsMADS1* transcript was associated with increased grain length and grain yield compared with native splicing (NS, which was accordant with that in the *japonica* rice cultivar Nipponbare)[30,31]. Haplotype analysis also showed that varieties with the *OsMADS1*^AS haplotype exhibiting larger grains than those with the *OsMADS1*^NS haplotype (Fig. 5g). RT-qPCR and western blot analysis showed no significant difference in *OsMADS1*, *OsMADS17*, and *OsAP2-39* transcript levels, as well as OsMADS17 protein levels between NIL-*OsMADS1*^AS and NIL-*OsMADS1*^NS plants (Supplementary Fig. 16). However, the expression levels of *OsMADS1*, *OsMADS17*, and *OsAP2-39* were significantly lower in the C418^RNAi-*OsMADS1* plants compared with the

control (Fig. 5d). These results suggested that *OsMADS1* might regulate grain length by controlling both *OsMADS17* and *OsAP2-39* expression.

To explore the genetic interactions among *OsMADS1*, *OsMADS17*, and *OsAP2-39*, we first analyzed the functions of different alleles of these genes with respect to the regulation of grain yield-related traits, based on the genotypes and phenotypes of accessions from the 3 K RG. We found that cultivars carrying the genotypes *OsMADS1*[AS], *OsMADS17*[De65-bp], or *OsAP2-39*[T] showed higher individual grain weight than those with the genotypes *OsMADS1*[NS], *OsMADS17*[In65-bp], or *OsAP2-39*[C], respectively (Supplementary Figs. 11b, c, 17a). The individual grain weight of the cultivars carrying both *OsMADS1*[AS] and *OsAP2-39*[T] was significantly higher than that in cultivars containing only *OsMADS1*[AS] or *OsAP2-39*[T] (Supplementary Fig. 17b). In addition, cultivars with the genotype *OsMADS17*[De65-bp] and *OsAP2-39*[T] produced larger grains than those carrying only *OsMADS17*[De65-bp] or *OsAP2-39*[T] (Supplementary Fig. 17c). These results suggested that all of the genes *OsMADS1*, *OsMADS17*, and *OsAP2-39* were involved into regulating individual grain weight. To further elucidate the genetic interaction between *OsMADS17* and *OsAP2-39*, we generated the null single mutants *osmads17* and *osap2-39*, and the double mutant *osmads17/osap2-39* in the ZH17 background using the CRISPR/Cas9 system (Supplementary Fig. 18a). Phenotypic analysis showed that the single mutants *osmads17* or *osap2-39* exhibited significantly increased individual grain weight and grain yield compared with the control (Supplementary Fig. 18b–o and Supplementary Data 4), while the double mutant *osmads17/osap2-39* further increased grain weight and grain yield compared with the single mutants *osmads17* or *osap2-39* (Supplementary Fig. 18b–o and Supplementary Data 4), indicating that *OsMADS17* and *OsAP2-39* have additive effects on grain yield regulation. To further investigate the regulatory relationship among *OsMADS1*, *OsMADS17*, and *OsAP2-39*, we analyzed the expression levels of these genes in RNAi transgenic plants in the C418 background, finding that the C418[RNAi-*OsMADS17*] plants showed decreased expression of *OsMADS17* and *OsAP2-39*, but no obvious change in *OsMADS1* expression compared with the control (Fig. 5e). In the C418[RNAi-*OsAP2-39*] plants, unlike the interfered gene *OsAP2-39*, there was no significant decrease in expression of *OsMADS1* and *OsMADS17* relative to the control (Fig. 5f). Taken together, these findings showed that *OsMADS1*, *OsMADS17*, and *OsAP2-39* participated in the regulatory network controlling grain yield (Fig. 5k).

To reveal the distribution of different genotypes for *OsMADS1*, *OsMADS17*, and *OsAP2-39* in rice populations. We studied the 82 accessions of wild rice, discovering that all of them possessed the genotype *OsMADS1*[NS] (Supplementary Data 3), whereas the genotype *OsMADS1*[AS] almost appeared only in *japonica* rice, with a frequency of 56.0% (Fig. 5h), which meant that the *OsMADS1*[AS] gene originated from *japonica* rice. In addition, the frequencies of combinations of *OsMADS1*[AS] with *OsMADS17*[De65-bp] or *OsAP2-39*[T] in cultivated rice accessions were both much lower than the frequency of accessions having only *OsMADS1*[AS], and only 0.2% of cultivars had the genotype with *OsMADS1*[AS], *OsMADS17*[De65-bp], and *OsAP2-39*[T] simultaneously (Fig. 5i). Furthermore, the geographical distribution of *OsMADS1*[AS] partially overlapped with the distribution of *OsAP2-39*[T] in Southeast Asia, whereas *OsMADS1*[AS] and *OsMADS17*[De65-bp] overlapped only at a very low frequency (Fig. 5j). Taken together, these findings indicated that *OsMAD1*[AS] and *OsMADS17*[De65-bp] were distributed independently in cultivated rice, and that *OsAP2-39*[T] and *OsMADS1*[AS] appeared simultaneously merely in *japonica* rice.

### *OsMADS17* or *OsAP2-39* downregulation may increase grain yield in rice breeding

As *OsMADS17* is a negative regulator controlling grain yield in rice, reducing its expression level perhaps could contribute to increasing rice grain yield. The above results had demonstrated that downregulated expression levels of *OsMADS17* with In65-bp in 8IL73 plants could increase grain yield (Fig. 2i–o, Supplementary Fig. 8b–j), which

meant great prospect for improving grain yield of the 75.9% cultivars harboring the 65-bp insertion of *OsMADS17* (Fig. 4q). To further explore the potential of *OsMADS17* increasing grain yield in the remaining of 24.1% cultivated rice with the beneficial 65-bp deletion (Fig. 4q), we generated the transgenic plants C418[RNAi-*OsMADS17*] with downregulated *OsMADS17* expression levels, which showed lower levels of OsMADS17 protein accumulation, higher grain number, bigger grain size, and higher grain yield compared with the controls (Fig. 6a–i and Supplementary Fig. 19). Field trials showed that the C418[RNAi-*OsMADS17*] transgenic plants increased grain yield by 6.5% in Beijing (40.1°N, 116.2°E), northern China, and 12.3% in Hainan Province (18.3°N, 109.2°E), southern China, compared with the control plants (Fig. 6j, k). It indicated that reducing the expression levels of *OsMADS17* in cultivated rice was desirable to improve grain yield.

Additionally, only 8.2% of cultivated rice carried a beneficial variation of genotype *OsAP2-39*[T] (Fig. 4q), which means that reducing the expression levels of *OsAP2-39* in cultivated rice to improve rice grain yield has tremendous potential. In C418[RNAi-*OsAP2-39*] transgenic plants, the grain number per main panicle, individual grain weight, and grain yield increased by 8.6, 6.2, and 15.6% compared with the control, respectively (Fig. 4m). Field trials showed that the C418[RNAi-*OsAP2-39*] transgenic plants increased grain yield by 7.3% in Beijing, compared with the control plants (Fig. 6l). These results indicated that downregulating expression levels of *OsAP2-39* were feasible and achievable strategies for further increasing grain yield in cultivated rice.

## Discussion

Crop domestication and improvement has been an important mission in the activities of human society for more than 10,000 years[45]. Understanding the mechanisms that can increase the grain yield of crop species will guide future breeding efforts, accelerating the development of new improved varieties. To date, a series of the key genes in rice controlling grain number and/or individual grain weight and associated with domestication and plant breeding, such as *PROG1*[1,2], *FZP*[6,7], *OsKRN2*[14], *Gn1a*[15], *IPA1*[20,21], *GS3*[27], and *GW8*[33,34], have been identified. However, the molecular mechanisms for simultaneously co-promoting grain number and individual grain weight in crop domestication and improvement remain largely unknown.

The trade-off effect between grain number and grain weight, in which, for example, increased grain number per panicle is associated with smaller grains, limits any increases in grain yield. Overcoming this trade-off effect has been a tough task for rice improvement. Here, we found that a 65-bp deletion in the 5′ UTR of the *OsMADS17* gene, decreasing translation efficiency or downregulated transcript levels of *OsMADS17*, could increase grain number and individual grain weight simultaneously. As a transcription factor, OsMADS17 probably participated in the regulation of the expression of multiple genes. For instance, the downstream gene *OsAP2-39*, which was positively and directly regulated by OsMADS17, also simultaneously controlled grain number and individual grain weight. Furthermore, the RNA-seq results revealed that the genes *OsGSR1/GW6/OsGASR7*[38–40], *OsCCA1*[46], *OsABCG18*[47], *OsCEP6.1*[48], *GIF1*[49], and *OsMKKK70*[50], which regulated grain number and/or grain weight, were differentially expressed between C418 and 8IL73 (Supplementary Table 2). It is worth mentioning that *OsABCG18* regulates grain number by affecting shootward transport of root-derived cytokinins[47], and *OsGSR1/GW6/OsGASR7* regulates grain size by controlling brassinosteroid (BR) and gibberellin (GA) signaling[38–40]. Therefore, we speculated that OsMADS17 was probably involved in regulating the expression levels of these genes to overcome the trade-off effect between grain number and grain weight. However, the detailed molecular mechanisms still remain to be elucidated.

Gene expression levels, which are normally presented as mRNA measurements[51], are focused on deciphering the mechanism of phenotypic variation in rice. However, the mRNA levels cannot completely reflect the protein levels, as sophisticated multistep processes exist

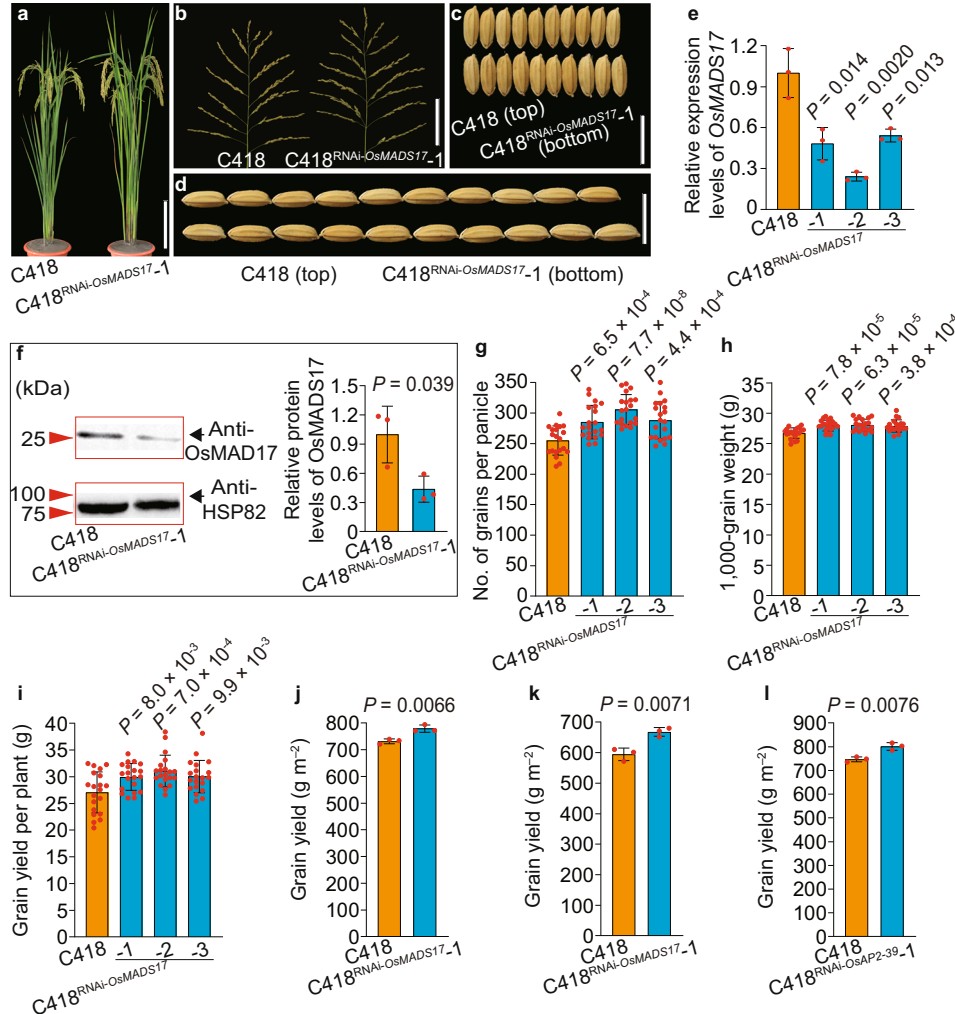

**Fig. 6 | Both *OsMADS17* and *OsAP2-39* exhibit the potential for increasing grain yield. a**–**d** The whole plants (**a**), main panicles (**b**), grain width (**c**), and grain length (**d**) of C418 and C418^RNAi-*OsMADS17*^ plants. **e** Relative expression levels of *OsMADS17* in C418 and C418^RNAi-*OsMADS17*^ plants (*n* = 3 replicates). **f** Comparison of OsMADS17 protein levels in young panicles between C418 and C418^RNAi-*OsMADS17*^ plants by western blot assays. HSP82 was used as the loading control (*n* = 3 replicates). **g**–**i** Comparison of grain number per main panicle (**g**), 1000-grain weight (**h**), and grain yield (**i**) between C418 and C418^RNAi-*OsMADS17*^ plants (*n* = 20 plants). **j, k** Comparison of grain yield in the field trials between C418 and C418^RNAi-*OsMADS17*^ plants (*n* = 3 replicates). 2019, Beijing, China (**j**) and 2020, Hainan Province, China (**k**). **l** Comparison of grain yield in the field trials between C418 and C418^RNAi-*OsAP2-39*^ plants (*n* = 3 replicates). 2021, Beijing, China. Scale bars, 25 cm (**a**), 10 cm (**b**), 1 cm (**c**, **d**). Data were means ± s.d. and comparisons are made by two-tailed Student's *t*-test. Source data underlying Fig. 6e–l are provided as a Source Data file.

from pre-mRNA to mature protein generated, during which translation is critical for the determination of protein accumulation, affecting rice growth and development profoundly. Nevertheless, the mechanisms behind variation in gene expression, caused by translational changes, in the regulation of rice grain yield-related traits are seldom elucidated. In the current study, the De65-bp in the 5′ UTR of *OsMADS17* was shown to decrease translation efficiency leading to altered grain yield resulting from simultaneous increases in grain number and individual grain weight, indicating that diversity of translation levels could also lead to dramatic phenotypic variation in rice, as well as changes in protein function and mRNA transcript levels. Furthermore, the 65-bp fragment in the 5′ UTR of the *OsMADS17* gene might be used as a translation enhancer, increasing endogenous protein levels to regulate these important traits in rice plants. Therefore, these findings provide insight into revealing the mechanism of variation in grain yield-related traits from the perspective of mRNA translation.

In this present study, we demonstrated that *OsMADS1-OsMADS17-OsAP2-39* participated in the regulatory network controlling grain number and grain weight in rice. On the one hand, members within the regulatory network could boost grain yield individually by altering protein function (*OsMADS1*), changing translation (*OsMADS17*), or transcription (*OsAP2-39*) processes. On the other hand, the genotype *OsAP2-39*^T^ could further increase grain weight when in combination with the genotype *OsMADS1*^AS^ or *OsMADS17*^De65-bp^, which meant that diverse combinations of the favorable alleles among the regulatory network could bring about additional avenues for achieving increased grain yield. In addition, downregulating the expression of *OsMADS17* or *OsAP2-39* in the regulatory network could co-enhance grain number and grain weight in rice. These findings not only helped to dissect the mechanism for increasing grain number and grain weight simultaneously for rice improvement, but also provided strategies for co-promoting these two key components of grain yield in rice breeding programs.

## Methods

### Plant materials and growing conditions
The DXCWR-derived introgression line 8IL73 and the recipient parent C418, an elite *japonica* variety, were used for phenotypic evaluation, fine mapping, expression pattern analysis, and the generation of transgenic plants. The *japonica* cultivar ZH17 was used for generating

null single mutants *osmads17* or *osap2-39*, and the double mutant *osmads17/osap2-39*. Materials for field trials of grain yield and associated yield components were planted in a randomized complete block design with three replicates (300 plants for each replication), and the planting density was one plant per 15 cm × 25 cm hill. Plant materials were grown under standard agronomic conditions (fertilizers, crop protection chemicals, etc.) at the experimental stations of China Agricultural University in Beijing and Hainan Province, China.

## Trait measurements

Yield-related traits were measured when the plants were mature. Panicle length, primary branch number, grain number on the primary branch, secondary branch number, grain number on the secondary branch, grain length, grain width, 1000-grain weight, grain number per main panicle, and grain yield per plant were recorded. Yield per plant was scored as the total weight of filled grains from the entire plant. Grain length, grain width, and 1000-grain weight were measured using an automatic seed counting and analyzing instrument (Model SC-G, Wanshen).

## Primers

The primers used in this study are listed in Supplementary Data 5.

## Genetic analysis and positional cloning

The grain number on the main panicle and 1000-grain weight were evaluated for QTL analysis using 203 $F_2$ individuals derived from the cross between C418 and 8IL73. Fine mapping was carried out using 6123 $F_2$ individuals.

## Transgene construct preparation and rice transformation

A total of four different combinations of constructs were used to generate reciprocal transgenic plants (C418[CTP], 8IL73[CTP], C418[CTP-WC], and C418[CTP-CW]) in this study. Target fragments described in the Results were inserted into the vector of pCAMBIA1300 to form complementary constructs. The RNAi constructs, which harbored a hairpin structure with 301-bp (RNAi for *OsMADS1*), 308-bp (RNAi for *OsMADS17*), and 348-bp (RNAi for *OsAP2-39*) inverted repeat cDNA sequences, were inserted into the vector pTCK303/JL1460. The specificity of the RNAi vectors was confirmed by testing the expression levels of 12 MADS family genes or 12 AP2 family genes. The coding sequence of *OsAP2-39* was amplified and inserted into the binary vector pCAMBIA1301 to generate the overexpression construct. To generate fragment deletions within the 65-bp sequence in 5′ UTR of *OsMADS17*, a robust CRISPR/Cas9 system was used for construct generation[52]. To acquire the null mutants for *osmads17*, *osap2-39*, and the double mutant *osmads17/osap2-39*, another CRISPR/Cas9 system was used[53]. All constructs were introduced into *Agrobacterium tumefaciens* strain EHA105 and subsequently transformed into C418, 8IL73, and ZH17 through *Agrobacterium*-mediated transformation for various purposes[22]. Target fragments of the *hygromycin* gene and the *OsMADS17* gene containing the 65-bp indel were amplified using the primers HYG and InDel-65bp, respectively, listed in Supplementary Data 5 to identify the positive transgenic plants. Transgenic plants of the $T_2$ generation were used for phenotypic evaluation.

## RNA extraction and RT-qPCR

Total RNAs were extracted from the various tissues using TRIzol reagent (Invitrogen Life Technologies) and purified with a purification kit (Qiagen). First-strand cDNA was synthesized from 2.5 μg of total RNA in a 20 μL reaction volume using 2 μL oligo(dT)18 primer (Takara) and 1 μL SuperScript® III Reverse Transcriptase (Invitrogen). RT-qPCR was carried out using a CFX96 Real-Time System (Bio-Rad) or StepOnePlus (Applied Biosystems), with the rice *Ubiquitin* gene used as an internal control. cDNA (30 × dilutions) was amplified using the SsoFast EvaGreen Supermix (Bio-Rad). The relative quantification method ($2^{-\Delta\Delta CT}$) was used to evaluate the quantitative variation of expression (relative to *Ubiquitin* expression), and each set of experiments were repeated independently three times[54].

## RNA in-situ hybridization experiment

Young panicles of accessions C418 and 8IL73 were fixed in FAA solution (50 ml of ethanol, 5 ml of acetic acid, 10 ml of 37% formaldehyde, and 35 ml of DEPC-$H_2O$) at 4 °C overnight, followed by a series of dehydrations using ethanol from 50 to 100% and infiltration using xylene from 50 to 100%, and were then embedded in paraffin (Paraplast Plus, Fisher Scientific). The tissues were sliced into approximately 8-μm-thick sections with a microtome (Leica RM2145), and the sections were placed on RNase-free glass slides. The 308-bp coding fragment of the *OsMADS17* gene was amplified from cDNA to form probes. Digoxigenin-labeled sense and antisense RNA probes were generated using a DIG Northern Starter Kit (Roche), according to the manufacturer's instructions. Slides were observed under a light microscope (Leica DMR) and photographed with a micro color camera (Apogee Instruments)[55].

## Western blot analysis

For this assay, young panicles (-1 cm) were collected and grounded into powder in liquid nitrogen and then protein extraction buffer was added (10% glycerol, 5 mM $MgCl_2$, 50 mM Tris-HCl, pH 7.5, 150 mM NaCl, 0.1% IGEPAL CA-630 (Sigma), 0.5 mM DTT (Sigma), 1 mM PMSF (Sigma) and 1× complete protease inhibitor cocktail (Roche)) for total protein extraction. The coding sequences of *OsMADS17* were amplified and inserted into the pMAL-c5x vector to generate recombinant constructs for expressing the MBP (Maltose Binding Protein) fused protein. The recombinant constructs were transformed into the *Escherichia coli* Rosetta (DE3) strain, which were cultured at 37 °C (220 rpm) until the $OD_{600}$ of the cell culture was 0.5. The MBP fused protein was then induced at 16 °C (180 rpm) with 0.5 mM of isopropyl β-D-1-thiogalactopyranoside (IPTG) for 16 h. *E. coli* cells were lysed by an ultrasonic cell crusher (Qsonica, Q700) and centrifuged. The supernatant containing MBP-OsMADS17 protein was affinity-purified using amylose resin (New England Biolabs). The proteins were boiled and electrophoretically separated by 8% (w/v) SDS-PAGE gel electrophoresis, and the polypeptides were transferred to a nitrocellulose membrane (GE Healthcare). A polyclonal antibody for OsMADS17 was raised in rabbits using a specific fragment consisting of OsMADS17 amino acids (C-NKINRQVTFSKRRN) by AbMART (http://www.ab-mart.com.cn/). Specific antibody to OsMADS17 (1:2,000 dilution), antibody to MBP (Cat# AbM59007-3-PU, Beijing Protein Innovation (BPI), 1:1,000 dilution) and HSP82 (Cat# AbM51099-31-PU, Beijing Protein Innovation (BPI), 1:5,000 dilution) were used for detection. HSP82 and OsMADS17 bands were analyzed with the Image J program (ImageJ 1.48, https://imagej.net/ij/) to calculate the relative protein levels according to the software instruction manual.

## Transient expression assays for translation efficiency analysis

For translation efficiency examination, the 5′ UTR of *OsMADS17* in C418, 8IL73, and Mut6 were amplified and inserted upstream of the *luciferase* (*LUC*) coding region in an expression cassette driven by the CaMV 35S promoter to form the reporter. A second expression cassette of *Renilla reniformis luciferase* (*REN*) as an internal vector control was also driven by the 35S promoter[56]. Rice protoplasts were isolated from 10-d-old seedlings grown on MS medium under dark conditions[6]. Constructs were transformed (10 μg plasmid) into rice protoplasts for transient expression. After incubating for 10 and 16 h at 28 °C in the dark, total RNA was extracted to generate cDNA for RT-qPCR analysis and the relative luciferase activities were measured by the Dual-Luciferase Reporter Assay System (Promega), respectively. *LUC/REN* mRNA levels and relative LUC/REN activity conferred by each construct were normalized to those of the controls[57]. To detect the Firefly

Luciferase (LUC) and Renilla Luciferase (REN) protein, antibodies Anti-LUC (Cat# T55401, AbMART, 1:1,000 dilution) and Anti-REN (Cat# T55403, AbMART, 1:1,000 dilution) were used for western blot analysis as described in the part of Western blot analysis. Translation efficiency (TE) is the rate of mRNA translated into proteins within cells. The relative TE was calculated as the ratio of relative LUC/REN activity to relative mRNA levels of *LUC/REN* in rice protoplast (TE=Relative LUC/REN activity/relative mRNA levels of *LUC/REN*).

## RNA-seq analysis

Total RNA was isolated from the young panicles (0.2–0.4 cm) of C418, 8IL73, and 8IL73^RNAi-*OsMADS17*, using TRIzol reagent (Invitrogen) and purified using an RNeasy Plant Mini Kit (Qiagen). Each sample contained 20 plants, with three biological replicates, and were used for paired-end library construction. Sequencing was performed on an Illumina platform at Novogene company, China (https://cn.novogene.com/). HTSeq v0.9.1 was used to count the reads numbers mapped to each gene. The DESeq R package (1.18.0), which provided statistical routines for determining differential expression in digital gene expression data using a model based on the negative binomial distribution, was used for analyzing the differentially expressed genes. The resulting *p* values were adjusted by Benjamini and Hochberg's approach for controlling the false discovery rate. Genes with an adjusted *p* value (*q* value) <0.01 found by DESeq were assigned as differentially expressed. The functional category analysis of the DEGs was performed using agriGO and KEGG[58,59].

## Yeast one-hybrid (Y1H) assays

The coding sequence of *OsMADS1* and *OsMADS17* were subcloned into the pGADT7 vector to generate prey constructs. The fragments with binding sites corresponding to *OsMADS17* and *OsAP2-39* were inserted into the pAbAi vector to generate bait constructs. The bait constructs combined with the pGADT7 empty vector were used as negative controls. The different combinations were co-transformed into yeast strain Y1H Gold. Transformants were grown on the SD/-Leu solid medium at 30 °C for 3 days, and then the transformants were transformed to the SD/-Leu solid medium with or without Aureobasidin A (AbA) at 30 °C for 3–4 days to examine the interactions.

## EMSA analysis

The MBP protein, MBP fused protein MBP-OsMADS1 and MBP-OsMADS17 were generated as described in the part of Western blot analysis. The 5′-biotin-labeled DNA fragments containing the binding site were synthesized and used as probes, and unlabeled DNA fragments of the same sequences were used in competitive reactions. Competition for binding was performed with the cold (unlabeled) probe containing the binding site at 20× and 200× the amount of the labeled probe. Each binding reaction in a total reaction volume of 20 μL contained 2 μL biotin-labeled probe, 5 μg protein, 2 μL 10×binding buffer, and 1 μL 50% (v/v) glycerol. The binding reactions were incubated for 30 min at room temperature and then resolved by electrophoresis on 6% (w/v) native polyacrylamide gels in 0.5×TBE buffer. The biotin-labeled probes were detected using a LightShift Chemiluminescent EMSA Kit (Thermo Fisher Scientific).

## Dual-luciferase reporter assay for transactivation analysis

The coding sequences of *OsMADS1* and *OsMADS17* were inserted into the vector pGreenII 62-SK to form effector constructs, and fragments containing CArG motifs in the regulatory region of *OsMADS17* and *OsAP2-39* were inserted into the vector pGreenII 0800-LUC to generate reporter constructs[56]. A CaMV 35S promoter-driven *Renilla luciferase* (*REN*) gene was used as an internal control. Constructs were co-transformed (5 μg effector plasmid and 5 μg reporter plasmid) into rice protoplasts for transient expression studies. Co-transfection with the reporter construct and an empty effector construct was used for

control. Rice protoplast isolation, plasmid transfection, protoplast incubation, and relative luciferase activities measurement were described earlier in the transient expression assays for translation efficiency analysis.

## Natural variation analysis for *OsMADS1*, *OsMADS17*, and *OsAP2-39*

The wild rice accessions used for genotype analysis are listed in Supplementary Data 3. All data on the cultivated rice accessions in the 3K RG for the association test, phenotype evaluation, genotype analysis, and geographical distribution analysis were obtained from the RFGB v2.0 database (https://www.rmbreeding.cn/)[60]. The frequency of favorable genotypes was analyzed according to the critical variation sites for *OsMADS17* (De65-bp/In65-bp), *OsAP2-39* (T/C), and *OsMADS1* (AS/NS) among 1,760 *indica* rice, 819 *japonica* rice, and 196 *aus* rice accessions. For grain weight analysis, the haplotypes of *OsMADS1* in the 3K RG were classified according to the variants with minor allele frequency >0.05 within the transcript region, and the haplotypes of *OsAP2-39* were constructed by the variants accepted for the association test. The rare haplotypes present in a total of <20 varieties are not shown among the 1039 *indica* rice and 562 *japonica* rice varieties. The geographical distribution of cultivated rice in Asia areas was analyzed by map package (R v4.1.1) using 1507 *indica* rice and 549 *japonica* rice varieties. The number of varieties <10 with only one favorable genotype of *OsMADS1*^AS, *OsMADS17*^De65-bp, or *OsAP2-39*^T and <3 with two or three favorable genotypes not shown.

## Data analysis

QTL analysis was performed by the Map Manager QTXb20[61]. Association analysis was performed with Fisher's exact test, as Fisher's exact test can be powerful for qualitative traits such as grain weight[22]. Comparisons are made by two-tailed Student's *t*-test and one-way ANOVA with Tukey's multiple comparisons test (*P* < 0.05) using SPSS version 17 (SPSS Inc., Chicago, IL, USA), and data were presented as mean ± standard deviation.

## Reporting summary

Further information on research design is available in the Nature Portfolio Reporting Summary linked to this article.

## Data availability

Data supporting the findings of this work are available within the paper and its Supplementary Information files. A reporting summary for this Article is available as a Supplementary Information file. Data of genotype, phenotype (1000-grain weight), subgroup classification, and origin for rice accessions are available on the RFGB v2.0 database [https://www.rmbreeding.cn/]. Information on gene sequence and annotation are available on Rice Genome Annotation Project [http://rice.uga.edu/]. Source data are provided with this paper.

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

## Acknowledgements

This research was supported by the National Natural Science Foundation of China (grant 31830065 and grant 91935302), the 2115 Talent Development Program of China Agricultural University, and the Science and Technology Innovation Program of Hunan Province (grant 2022RC4036). We thank Professor Zichao Li and Professor Zhanying Zhang for providing the NIL-*OsMADS*AS and NIL-*OsMADS*NS materials. We thank Professor Susan R. McCouch for providing the cultivated rice germplasm.

## Author contributions

C.S. and H.S. designed and supervised this study. Y.L. performed experiments and analysis. S.W. provided assistance in collecting the phenotypic data. Y.H. constructed the introgression line 8IL73. X.M. helped analyze the data. F.L., Z.Z., Y.F., K.Z., and P.G. conducted the collection of rice germplasm. Q.L. and M.H. provided technical assistance. Y.L., L.T., D.X., H.S., and C.S. wrote and edited the manuscript. All authors read and approved the final manuscript.

## Competing interests

The authors declare no competing interests.
