## [Peer Review File · Nature Communications]

OsMADS17 simultaneously increases grain number and grain weight in riceREVIEWER COMMENTS

Reviewer #1 (Remarks to the Author):

Several works have identified multiple genes and pathways that control grain number or grain size in cereals, including rice, maize or wheat. However, most of the time those mechanisms modify only one of those traits, and on some occasion, negative correlation between grain size and number have been observed.

In this manuscript Li and collaborators described a genetic pathway to simultaneously control grain size and number in rice.

Initially, the author identified and characterized a QTL, named CONTROL OF GRAIN YIELD 1 (COG1) affecting both traits in a wild introgression line, which harbor chromosomal segments from a donor wild rice into an elite japonica cultivar. Using a fine mapping approach, they identified a MADS-box gene (OsMADS17) as responsible of the phenotype, which was validated by functional complementation and RNAi strategies.

Analysis of genetic variations at MADS17 between cultivars revealed a 65bp deletion in the 5'UTR of the favorable allele that confers more and larger grain phenotype. Importantly, the author used CRISPR-Cas9 to generate a similar 36bp deletion in the 5'UTR that phenocopy the natural 65bp deletion. Since the deletion does not affect transcript level, but reduce protein levels, the author concluded that the 65bp deletion reduces mRNA translation.

Next, the authors were interested in identifying downstream genes of MADS17, and performed RNA-seq in developing apices of plants with different levels of MADS17. They identified a group of 11 AP2 transcription factors, and selected one of them (AP2-39) for functional analysis. Overexpression of AP2-39 resulted in less and smaller grains, while RNAi resulted in more and larger grains, similar to the phenotypes described in plants with reduced levels of MADS17. Next the authors performed biochemical analysis and presented evidence that suggest that MADS17 is a positive regulator of AP2-39. Interestingly, the author also identified natural variation associated with AP2-39 that affect grain weight, with the favorable allele being at low frequency in cultivated rice.

Next, the authors were interested in genes acting upstream of MADS17. They explored MADS1, which is an important regulator of panicle and spikelet development in rice. Plants with reduced expression of MADS1 had low expression of MADS17 and AP2-39, and biochemical analysis suggest that MADS1 could regulate MADS17 expression directly.

Finally, the author proposed a gene network involving these three genes acting sequentially to control grain size and number. The author suggested that these three genes (and favorable alleles) could be valuable resources for rice breeding programs, as the favorable alleles are presented at low frequency in modern cultivars. Supporting that, they presented evidence that reducing expression of MADS17 and AP2-39 by RNAi approach increase grain yield by improving grain number and grain weight in a modern rice variety.

In summary, the manuscript contains extensive and valuable data, which will be very useful for both basic research on inflorescence development and breeding. Still, I think that in this current version the conclusions and model are not well supported by the data presented. Below I described my comments and suggestion in details.

Main comments:

> I think that mapping and characterization of the natural variation in MADS17 is one of the most important results in the manuscript. The authors presented evidence indicating that a deletion in the 5' UTR of MADS17 can explain the increased grain size and grain number phenotype. They also show that the 5'UTR deletion may reduce protein synthesis, since mRNA expression level is not affected but protein level (as determined by western blots) is reduced in plants carrying the 5' UTR deletion. The authors also used a reporter assay in rice protoplast to test this. In my opinion the western blot results are critical to support the author's hypothesis, however the manuscript did not contain quantitative data, instead only single gel experiments are shown. Since this is one of the most important result of the manuscript, I consider that more evidence is needed. For example, the authors should do experimental replicates in western blots to quantify changes in protein levels.

> In the experiments using transgenic approaches, the authors presented data from single transgenic events. That includes the functional validation of MADS17 (Figure 2b-e and Figure 2f-i), promoter swapping (Figure 2j-m), AP2-39 overexpression (Figure 3f-i) and AP2-39 silencing (Figure 3j-m), MADS1 RNAi (Figure 4d-f), MADS17 and AP2-39 RNAi in elite cultivar (Figure 5). Since observations based on single transgenic events could be bias because of the unpredictable T-DNA insertion, data from at least 2 or 3 independent events is necessary to make conclusions. It is not clear whether the authors characterized more than one event for each transgenic experiment. In my opinion this is critical, and, as example, I discuss here the experiments presented in figure 2. The authors used the comparison pCPL1 vs pCPL2 lines (Figure 2j-m) to infer that sequence variations in the transcribed region are responsible for the grain number and size phenotypes. On the other hand, I understand that CPD-1 line expresses a full genomic sequence (promoter and transcribed sequence) from *O. rufipogon* (Figure 2b-e). Compared to control C418, I noticed that CPD-1 has a larger reduction in grain number, grain size and grain yield per plant than pCPL2 line (which has promoter from C418 and transcribed region from 8IL73). So, based on those single lines (CPD-1 vs pCPL2), it could be inferred that variations in the promoter sequence also contribute to the phenotype. I understand that these differences in the strength of the phenotypes could also be due to just differences in transgene expression levels in the different lines. Therefore, this example show why it is critical to include more than one line in these kinds of analysis.

In addition to that, the authors should provide evidence that their RNAi vectors are specific to the targeted gene. Since they are targeting MADS and AP2 genes that are part of large gene families, it is possible that RNAi affect the expression of other related genes.

> The authors proposed a genetic network involving MADS1>MADS17>AP2-39. I agree that biochemical, transgenic and expression data presented support this model. However, genetic data is missing in the manuscript. Without experimental data showing that genetic interactions between these genes occur in planta, the model is not well supported.

Other comments:

Abstract: I suggest mentioning that COG1 is MADS17.

Fig 1a and 1b are not mentioned in the text.

Line104. CRISPR-Cas9 was used to target the 5'UTR and generate deletions. Knock-out lines were not generated.

Line 108. "degradation of COG1 protein levels". The authors observed lower signal in a western blot, but there is not evidence of protein degradation.

Lines 118-125. It is not clear how authors performed the genotype analysis. Have the authors checked for other alleles? Also, when considering the natural population, did the authors observe changes in grain number and size associated with the 65bp deletion?

Line 129. "355 coexisted DEGs". To understand this number, it would be good if the authors explain the comparisons performed.

Line 136. "was proved to be correlated with rice biomass in the previous study". Yaish et al., also showed that overexpressing AP2-39 resulted in reduced grain number and yield in rice, which should be mentioned.

Line 140. "upstream of the initiation codes". Do the authors mean the "start codon (ATG)"?

Line 151. "To identify the nucleotide polymorphisms at the OsAP2-39 locus responsible for regulating

grain yield related traits, ...". Since no phenotype associated with AP2-39 in natural populations was presented until this point, I suggest rewriting this sentence.

Line 156-157. Have the authors observed changes in grain number in OsAP2-39 T vs OsAP2-39 C?

Line 183. A reference reporting the OsMADS1 alternative splicing is missing. In addition, I think it could be good to check COG1 and AP2-39 expression in OsMADS1 natural variants. This important, because it may indicate if OsMADS1-mediated regulation of COG1 is relevant to control grain size.

Lines 188-190. RNAi-OsMADS1 lines has lower expression of COG1 and AP2-39. Did the author observe changes in grain number or size? As above, this may indicate if OsMADS1-mediated regulation of COG1 is relevant to control grain size and number.

Line 210. "COG1 or OsAP2-39 may increase grain yield in rice breeding." I suggest changing to "COG1 or OsAP2-39 downregulation may increase grain yield in rice breeding".

Discussion. Do the authors think that any of the other 10 AP2 genes differentially expressed in RNA-seq could also control grain size and/or number? Out of curiosity, did the authors find any other gene already described in grain size and grain number control, differently expressed in RNA-seq?

Reviewer #2 (Remarks to the Author):

This manuscript shows that the 65-bp fragment in 5'UTR of COG1 is important for rice grain yield and so on.

Major points:

1. Supplementary Figure 6 shows relative expression of COG1 mRNA in several developing panicles stage. The authors should show western blot analyses of COG1 protein by anti COG1 same as Fig2. r. This manuscript shows many breeding good results by decrease and increase of the 65-bp fragment in 5'UTR of COG1. However, the molecular mechanisms are unclear. There are many incomprehensible results.
2. Fig.2 s shows results 35S promoter_5'UTR(65bp+/-)_LUC system. First of all, the authors should display how calculate relative translation efficiency (TE). TE compared mere two times plus 65bp case with minus one. Fig.2 r3 shows COG1 protein decrease in Mut6 by western blot. Show the translation efficiency also in this case.
3. Furthermore, Supplementary Fig. 8a shows target1&2, PAM1& 2, but no explain and no evidence. A 34-bp deletion in the Mut-6 is important for grain yield than 65-bp deletion. The authors should describe this points. What molecule contacts the target 1&2 sites? At least the authors should address in vitro translation using these different deplete of 5' UTR of mRNA and search translation efficiency.
4. Is RNAi-COG1 in Fig. 4 e and C418 RNAi-COG1 different or not? In Fig.2 s case, 5'UTR(65bp+/-) mRNAs are same level but in minus 65bp case, protein level decrease to one half. In Supplementary Fig.12a, COG1 mRNA level of C418 RNAi-COG1 is 0.4 times. In this case, detect and show COG1 protein level by western bot.

If this manuscript for breeding journal, it maybe is sufficient. But this journal is not only breeding one.

Minor points:

5. Line 74, what means CPD, what abbreviated name ? Usually, CPD means the key gene of brassinosteroid biosynthesis in plant hormone field. Never use CPD. It is confusing.
6. Line102, And also, PV65-bp and AV65-bp were confusion. What is V? Use other words.
7. Line 120, all reader is not rice science field. "3K RG" word needs URL.

Reviewer #3 (Remarks to the Author):

The improvement in crop production remains a challenge in the modern agriculture, because rice yield potential is the sum of the multiplicative integration of three major components (tiller numbers per plant, grain numbers per panicle and 1,000-grain weight), which are typically negatively correlated to one another. Thus, simultaneous increase in tiller number, grain number and grain weight is a key cereal breeding goal. In this manuscript, the authors identified an elite allele COG1, a gene encoding MADS box transcription factor OsMADS17, which enhanced grain number and grain size by decreasing mRNA translation efficiency. Further experiments demonstrated that the regulatory module OsMADS1-OsMADS17-OsAP2-39 increases grain number and grain weight during rice domestication. The topic of this manuscript is general interest for researchers follow trends and development in understanding co-domestication and convergent selection of the yield-related traits in cereal crops, and I think authors should give attention to the following points and suggestions.

- 1, the authors found that all of wild rice harbored the PV-65bp haplotype, its frequency in indica rice was 90% and that in japonica rice was close to 55%, indicating that the favorable haplotype AV-65bp was only occurred in part of the rice cultivars. The title of this manuscript should be modified to more accurate description of the experimental results.
- 2, to understand the functional roles of the COG1/OsMADS17 and OsAP2-39 in the regulation of plant development and growth in rice, such as plant architecture, panicle branching and grain size, and the authors should generate the null mutants by using CRISPR/Cas9 method or mutations.
- 3, as mentioned above, grain yield is controlled by the three major components: the number of tillers per plant, the number of grains per panicle and 1,000-grain weight, and that are typically negatively correlated with one another. In the current version of the manuscript, the authors only described the effects of the OsMADS1-COG1-OsAP2-39 module on the grain number and grain size, how about tiller number? The authors should add these additional information about the effects of the COG1, OsMADS1 and OsAP2-39 genes on tiller number.
- 4, the authors reported that a 65-bp deletion in the 5' untranslated region (5' UTR) was associated with the increased grain number and grain weight through decreasing mRNA translation efficiency. In Fig. 2r, the authors showed the accumulation of the COG1 protein in young panicles by analyzing the three different experimental comparisons. To compare the levels of the COG1 protein in isogenic lines and/or transgenic plants, the authors perform western blot experiment with the same nitrocellulose or PVDF membrane. In addition, they should also quantify the transcript levels of the COG1 gene using the same samples in the main figures.
- 5, in Fig. 2s, the authors should represent the data of the accumulation of LUC protein.
- 6, in Fig.S6, the authors showed the comparisons of transcriptional levels between C418 and its near-isogenic line 8IL73, it will be better to show the comparison of the accumulation of COG1 protein in the various tissues.

Reviewer #4 (Remarks to the Author):

The manuscript entitled "Synchronous increase of grain number and grain weight during rice domestication" by Li et al. describes identification of causal mutation at COG1 locus that controls grain number and weight in rice. Based on QTL analysis of grain traits between cultivated rice C418 and 8IL73, an introgressions line that harbours wild chromosomal segments in C418. Genetic analysis identified a causal mutation of 65-bp deletion in the 5'UTR of COG1 that encodes OsMADS17. Furthermore, downstream and upstream factors of OsAP2-39 and OsMADS1 were identified, respectively, and their roles were investigated. Grain number and weight are important scope in rice breeding but their trade off effect is becoming an obstacle for increasing yields. Although the authors provided a large number of data supporting the genetic module of OsMADS1-COG1-OsAP2-39, there

are some issues that need to be clarified. Addressing following concerns may improve the quality of the manuscript.

Major points

1) QTL analysis

The authors detected a QTL for grain number and weight on chr. 4. However, I do not know the traits are basically controlled by a single locus between the C418 and 8IL73.

Provide results of QTL analysis of markers covering other wild segments introgressed in C418 background. Also provide data about the contribution of the COG1 on phenotypic variance. This is important to clarify the difference of grain size and number between C418 and 8IL73 controlled by COG1 based on single Mendelian inheritance.

2) Complementation test

In the first section of the text, authors showed detection of the QTL controlling grain size and number based on the segregating population between C418 and 8IL73, and further fine mapping analysis identified a candidate gene of OsMADS17. Then, complementation and RNAi experiments were carried out to confirm the causal gene. However, I am puzzled which allele is functional at this stage of gene identification. Authors carried out complementation test by introducing wild allele to C418 without any evidence. I felt this approach is a lack of logical flow. I suggest reconstructing the process for identification and confirmation of the causal mutation. Complementation test needs to be carried out after clarifying the function of two alleles.

3) Transgenic experiments

There are only a single transgenic lines presented in each experiment. As authors suggested that expression or translational control of the related genes are important for the grain size and number, the dosage in copy number of the transgenes is important matter in transgenic experiments. Especially, transgenic experiments were carried out using pCAMBIA, which can be a multi-copy vector, as long as I know. If the results are supported by multiple transgenic lines, I agree the authors conclusions, however the single transgenic line is insufficient generally. I also have a concern for RNAi experiments about their target specificity such as MADS genes. The expression levels missing in the data can be added (i.e. RNAi plants).

4) Wild rice accessions

In supplementary Table 1 of 68 wild rice accessions, there are non-*O. rufipogon* species. Those are *O. officinalis*, *O. longistaminata*, *O. grandiglumis*. I do not understand why different wild rice species were included. Moreover, more than half of wild rice accessions used in this study are from China, which seems to be biased, therefore strongly recommended to analyse wild rice accessions broadly.

5) Selection of the loci in rice domestication.

L. 125 AV-65bp was originated from the mutation in cultivated rice, and then artificial selection were occurred in part of them.

Since the COG1AV-65bp allele was found in 10.1% and 44.2% of indica and japonica cultivars in the 3K rice genome, respectively. This results let me think the mutation was selected relatively recent, thus I do not agree that the locus was under selection in rice domestication. Without evidences (i.e. selective sweep at the locus), removal of "domestication" in the title and text is recommended. I request the authors rephrase this section of the manuscript to tone down their conclusions. COG1 can be improvement gene.

6) Identification of downstream and upstream genes.

Authors identified OsAP2-39 and OsMADS1, as downstream and upstream factors. However, selections of these two genes are biased. Involvement of other factors are left behind. Involvement of other family genes can be addressed in the text.

Minor points

7) Use of "respectively" is grammatically wrong in many part of text. These must be corrected with English edits.

8) Consider about use of "haplotype" in the text.

A haplotype is a group of alleles that are inherited together. Authors explain about a single locus in the work. I think "genotype" or "allele" may be appropriate.

9) L. 23. I do not understand what "Development process of human social" means.

10) Fig. 2a. Provide genotype information for black and white bars as well as result of statistical analysis for "a" and "b". Also, the coding region of COG1(OsMADS17) is about 5.9-Kb region according to the Supplementary Figure 7. The display of the gene in Fig. 2 looks quite large in a mapped region of 19.6kb. I recommend to adjust it.

11) Fig. 2r. Gel blot panels for Mut-6 and pCPL1/2 can be swapped to follow the text.

12) Fig. 2t. Some cultivars are removed form rice 3K (i.e. $n=1760+819=2579<3000$). Need to clarify procedure of these removals and explain the removal is reasonable.

13) Fig. 4k. I could not find yellow circles in the map.

14) Supplementary Table 1. Clarify about 'NS' in the table. Need a space between Oryza (O.) and rufipogon.

15) Supplementary Figure 5. Expression of COG1 in two transgenic lines are very different among lines. This can be due to copy numbers of the transgene or other factors. I do not know this is technical or biological replications. Please clarify these concerns.

16) Supplementary Figure 6. L.668. I afraid I do not understand what "pulvin" means. Is this a "node"? For in situ hybridization, provide plant information (C418 or 8IL73) for sense probe as negative control (b).

17) Supplementary Figure 11. Authors studied grain size and OsAP2-39 expression level using 9 cultivars. The reason for the selection of these cultivars as well as their genotypic information for COG1 needs to be provided. To meet up with authors idea, COG1 genotype should be identical to discuss about the effect of OsAP2-39. In addition, 4542A3-49B-2B12 is a name of cultivar?

18) Use same style for plant materials. For example, C418RNAi-COG1 and RNAi-OsAP2-39-1.

19) In discussion section, how the COG1 overcome the trade-off effect between grain and weight can be discussed. This can be an important point for the discovery of COG1.

20) L. 172. What "mutated cultivate rice" means?

Dear Reviewers:

Thank you for your letter and the reviewers' comments concerning our manuscript entitled "*Synchronous increase of grain number and grain weight during rice domestication*" (ID: NCOMMS-22-20932A). Those comments are all valuable and very helpful for revising and improving our paper.

We have studied comments carefully and have revised the manuscript according the valuable suggestions from you and the reviewers. The main corrections in the manuscript and a detailed point-by-point response addressing the reviewers' comments and queries are described below. We would be grateful if our revised manuscript can be accepted for publication in the *Nature Communications*.

Thank you very much for your time and efforts to take care of our manuscript.

Sincerely yours,

Chuanqing Sun

Professor, Department of Plant Genetics and Breeding,

China Agricultural University, Beijing 100193, China

REVIEWER COMMENTS

Reviewer #1 (Remarks to the Author):

Several works have identified multiple genes and pathways that control grain number or grain size in cereals, including rice, maize or wheat. However, most of the time those mechanisms modify only one of those traits, and on some occasion, negative correlation between grain size and number have been observed.

In this manuscript Li and collaborators described a genetic pathway to simultaneously control grain size and number in rice.

Initially, the author identified and characterized a QTL, named CONTROL OF GRAIN YIELD 1 (COG1) affecting both traits in a wild introgression line, which harbor chromosomal segments from a donor wild rice into an elite japonica cultivar. Using a fine mapping approach, they identified a MADS-box gene (OsMADS17) as responsible of the phenotype, which was validated by functional complementation and RNAi strategies.

Analysis of genetic variations at MADS17 between cultivars revealed a 65bp deletion in the 5'UTR of the favorable allele that confers more and larger grain phenotype. Importantly, the author used CRISPR-Cas9 to generate a similar 36bp deletion in the 5'UTR that phenocopy the natural 65bp deletion. Since the deletion does not affect transcript level, but reduce

protein levels, the author concluded that the 65bp deletion reduces mRNA translation.

Next, the authors were interested in identifying downstream genes of MADS17, and performed RNA-seq in developing apices of plants with different levels of MADS17. They identified a group of 11 AP2 transcription factors, and selected one of them (AP2-39) for functional analysis. Overexpression of AP2-39 resulted in less and smaller grains, while RNAi resulted in more and larger grains, similar to the phenotypes described in plants with reduced levels of MADS17. Next the authors performed biochemical analysis and presented evidence that suggest that MADS17 is a positive regulator of AP2-39. Interestingly, the author also identified natural variation associated with AP2-39 that affect grain weight, with the favorable allele being at low frequency in cultivated rice.

Next, the authors were interested in genes acting upstream of MADS17. They explored MADS1, which is an important regulator of panicle and spikelet development in rice. Plants with reduced expression of MADS1 had low expression of MADS17 and AP2-39, and biochemical analysis suggest that MADS1 could regulate MADS17 expression directly.

Finally, the author proposed a gene network involving these three genes acting sequentially to control grain size and number. The author suggested that these three genes (and favorable alleles) could be valuable resources for rice breeding programs, as the favorable alleles are presented at low

frequency in modern cultivars. Supporting that, they presented evidence that reducing expression of MADS17 and AP2-39 by RNAi approach increase grain yield by improving grain number and grain weight in a modern rice variety.

In summary, the manuscript contains extensive and valuable data, which will be very useful for both basic research on inflorescence development and breeding. Still, I think that in this current version the conclusions and model are not well supported by the data presented. Below I described my comments and suggestion in details.

Main comments:

> I think that mapping and characterization of the natural variation in MADS17 is one of the most important results in the manuscript. The authors presented evidence indicating that a deletion in the 5' UTR of MADS17 can explain the increased grain size and grain number phenotype. They also show that the 5'UTR deletion may reduce protein synthesis, since mRNA expression level is not affected but protein level (as determined by western blots) is reduced in plants carrying the 5' UTR deletion. The authors also used a reporter assay in rice protoplast to test this. In my opinion the western blot results are critical to support the author's hypothesis, however the manuscript did not contain quantitative data, instead only single gel experiments are shown. Since this is one of

the most important result of the manuscript, I consider that more evidence is needed. For example, the authors should do experimental replicates in western blots to quantify changes in protein levels.

> In the experiments using transgenic approaches, the authors presented data from single transgenic events. That includes the functional validation of MADS17 (Figure 2b-e and Figure 2f-i), promoter swapping (Figure 2j-m), AP2-39 overexpression (Figure 3f-i) and AP2-39 silencing (Figure 3j-m), MADS1 RNAi (Figure 4d-f), MADS17 and AP2-39 RNAi in elite cultivar (Figure 5). Since observations based on single transgenic events could be bias because of the unpredictable T-DNA insertion, data from at least 2 or 3 independent events is necessary to make conclusions. It is not clear whether the authors characterized more than one event for each transgenic experiment. In my opinion this is critical, and, as example, I discuss here the experiments presented in figure 2. The authors used the comparison pCPL1 vs pCPL2 lines (Figure 2j-m) to infer that sequence variations in the transcribed region are responsible for the grain number and size phenotypes. On the other hand, I understand that CPD-1 line expresses a full genomic sequence (promoter and transcribed sequence) from *O. rufipogon* (Figure 2b-e). Compared to control C418, I noticed that CPD-1 has a larger reduction in grain number, grain size and grain yield per plant than pCPL2 line (which has promoter from C418 and

transcribed region from 8IL73). So, based on those single lines (CPD-1 vs pCPL2), it could be inferred that variations in the promoter sequence also contribute to the phenotype. I understand that these differences in the strength of the phenotypes could also be due to just differences in transgene expression levels in the different lines. Therefore, this example show why it is critical to include more than one line in these kinds of analysis.

In addition to that, the authors should provide evidence that their RNAi vectors are specific to the targeted gene. Since they are targeting MADS and AP2 genes that are part of large gene families, it is possible that RNAi affect the expression of other related genes.

> The authors proposed a genetic network involving MADS1>MADS17>AP2-39. I agree that biochemical, transgenic and expression data presented support this model. However, genetic data is missing in the manuscript. Without experimental data showing that genetic interactions between these genes occur in planta, the model is not well supported.

Other comments:

Abstract: I suggest mentioning that COG1 is MADS17.

Fig 1a and 1b are not mentioned in the text.

Line104. CRISPR-Cas9 was used to target the 5'UTR and generate deletions. Knock-out lines were not generated.

Line 108. “degradation of COG1 protein levels”. The authors observed lower signal in a western blot, but there is not evidence of protein degradation.

Lines 118-125. It is not clear how authors performed the genotype analysis. Have the authors checked for other alleles? Also, when considering the natural population, did the authors observe changes in grain number and size associated with the 65bp deletion?

Line 129. “355 coexisted DEGs”. To understand this number, it would be good if the authors explain the comparisons performed.

Line 136. “was proved to be correlated with rice biomass in the previous study”. Yaish et al., also showed that overexpressing AP2-39 resulted in reduced grain number and yield in rice, which should be mentioned.

Line 140. “upstream of the initiation codes”. Do the authors mean the “start codon (ATG)”?

Line 151. “To identify the nucleotide polymorphisms at the OsAP2-39 locus responsible for regulating grain yield related traits, ...”. Since no phenotype associated with AP2-39 in natural populations was presented until this point, I suggest rewriting this sentence.

Line 156-157. Have the authors observed changes in grain number in OsAP2-39 T vs OsAP2-39 C?

Line 183. A reference reporting the OsMADS1 alternative splicing is missing. In addition, I think it could be good to check COG1 and AP2-39

expression in OsMADS1 natural variants. This important, because it may indicate if OsMADS1-mediated regulation of COG1 is relevant to control grain size.

Lines 188-190. RNAi-OsMADS1 lines has lower expression of COG1 and AP2-39. Did the author observe changes in grain number or size? As above, this may indicate if OsMADS1-mediated regulation of COG1 is relevant to control grain size and number.

Line 210. "COG1 or OsAP2-39 may increase grain yield in rice breeding." I suggest changing to "COG1 or OsAP2-39 downregulation may increase grain yield in rice breeding".

Discussion. Do the authors think that any of the other 10 AP2 genes differentially expressed in RNA-seq could also control grain size and/or number? Out of curiosity, did the authors find any other gene already described in grain size and grain number control, differently expressed in RNA-seq?

Reviewer #2 (Remarks to the Author):

This manuscript shows that the 65-bp fragment in 5'UTR of COG1 is important for rice grain yield and so on.

Major points:

1. Supplementary Figure 6 shows relative expression of COG1 mRNA in

several developing panicles stage. The authors should show western blot analyses of COG1 protein by anti COG1 same as Fig2. r. This manuscript shows many breeding good results by decrease and increase of the 65-bp fragment in 5'UTR of COG1. However, the molecular mechanisms are unclear. There are many incomprehensible results.

2. Fig.2 s shows results 35S promoter_5'UTR(65bp+/-)_LUC system. First of all, the authors should display how calculate relative translation efficiency (TE). TE compared mere two times plus 65bp case with minus one. Fig.2 r3 shows COG1 protein decrease in Mut6 by western blot. Show the translation efficiency also in this case.

3. Furthermore, Supplementary Fig. 8a shows target1&2, PAM1& 2, but no explain and no evidence. A 34-bp deletion in the Mut-6 is important for grain yield than 65-bp deletion. The authors should describe this points. What molecule contacts the target 1&2 sites? At least the authors should address in vitro translation using these different deplete of 5' UTR of mRNA and search translation efficiency.

4. Is RNAi-COG1 in Fig. 4 e and C418 RNAi-COG1 different or not? In Fig.2 s case, 5'UTR(65bp+/-) mRNAs are same level but in minus 65bp case, protein level decrease to one half. In Supplementary Fig.12a, COG1 mRNA level of C418 RNAi-COG1 is 0.4 times. In this case, detect and show COG1 protein level by western bot.

If this manuscript for breeding journal, it maybe is sufficient. But this journal is not only breeding one.

Minor points:

5. Line 74, what means CPD, what abbreviated name ? Usually, CPD means the key gene of brassinosteroid biosynthesis in plant hormone field.

Never use CPD. It is confusing.

6. Line102, And also, PV65-bp and AV65-bp were confusion. What is V?

Use other words.

7. Line 120, all reader is not rice science field. “3K RG” word needs URL.

Reviewer #3 (Remarks to the Author):

The improvement in crop production remains a challenge in the modern agriculture, because rice yield potential is the sum of the multiplicative integration of three major components (tiller numbers per plant, grain numbers per panicle and 1,000-grain weight), which are typically negatively correlated to one another. Thus, simultaneous increase in tiller number, grain number and grain weight is a key cereal breeding goal. In this manuscript, the authors identified an elite allele COG1, a gene encoding MADS box transcription factor OsMADS17, which enhanced grain number and grain size by decreasing mRNA translation efficiency. Further experiments demonstrated that the regulatory module OsMADS1-

OsMADS17-OsAP2-39 increases grain number and grain weight during rice domestication. The topic of this manuscript is general interest for researchers follow trends and development in understanding co-domestication and convergent selection of the yield-related traits in cereal crops, and I think

authors should give attention to the following points and suggestions.

1, the authors found that all of wild rice harbored the PV-65bp haplotype, its frequency in indica rice was 90% and that in japonica rice was close to 55%, indicating that the favorable haplotype AV-65bp was only occurred in part of the rice cultivars. The title of this manuscript should be modified to more accurate description of the experimental results.

2, to understand the functional roles of the COG1/OsMADS17 and OsAP2-39 in the regulation of plant development and growth in rice, such as plant architecture, panicle branching and grain size, and the authors should generate the null mutants by using CRISPR/Cas9 method or mutations.

3, as mentioned above, grain yield is controlled by the three major components: the number of tillers per plant, the number of grains per panicle and 1,000-grain weight, and that are typically negatively correlated with one another. In the current version of the manuscript, the authors only described the effects of the OsMADS1-COG1-OsAP2-39 module on the grain number and grain size, how about tiller number? The authors should

add these additional information about the effects of the COG1, OsMADS1 and OsAP2-39 genes on tiller number.

4, the authors reported that a 65-bp deletion in the 5' untranslated region (5' UTR) was associated with the increased grain number and grain weight through decreasing mRNA translation efficiency. In Fig. 2r, the authors showed the accumulation of the COG1 protein in young panicles by analyzing the three different experimental comparisons. To compare the levels of the COG1 protein in isogenic lines and/or transgenic plants, the authors perform western blot experiment with the same nitrocellulose or PVDF membrane. In addition, they should also quantify the transcript levels of the COG1 gene using the same samples in the main figures.

5, in Fig. 2s, the authors should represent the data of the accumulation of LUC protein.

6, in Fig.S6, the authors showed the comparisons of transcriptional levels between C418 and its near-isogenic line 8IL73, it will be better to show the comparison of the accumulation of COG1 protein in the various tissues.

Reviewer #4 (Remarks to the Author):

The manuscript entitled "Synchronous increase of grain number and grain weight during rice domestication" by Li et al. describes identification of causal mutation at COG1 locus that controls grain number and weight in

rice. Based on QTL analysis of grain traits between cultivated rice C418 and 8IL73, an introgressions line that harbours wild chromosomal segments in C418. Genetic analysis identified a causal mutation of 65-bp deletion in the 5'UTR of COG1 that encodes OsMADS17. Furthermore, downstream and upstream factors of OsAP2-39 and OsMADS1 were identified, respectively, and their roles were investigated. Grain number and weight are important scope in rice breeding but their trade off effect is becoming an obstacle for increasing yields. Although the authors provided a large number of data supporting the genetic module of OsMADS1-COG1-OsAP2-39, there are some issues that need to be clarified. Addressing following concerns may improve the quality of the manuscript.

Major points

1) QTL analysis

The authors detected a QTL for grain number and weight on chr. 4. However, I do not know the traits are basically controlled by a single locus between the C418 and 8IL73.

Provide results of QTL analysis of markers covering other wild segments introgressed in C418 background. Also provide data about the contribution of the COG1 on phenotypic variance. This is important to clarify the difference of grain size and number between C418 and 8IL73 controlled by COG1 based on single Mendelian inheritance.

2) Complementation test

In the first section of the text, authors showed detection of the QTL controlling grain size and number based on the segregating population between C418 and 8IL73, and further fine mapping analysis identified a candidate gene of OsMADS17. Then, complementation and RNAi experiments were carried out to confirm the causal gene. However, I am puzzled which allele is functional at this stage of gene identification. Authors carried out complementation test by introducing wild allele to C418 without any evidence. I felt this approach is a lack of logical flow. I suggest reconstructing the process for identification and confirmation of the causal mutation. Complementation test needs to be carried out after clarifying the function of two alleles.

3) Transgenic experiments

There are only a single transgenic lines presented in each experiment. As authors suggested that expression or translational control of the related genes are important for the grain size and number, the dosage in copy number of the transgenes is important matter in transgenic experiments. Especially, transgenic experiments were carried out using pCAMBIA, which can be a multi-copy vector, as long as I know. If the results are supported by multiple transgenic lines, I agree the authors conclusions, however the single transgenic line is insufficient generally. I also have a concern for RNAi experiments about their target specificity such as MADS

genes. The expression levels missing in the data can be added (i.e. RNAi plants).

4) Wild rice accessions

In supplementary Table 1 of 68 wild rice accessions, there are non-*O. rufipogon* species. Those are *O. officinalis*, *O. longistaminata*, *O. grandiglumis*. I do not understand why different wild rice species were included. Moreover, more than half of wild rice accessions used in this study are from China, which seems to be biased, therefore strongly recommended to analyse wild rice accessions broadly.

5) Selection of the loci in rice domestication.

L. 125 AV-65bp was originated from the mutation in cultivated rice, and then artificial selection were occurred in part of them.

Since the COG1AV-65bp allele was found in 10.1% and 44.2% of indica and japonica cultivars in the 3K rice genome, respectively. This results let me think the mutation was selected relatively recent, thus I do not agree that the locus was under selection in rice domestication. Without evidences (i.e. selective sweep at the locus), removal of “domestication” in the title and text is recommended. I request the authors rephrase this section of the manuscript to tone down their conclusions. COG1 can be improvement gene.

6) Identification of downstream and upstream genes.

Authors identified OsAP2-39 and OsMADS1, as downstream and

upstream factors. However, selections of these two genes are biased. Involvement of other factors are left behind. Involvement of other family genes can be addressed in the text.

Minor points

7) Use of “respectively” is grammatically wrong in many part of text. These must be corrected with English edits.

8) Consider about use of “haplotype” in the text.

A haplotype is a group of alleles that are inherited together. Authors explain about a single locus in the work. I think “genotype” or “allele” may be appropriate.

9) L. 23. I do not understand what “Development process of human social” means.

10) Fig. 2a. Provide genotype information for black and white bars as well as result of statistical analysis for “a” and “b”. Also, the coding region of COG1(OsMADS17) is about 5.9-Kb region according to the Supplementary Figure 7. The display of the gene in Fig. 2 looks quite large in a mapped region of 19.6kb. I recommend to adjust it.

11) Fig. 2r. Gel blot panels for Mut-6 and pCPL1/2 can be swapped to follow the text.

12) Fig. 2t. Some cultivars are removed form rice 3K (i.e. $n=1760+819=2579$)

- 13) Fig. 4k. I could not find yellow circles in the map.
- 14) Supplementary Table 1. Clarify about 'NS' in the table. Need a space between *Oryza* (O.) and *rufipogon*.
- 15) Supplementary Figure 5. Expression of COG1 in two transgenic lines are very different among lines. This can be due to copy numbers of the transgene or other factors. I do not know this is technical or biological replications. Please clarify these concerns.
- 16) Supplementary Figure 6. L.668. I afraid I do not understand what "pulvin" means. Is this a "node"? For in situ hybridization, provide plant information (C418 or 8IL73) for sense probe as negative control (b).
- 17) Supplementary Figure 11. Authors studied grain size and OsAP2-39 expression level using 9 cultivars. The reason for the selection of these cultivars as well as their genotypic information for COG1 needs to be provided. To meet up with authors idea, COG1 genotype should be identical to discuss about the effect of OsAP2-39. In addition, 4542A3-49B-2B12 is a name of cultivar?
- 18) Use same style for plant materials. For example, C418RNAi-COG1 and RNAi-OsAP2-39-1.
- 19) In discussion section, how the COG1 overcome the trade-off effect between grain and weight can be discussed. This can be an important point for the discovery of COG1.
- 20) L. 172. What "mutated cultivate rice" means?

We are appreciated for the reviewers' careful review and valuable suggestions. According to the reviewers' suggestions, we have revised the manuscript carefully. A detailed point-by-point response addressing the reviewers' comments and queries are described below.

Reviewer #1's questions and our responses:

R1Q1 (Reviewer #1's question 1):, the manuscript contains extensive and valuable data, which will be very useful for both basic research on inflorescence development and breeding.

Our response to R1Q1: Thank you very much for your appreciation.

R1Q2 (Reviewer #1's question 2): I think that mapping and characterization of the natural variation in MADS17 is one of the most important results in the manuscript. The authors presented evidence indicating that a deletion in the 5' UTR of MADS17 can explain the increased grain size and grain number phenotype. They also show that the 5'UTR deletion may reduce protein synthesis, since mRNA expression level is not affected but protein level (as determined by western blots) is reduced in plants carrying the 5' UTR deletion. The authors also used a reporter assay in rice protoplast to test this. In my opinion the western blot results are critical to support the author's hypothesis, however the manuscript did not contain quantitative data, instead only single gel experiments are shown. Since this is one of the most important result of the

manuscript, I consider that more evidence is needed. For example, the authors should do experimental replicates in western blots to quantify changes in protein levels.

Our response to R1Q2: Thank you for your suggestions. We have followed your suggestions to quantify COG1 protein levels in the young panicles of C418, introgression line 8IL73, and different transgenic plants by western blots with 3 replications. It showed that 8IL73 (with a 65-bp insertion in the 5' UTR of *COG1*) and C418^{CTP} plants (complement transgenic plants harboring the *COG1* gene from 8IL73 in C418 background) had higher COG1 protein accumulation than C418 plants (with the 65-bp deletion) (Fig. 3p, q in the revised manuscript). The transgenic plants C418^{CTP-CW} (having the 2.5-kb *COG1* promoter from C418 fused with the *COG1* transcript region from 8IL73 in C418 background) accumulated more COG1 protein contents than C418 and the transgenic plants C418^{CTP-WC} (harboring the 2.5-kb *COG1* promoter from 8IL73 fused with the *COG1* transcript region from C418 in C418 background), and the transgenic plants C418^{CTP-WC} showed no obvious change of COG1 protein levels compared to the C418 plants (Fig. 3r in the revised manuscript). Correspondingly, the transgenic plants 8IL73^{RNAi-COG1} (interfering the *COG1* gene in 8IL73 plants) and C418^{RNAi-COG1} (interfering the *COG1* gene in C418 plants) displayed decreased COG1 protein

accumulation compared to the 8IL73 and C418, respectively (Fig. 3s, Fig. 6f in the revised manuscript). These results indicated that the transcript region of *COG1* from 8IL73 generated higher COG1 protein levels than that in C418 plants. Further investigations of the COG1 protein levels in the transgenic plants generated by CRISPR/Cas9 system targeting the 65-bp in 8IL73 plants showed that COG1 protein levels of the Mut-6 (with 34-bp deletion in 5' UTR of the *COG1* gene) were declined significantly compared to 8IL73 (Fig. 3t in the revised manuscript). Taken together, the 65-bp deletion in 5' UTR of the *COG1* gene declined the accumulation of COG1 protein.

We have added the new results in the revised manuscript in Figure 3p-t and Figure 6f. The descriptions for the new results are shown as follows:

In line 120-126: we carried out western blot assays and found that C418 contained lower COG1 protein levels than that of 8IL73 in young panicles (Fig. 3p, Supplementary Fig. 5b). Further analysis for COG1 protein levels in transgenic plants showed that C418^{CTP} and C418^{CTP-CW} contained higher COG1 protein levels than in C418 plants (Fig. 3q, r), although no significant change of COG1 protein levels were apparent between C418^{CTP-WC} and C418 plants (Fig. 3r). Correspondingly, the transgenic plants 8IL73^{RNAi-COG1} had lower COG1 protein accumulation than did the control (Fig. 3s).

In line 142-145: only the Mut-6 plants with the 34-bp deletion containing the F3 fragment exhibited significantly increased grain number, grain weight, and grain yield compared to 8IL73, accompanied by decreased TE and accumulation levels of COG1 protein

In line 289-292: Downregulation of COG1 expression levels in the transgenic plants C418^{RNAi-COG1} caused lower levels of COG1 protein accumulation, higher grain number, bigger grain size, and higher grain yield compared with the controls (Fig. 6a-i, Supplementary Fig. 19).

R1Q3 (Reviewer #1's question 3): In the experiments using transgenic approaches, the authors presented data from single transgenic events. That includes the functional validation of MADS17 (Figure 2b-e and Figure 2f-i), promoter swapping (Figure 2j-m), AP2-39 overexpression (Figure 3f-i) and AP2-39 silencing (Figure 3j-m), MADS1 RNAi (Figure 4d-f), MADS17 and AP2-39 RNAi in elite cultivar (Figure 5). Since observations based on single transgenic events could be bias because of the unpredictable T-DNA insertion, data from at least 2 or 3 independent events is necessary to make conclusions. It is not clear whether the authors characterized more than one event for each transgenic experiment. In my opinion this is critical, and, as example, I discuss here the experiments presented in figure 2. The authors used the comparison pCPL1 vs pCPL2 lines (Figure 2j-m) to infer that sequence variations in the transcribed

region are responsible for the grain number and size phenotypes. On the other hand, I understand that CPD-1 line expresses a full genomic sequence (promoter and transcribed sequence) from *O. rufipogon* (Figure 2b-e). Compared to control C418, I noticed that CPD-1 has a larger reduction in grain number, grain size and grain yield per plant than pCPL2 line (which has promoter from C418 and transcribed region from 8IL73). So, based on those single lines (CPD-1 vs pCPL2), it could be inferred that variations in the promoter sequence also contribute to the phenotype. I understand that these differences in the strength of the phenotypes could also be due to just differences in transgene expression levels in the different lines. Therefore, this example shows why it is critical to include more than one line in these kinds of analysis.

In addition to that, the authors should provide evidence that their RNAi vectors are specific to the targeted gene. Since they are targeting MADS and AP2 genes that are part of large gene families, it is possible that RNAi affect the expression of other related genes.

Our response to R1Q3: Thank you for your constructive suggestions.

(1) In fact, we obtained more than three independent transgenic lines for each construct. According to your suggestion, we added the data of expression levels and plant phenotype of two independent transgenic lines. Phenotype analysis using three transgenic lines also showed that grain number, grain weight, and grain yield of the complement transgenic plants

C418^{CTP} and C418^{CTP-CW} were declined compared to the C418 plants, while the C418^{CTP-WC} transgenic plants had no obvious change in grain yield related traits than C418 plants. Particularly, the line C418^{CTP}-2 showed a similar reduction in grain number (22.9% VS 19.5%), grain size (9.8% VS 8.1%), and grain yield per plant (21.5% VS 23.8%) with the C418^{CTP-CW}-2 line (Fig. 2f-h, Fig. 3e-g in the revised manuscript).

We also found that all lines of transgenic plants 8IL73^{RNAi-CO₁}, C418^{RNAi-CO₁}, and C418^{RNAi-OsAP2-39} showed increase of grain number, grain weight, and grain yield compared to the controls (Fig. 2m-o, Fig. 6g-i, Fig. 4m in the revised manuscript). The three independent lines of transgenic plants C418^{OE-OsAP2-39} displayed decrease of grain number, grain weight, and grain yield than the control (Fig. 4i in the revised manuscript). We also analyzed the expression levels using three independent transgenic lines, and found that the expression levels of *COG1* and *O_sAP2-39* were accordant with the previous results.

(2) The specificity of targeted gene sequence for RNAi experiments is important, especially for those with multiple homologous genes, such as MADS family and *AP2* family genes. Therefore, we carefully considered the specificity of targeted gene sequence when we generated the RNAi constructions for *COG1* and *O_sAP2-39*. qRT-PCR analysis for the 13 MADS family genes and 13 *AP2* family genes showed none of them exhibited significant alteration of expression levels in RNAi transgenic

plants compared to the controls except for the targeted genes *COG1* and *OsAP2-39* (Response Fig. 1), indicating that the RNAi vector for *COG1* or *OsAP2-39* was specific to the targeted gene. We have added the contents in the part of methods in the revised manuscript in line 385-386.

Response Fig. 1 Expression levels of *MADS* (a) and *AP2* (b) family genes.

R1Q4 (Reviewer #1's question 4): The authors proposed a genetic network involving MADS1>MADS17>AP2-39. I agree that biochemical, transgenic and expression data presented support this model. However, genetic data is missing in the manuscript. Without experimental data showing that genetic interactions between these genes occur in planta, the

model is not well supported.

Our response to R1Q4: Thank you for your appreciation. Experimental data of genetic interactions between these genes may well support the genetic module involving *OsMADS1*, *COG1*, and *OsAP2-39*. We first analyzed the functions of different alleles of these genes with respect to regulation of grain yield-related traits, based on the genotypes and phenotypes of accessions from the 3K RG (RFGB v2.0 database, <https://www.rmbreeding.cn/>). We found that the cultivars carrying the genotypes *OsMADS1*^{AS}, *COG1*^{De65-bp}, or *OsAP2-39*^T showed higher individual grain weight than those with the genotypes *OsMADS1*^{NS}, *COG1*^{In65-bp}, or *OsAP2-39*^C, respectively (Supplementary Fig. 11c, Supplementary Fig. 17a in the revised manuscript). The individual grain weight of the cultivars carrying both *OsMADS1*^{AS} and *OsAP2-39*^T was significantly higher than that in cultivars containing only *OsMADS1*^{AS} or *OsAP2-39*^T (Supplementary Fig. 17b in the revised manuscript). In addition, cultivars with the genotype *COG1*^{AS} and *OsAP2-39*^T produced larger grains than those carrying only *COG1*^{AS} or *OsAP2-39*^T (Supplementary Fig. 17c in the revised manuscript). These results suggested that genes *OsMADS1*, *COG1*, and *OsAP2-39* functioned together to regulate individual grain weight. Besides, phenotypic analysis for ZH17 and null mutants showed that the single mutants *cog1* or *osap2-39* exhibited significantly increased individual grain weight and grain yield

compared with the control (Supplementary Fig. 18b-o in the revised manuscript), while the double mutant *cog1/osap2-39* further increased grain weight and grain yield compared with the single mutants *cog1* or *osap2-39* (Supplementary Fig. 18b-o in the revised manuscript), indicating that *COG1* and *OsAP2-39* have additive effects on grain yield regulation. We have added these results in the revised manuscript in line 244-262.

R1Q5 (Reviewer #1's question 5): Abstract: I suggest mentioning that COG1 is MADS17.

Our response to R1Q5: We have followed your suggestion to add the words “which is identical to *OsMADS17*” in line 18 in the revised manuscript. Thank you!

R1Q6 (Reviewer #1's question 6): Fig 1a and 1b are not mentioned in the text.

Our response to R1Q6: Thank you for your suggestion. Fig 1a and 1b are cited in the revised manuscript in line 31.

R1Q7 (Reviewer #1's question 7): Line104. CRISPR-Cas9 was used to target the 5'UTR and generate deletions. Knock-out lines were not generated.

Our response to R1Q7: We have revised this sentence to “To further assess the function of the 65-bp sequence for increasing TE of the *COG1* gene in rice plants, we selected the F3 fragment which was located at the

interval of PAM1 (GGG) and PAM2 (TGG) as the target for genome editing by the CRISPR/Cas9 system in 8IL73 plants (Supplementary Fig. 10a).” in line 139-142 in the revised manuscript. Thank you!

R1Q8 (Reviewer #1’s question 8): Line 108. “degradation of COG1 protein levels”. The authors observed lower signal in a western blot, but there is not evidence of protein degradation.

Our response to R1Q8: Indeed, our works were focused on the variation of the 65-bp in 5’ UTR of *COG1* changing translation efficiency resulted in alteration of the COG1 protein accumulation rather than degradation. We have made the correction as “accompanied by decreased TE and accumulation levels of COG1 protein” in line 144-145 in the revised manuscript. Thank you!

R1Q9 (Reviewer #1’s question 9): Lines 118-125. It is not clear how authors performed the genotype analysis. Have the authors checked for other alleles? Also, when considering the natural population, did the authors observe changes in grain number and size associated with the 65bp deletion?

Our response to R1Q9: Thank you for your suggestion.

(1) Sequence analysis showed a set of ten SNPs and six Indels in *COG1* transcript region between C418 and 8IL73. Among these variations, a synonymous mutation and a 65-bp insertion/deletion existed in the

coding region and the 5' UTR, respectively (Supplementary Fig. 4 in the revised manuscript). We had convinced the Indel of 65-bp was the critical functional site for grain yield regulation. Therefore, we checked for the 65-bp Indel in 5' UTR of *COGI* among 92 accessions of wild rice and 169 varieties of cultivated rice (including 79 *indica* and 90 *japonica* varieties). The results showed that 92 accessions of wild rice all harbored the 65-bp insertion, while 111 cultivars contained the 65-bp insertion, and the rest of 58 cultivars had the 65-bp deletion, and no else allele was found (Supplementary Table 2 in the revised manuscript).

(2) Further investigating the relationship between the 65-bp Indel and grain number and grain weight among the 169 cultivated rice showed that 58 cultivars with the 65-bp deletion (*COGI*^{De65-bp}) displayed higher grain weight (25.79 ± 3.20 g) than that (23.35 ± 3.21 g) of 111 cultivars with the 65-bp insertion (*COGI*^{In65-bp}), while no significant difference of grain number was observed between the two groups (167.57 ± 32.50 VS 169.45 ± 30.54) (Supplementary Figure 11a, b in the revised manuscript). We further analyzed the relationship between the 65-bp Indel and grain weight using 1,591 cultivars from 3K RG (RFGB v2.0 database, <https://www.rmbreeding.cn/>). It showed that 403 cultivars containing the genotype *COGI*^{De65-bp} showed higher grain weight than the rest of 1,188 cultivars with the genotype *COGI*^{In65-bp} (Supplementary Fig. 11c in the

revised manuscript). These results proved that the 65-bp deletion in the *COG1* gene could increase grain weight.

We have rewritten this paragraph in the revised manuscript in line 148-167 as follows:

We screened 92 accessions of wild rice and 169 varieties of cultivated rice (including 79 *indica* and 90 *japonica* varieties) for the 65-bp indel in the 5' UTR of *COG1* and found that all 92 accessions of wild rice harbored the 65-bp insertion (Supplementary Table 2), whereas 111 of the cultivars contained the In65-bp, with the rest of 58 cultivars carrying the De65-bp; no other allele was found (Supplementary Data 1). We further analyzed the distribution of the 65-bp indel in 2,579 cultivated rice accessions from the 3K RG (RFGB v2.0 database, <https://www.rmbreeding.cn/>). The distribution study showed that the frequency of cultivars with the genotype *COG1*^{De-65bp} in japonica rice and indica rice was 44.2% and 10.1%, respectively (Fig. 3w), with the japonica rice accessions with *COG1*^{De-65bp} being distributed mainly in East Asia, whereas the indica rice accessions with *COG1*^{De-65bp} were mainly distributed in Southeast Asia (Fig. 3x).

To investigate the relationship between the 65-bp indel and grain number or grain weight, we analyzed 169 cultivated rice accessions with respect to the phenotype and the genotype at the *COG1* locus. Analysis showed that the 58 cultivars with the genotype *COG1*^{De65-bp} displayed

significantly higher individual grain weight than the 111 cultivars with the genotype *COGI*^{In65-bp}, whereas no significant difference of grain number was observed between the two groups (Supplementary Fig. 11a, b). Further analysis of 1,591 cultivars from 3K RG showed that 403 cultivars containing the genotype *COGI*^{De65-bp} exhibited significantly higher grain weight than the 1,188 cultivars with the genotype *COGI*^{In65-bp} (Supplementary Fig. 11c). These results confirm that the 65-bp deletion in the 5' UTR of the *COGI* gene originated from the mutation in cultivated rice.

R1Q10 (Reviewer #1's question 10): Line 129. "355 coexisted DEGs". To understand this number, it would be good if the authors explain the comparisons performed.

Our response to R1Q10: Thank you for your suggestion. We have added the contents in the revised manuscript in line 171-174 as follows:

As a result, 536 genes were shown to be significantly differentially expressed between C418 and 8IL73, whereas 953 differentially expressed genes (DEGs) were detected between 8IL73^{RNAi-COGI} and 8IL73 (Supplementary Fig. 12a). We identified 355 overlapping DEGs between the two comparative groups for analysis with respect to the Gene Ontology (GO) and Kyoto Encyclopedia of Genes and Genomes (KEGG) databases.

R1Q11 (Reviewer #1's question 11): Line 136. “was proved to be correlated with rice biomass in the previous study”. Yaish et al., also showed that overexpressing AP2-39 resulted in reduced grain number and yield in rice, which should be mentioned.

Our response to R1Q11: We have revised this sentence to “*OsAP2-39 (LOC_Os04g52090)* expression was proved to be negatively correlated with rice grain number and yield in a previous study⁴⁴” in line 180-181 in the revised manuscript. Thank you!

R1Q12 (Reviewer #1's question 12): Line 140. “upstream of the initiation codes”. Do the authors mean the “start codon (ATG)”?

Our response to R1Q12: Yes, it means the start codon (ATG). For better understanding, we have revised this description as “start codon (ATG)” in line 184 in the revised manuscript. Thank you!

R1Q13 (Reviewer #1's question 13): Line 151. “To identify the nucleotide polymorphisms at the *OsAP2-39* locus responsible for regulating grain yield related traits, ...”. Since no phenotype associated with AP2-39 in natural populations was presented until this point, I suggest rewriting this sentence.

Our response to R1Q13: Thanks for your suggestion. We have rewritten this sentence as “To explore the relationship between the grain yield-

related traits and nucleotide polymorphisms at the *OsAP2-39* locus,” in line 196-198 in the revised manuscript

R1Q14 (Reviewer #1’s question 14): Line 156-157. Have the authors observed changes in grain number in *OsAP2-39* T vs *OsAP2-39* C?

Our response to R1Q14: We have investigated the relationship between grain number and the variation at *OsAP2-39* locus using a total of 169 cultivated rice. As *OsAP2-39^T* is a rare mutation, only 6 cultivars harbored this mutation and showed an average of grain number by 181.44 ± 20.37 . The rest of 163 cultivars containing the genotype *OsAP2-39^C* showed an average of grain number by 168.34 ± 31.50 . Even so, it was difficult to evaluate the effects of *OsAP2-39^T/OsAP2-39^C* on grain number.

R1Q15 (Reviewer #1’s question 15): Line 183. A reference reporting the *OsMADS1* alternative splicing is missing. In addition, I think it could be good to check *COG1* and *AP2-39* expression in *OsMADS1* natural variants. This important, because it may indicate if *OsMADS1*-mediated regulation of *COG1* is relevant to control grain size.

Our response to R1Q15: Thank you for your constructive suggestion. Reference reported that an alternative splicing (AS) of *OsMADS1* increased grain length and grain yield compared to its native splicing (NS) (Liu et al., 2018; Yu et al., 2018). Therefore, we carried out qRT-PCR analysis for *OsMADS1*, *COG1*, and *OsAP2-39* in NIL-*OsMADS1^{AS}* and

NIL-*OsMADS1*^{NS} plants. We found that the expression levels of *OsMADS1*, *COG1*, and *OsAP2-39* showed no significant difference in NIL-*OsMADS1*^{AS} and NIL-*OsMADS1*^{NS} plants (Supplementary Fig. 16 in the revised manuscript), suggesting the AS/NS variation of *OsMADS1* might not affect the expression of *COG1* and *OsAP2-39*.

We have added the new contents in our revised manuscript in line 239-240 as follows:

RT-qPCR analysis showed no significant difference in *OsMADS1*, *COG1*, and *OsAP2-39* expression between NIL-*OsMADS1*^{AS} and NIL-*OsMADS1*^{NS} plants (Supplementary Fig. 16).

R1Q16 (Reviewer #1's question 16): Lines 188-190. RNAi-*OsMADS1* lines has lower expression of *COG1* and *AP2-39*. Did the author observe changes in grain number or size? As above, this may indicate if *OsMADS1*-mediated regulation of *COG1* is relevant to control grain size and number.

Our response to R1Q16: Thank you for your suggestion.

We investigated the grain size and grain number in the transgenic plants C418^{RNAi-*OsMADS1*}, and found that C418^{RNAi-*OsMADS1*} plants displayed longer grains compared to the control (Supplementary Fig. 15d, j in the revised manuscript), suggesting that down-regulated expression of

OsMADS1 increased grain length and decreased the *COG1* and *OsAP2-39* expression.

Unfortunately, there were no obvious change in grain number between C418^{RNAi-*OsMADS1*} and the control plants (Supplementary Fig. 15b, f in the revised manuscript). We also observed that the grains of C418^{RNAi-*OsMADS1*} displayed unclosed hulls (Supplementary Fig. 15c, d in the revised manuscript), suggesting down-regulated expression of *OsMADS1* aroused adverse effects on floral organ development.

We have added the contents in line 229-234 in the revised manuscript as follows:

To identify the mechanism of *OsMADS1* in controlling grain yield, we generated RNAi transgenic plants of *OsMADS1* in the C418 background (C418^{RNAi-*OsMADS1*}). Analysis showed that the C418^{RNAi-*OsMADS1*} plants generated significantly longer grains but there was no significant difference in grain number compared with the control (Supplementary Fig. 15). As the C418^{RNAi-*OsMADS1*} plants formed abnormal grains with hulls which were not closed, they showed a decrease in TGW and yield compared with C418 (Supplementary Fig. 15).

R1Q17 (Reviewer #1's question 17): Line 210. "COG1 or OsAP2-39 may increase grain yield in rice breeding." I suggest changing to "COG1 or OsAP2-39 downregulation may increase grain yield in rice breeding".

Our response to R1Q17: We have revised this sentence according to your suggestion in line 285 in the revised manuscript. Thank you very much!

R1Q18 (Reviewer #1's question 18): Discussion. Do the authors think that any of the other 10 AP2 genes differentially expressed in RNA-seq could also control grain size and/or number? Out of curiosity, did the authors find any other gene already described in grain size and grain number control, differently expressed in RNA-seq?

Our response to R1Q18: Thanks for your concern.

(1) In rice genome, a total of 163 AP2/EREBP (APETALA2/ethylene responsive element-binding protein) genes were identified (Sharoni et al., 2011). In the results of RNA-seq, 15 AP2 genes were differentially expressed between C418 and 8IL73. 16 AP2 genes were differentially expressed between 8II73^{RNAi-COGI} and 8IL73 (Supplementary Data 1 in the revised manuscript). Between the two comparative groups, 11 AP2 genes were overlapped (Fig. 4a in the revised manuscript). Among the 11 AP2 genes, *OsERF3* (*LOC_Os01g58420*) was proved to improve grain weight and grain yield under drought conditions in previous study (Oh et al., 2009; Ramegowda et al., 2014). Another AP2 gene, *LOC_Os08g36920*, was up-regulated aroused by elevated expression of *OsNAC10*, which also improved grain yield under drought conditions (Jeong et al., 2010).

(2) In the RNA-seq results, 536 genes were differentially expressed (DEGs) between C418 and 8IL73, and 953 DEGs were identified between 8IL73^{RNAi-COG1} and 8IL73 (Supplementary Fig. 12a, Supplementary Data 1 in the revised manuscript). Among these overlapped DEGs, except for the 11 *AP2* family genes, a set of other genes were found for regulating grain number or/and grain weigh. For instance, *OsCCA1* (Wang et al. 2020), *OsABCG18* (Zhao et al. 2019), and *OsCEP6.1* (Sui et al. 2016) regulated grain number and grain weight, *GIFI* regulated grain weight by controlling grain-filling (Wang et al., 2008), *OsGSRI/GW6/OsGASR7* (Wang et al., 2009; Shi et al., 2020; Tang et al. 2021), and *OsMKKK70* (Liu et al., 2021) controlled grain weight.

Reviewer #2's questions and our responses:

R2Q1 (Reviewer #2's question 1): This manuscript shows that the 65-bp fragment in 5'UTR of COG1 is important for rice grain yield and so on.

Our response to R2Q1: We are very appreciated for your positive evaluation for our manuscript.

R2Q2 (Reviewer #2's question 2): Supplementary Figure 6 shows relative expression of COG1 mRNA in several developing panicles stage. The authors should show western blot analyses of COG1 protein by anti COG1 same as Fig2. r. This manuscript shows many breeding good results by decrease and increase of the 65-bp fragment in 5'UTR of COG1.

However, the molecular mechanisms are unclear. There are many incomprehensible results.

Our response to R2Q2: Thank you for your appreciation and valuable advice.

(1) We have carried out western blot assays in various tissues including different stage of developing young panicles from C418 and 8IL73 plants. It showed richer COG1 accumulation levels in 8IL73 than that in C418 plants. We have added this new results in Supplementary Figure 5b in the revised manuscript.

(2) We have identified the *COG1* gene through genetic analysis and map-based cloning, and confirmed the function of *COG1* simultaneously regulating grain number and grain weight. Further analysis revealed that the 65-bp deletion in 5' UTR of the *COG1* gene resulted in lower accumulation levels of COG1 protein by attenuating translation efficiency, which increased rice grain number, grain weight, and grain yield. We also verified a pathway that COG1 positively regulated the expression of *OsAP2-39* by binding to its promoter region controlling grain number, grain weight, and grain yield. Functional identification for *OsAP2-39* also confirmed it regulated rice grain number, grain weight, and grain yield negatively. Our findings will be very useful for both basic research on inflorescence development and breeding. We will further elucidate the

molecular mechanisms for *COG1* regulating rice grain yield in future.

Thank you!

R2Q3 (Reviewer #2's question 3): Fig.2 s shows results 35S promoter_5'UTR(65bp+/-)_LUC system. First of all, the authors should display how calculate relative translation efficiency (TE). TE compared mere two times plus 65bp case with minus one. Fig.2 r3 shows COG1 protein decrease in Mut6 by western blot. Show the translation efficiency also in this case.

Our response to R2Q3: Thank you for your suggestion.

(1) Translation efficiency is the rate of mRNA translated into proteins within cells. In our manuscript, the relative TE was calculated as the ratio of relative LUC/REN activity to relative mRNA levels of *LUC/REN* in rice protoplast ($TE = \text{Relative LUC/REN activity} / \text{relative mRNA levels of } LUC/REN$). We have added this information in line 434-437.

(2) In the original manuscript, we only showed the 5' UTR^{In65-bp} of *COG1* (with an insertion of 65-bp) resulted in higher translation efficiency than that with the 5' UTR^{De65-bp} (with a deletion of 65-bp). To further reveal the function of the 65-bp for regulating translation efficiency, we analyzed the effects of fragments within the 65-bp with different length on translation efficiency using the dual-luciferase reporter system. We found that both the fragments F3 (22-bp in the end of 5' region in the 65-bp) and

F5 (32-bp in the end of 5' region in the 65-bp) resulted in similar translation efficiency with the full length of the 65-bp, while the fragments F1 (21-bp in the end of 3' region in the 65-bp), F2 (the 23rd base to the 44th base from 5' to 3' region in the 65-bp) and F4 (33-bp in the end of 3' region in the 65-bp) led to lower translation efficiency than the full length of 65-bp (Fig. 3u in the revised manuscript). These results not only confirmed the function of the 65-bp for increasing translation efficiency, but also indicated the 22-bp in the end of 5' region was critical for increasing translation efficiency. These new results were described in line 133-138 in the revised manuscript as follows:

Further analysis indicated that the fragments F3 (22-bp of the 5' region of the 65-bp indel), F5 (containing the fragment F3), and the full length of 65-bp sequence promoted TE significantly compared with the other fragments within the 65-bp sequence. These results confirmed that the In65-bp in the 5' UTR of the *COG1* gene promoted TE significantly, and that the 22-bp sequence at the end of the 5' region of the 65-bp was crucial for increasing translation (Fig. 3v).

(3) According to your suggestion, we analyzed the effects of 5' UTR in the *COG1* gene in the Mut-6 and 8IL73 plants on translation efficiency using the dual-luciferase reporter system. It showed that the *COG1* gene with 34-bp deletion in 5' UTR in Mut-6 plants exhibited lower translation

efficiency than the control plants (Supplementary Fig. 10b in the revised manuscript). These results further demonstrated that the deletion of the critical 22-bp in the end of 5' region in 65-bp decreased translation efficiency of the *COG1* gene. We have added this new data in line 144-145 and Supplementary Fig. 10b in the revised manuscript.

R2Q4 (Reviewer #2's question 4): Furthermore, Supplementary Fig. 8a shows target1&2, PAM1& 2, but no explain and no evidence. A 34-bp deletion in the Mut-6 is important for grain yield than 65-bp deletion. The authors should describe this points. What molecule contacts the target 1&2 sites? At least the authors should address in vitro translation using these different deplete of 5' UTR of mRNA and search translation efficiency.

Our response to R2Q4: Thank you for your suggestions.

(1) Analysis of the effects of fragment within the 65-bp with different length on translation efficiency of the *COG1* gene showed that the 22-bp in the end of 5' region in the 65-bp was critical for increasing translation efficiency of the *COG1* gene (Please refer to our response to R2Q3). Therefore, we selected the F3 fragment (the 22-bp in the end of 5' region in the 65-bp) which located on the interval of PAM1 (GGG) and PAM2 (TGG) as the target for genome editing by CRISPR/Cas9 system in 8IL73 plants to further convince the function of the 65-bp for increasing translation efficiency of the *COG1* gene in rice plants (Supplementary Fig.

10a in the revised manuscript). According to your suggestion, we added the correspond explain for the edited targets in the 65-bp in the revised manuscript in line 133-138.

(2) We obtained six types of mutated plants by editing the 22-bp in the end of 5' region in the 65-bp using CRISPR/Cas9 system in 8IL73 plant , and only the Mut-6 plants which had the 34-bp deletion containing the F3 fragment (the critical 22-bp in the end of 5' region in the 65-bp) increased grain number, grain weight, and grain yield compared to 8IL73, accompanying with lower COG1 protein levels (Fig. 3l-n, t, Supplementary Fig. 10 in the revised manuscript), suggesting that the 34-bp was important for increasing translation efficiency and grain yield in the Mut-6 plants. To reveal the molecule contacts the target 1&2 sites, we analyzed the effects of fragments within the 65-bp with different length on translation efficiency using the dual-luciferase reporter system. We found that the 22-bp within the 34-bp fragment was critical for increasing translation efficiency (Fig. 3u in the revised manuscript). It also showed that the 34-bp deletion in 5' UTR of the *COG1* gene in Mut-6 plants decreased translation efficiency than the control plants (Supplementary Fig. 10b in the revised manuscript). Nevertheless, how the 22-bp increasing translation efficiency still remains to be further studied.

We have reconstructed this section in line 133-147 in the revised manuscript as follows:

Further analysis indicated that the fragments F3 (22-bp of the 5' region of the 65-bp indel), F5 (containing the fragment F3), and the full length of 65-bp sequence promoted TE significantly compared with the other fragments within the 65-bp sequence. These results confirmed that the In65-bp in the 5' UTR of the *COG1* gene promoted TE significantly, and that the 22-bp sequence at the end of the 5' region of the 65-bp was crucial for increasing translation (Fig. 3v).

To further assess the function of the 65-bp sequence for increasing TE of the *COG1* gene in rice plants, we selected the F3 fragment which was located at the interval of PAM1 (GGG) and PAM2 (TGG) as the target for genome editing by the CRISPR/Cas9 system in 8IL73 plants (Supplementary Fig. 10a). Among the six types of transgenic plant, only the Mut-6 plants with the 34-bp deletion containing the F3 fragment exhibited significantly increased grain number, grain weight, and grain yield compared to 8IL73, accompanied by decreased TE and accumulation levels of COG1 protein, but no significant differences in mRNA levels (Fig. 3h-n, t, Supplementary Fig. 10b-j). Taken together, it was clear that the De65-bp in the 5' UTR of the *COG1* gene significantly attenuated TE.

R2Q5 (Reviewer #2's question 5): Is RNAi-COG1 in Fig. 4 e and C418^{RNAi-COG1} different or not? In Fig.2 s case, 5'UTR(65bp+/-) mRNAs are same level but in minus 65bp case, protein level decrease to one half. In Supplementary Fig.12a, COG1 mRNA level of C418 RNAi-COG1 is 0.4 times. In this case, detect and show COG1 protein level by western bot.

Our response to R2Q5: (1) RNAi-*COG1* in Fig. 4e and C418^{RNAi-COG1} in the previous manuscript are different RNA interference transgenic lines by targeting the *COG1* gene. To avoid confusing, we have renamed "RNAi-*COG1*" in Fig. 4e in the original manuscript as "C418^{RNAi-COG1-2}" in Fig. 5e in the revised manuscript. Thank you!

(2) We have detected the COG1 protein levels by western blot in C418 and C418^{RNAi-COG1} plant, and it showed that C418^{RNAi-COG1} plants contained lower COG1 protein level than C418. The results were presented in line 289-290 and Fig. 6f in the revised manuscript. Thank you!

R2Q6 (Reviewer #2's question 6): Line 74, what means CPD, what abbreviated name? Usually, CPD means the key gene of brassinosteroid biosynthesis in plant hormone field. Never use CPD. It is confusing.

Our response to R2Q6: We apologize for this confusion, and we renamed "CPD" in the original manuscript as "C418^{CTP}" for complementary transgenic plants in C418 background with *COG1* genomic sequence from

Dongxiang common wild rice in the revised manuscript. Thank you very much!

R2Q7 (Reviewer #2's question 7): Line102, And also, PV65-bp and AV65-bp were confusion. What is V? Use other words.

Our response to R2Q7: We apologize for this confusion. In the original manuscript, V meant variation. PV65-bp was the abbreviation for Present Variation of 65-bp, and AV65-bp was the abbreviation for Absent Variation of 65-bp. According to your suggestion, we renamed the “Deletion of 65-bp” as “De65-bp” and the “Insertion of 65-bp” as “In65-bp” in the revised manuscript. Thank you!

R2Q8 (Reviewer #2's question 8): Line 120, all reader is not rice science field. “3K RG” word needs URL.

Our response to R2Q8: Thank you for your valuable suggestion. We have added the contents “(RFGB v2.0 database, [https://www. rmbreeding.cn/](https://www.rmbreeding.cn/))” in the revised manuscript in line 153-154.

Reviewer #3's questions and our responses:

R3Q1 (Reviewer #3's question 1): The topic of this manuscript is general interest for researchers follow trends and development in understanding co-domestication and convergent selection of the yield-related traits in cereal crops,

Our response to R3Q1: We are very appreciated for your positive evaluation for our manuscript.

R3Q2 (Reviewer #3's question 2): the authors found that all of wild rice harbored the PV-65bp haplotype, its frequency in indica rice was 90% and that in japonica rice was close to 55%, indicating that the favorable haplotype AV-65bp was only occurred in part of the rice cultivars. The title of this manuscript should be modified to more accurate description of the experimental results.

Our response to R3Q2: Thank you for your suggestion. We have modified the title to “The genetic module *OsMADS1-COG1-OsAP2-39* simultaneously increases grain number and grain weight in rice” according to your suggestion.

R3Q3 (Reviewer #3's question 3): to understand the functional roles of the COG1/*OsMADS17* and *OsAP2-39* in the regulation of plant development and growth in rice, such as plant architecture, panicle branching and grain size, and the authors should generate the null mutants by using CRISPR/Cas9 method or mutations.

Our response to R3Q3: Thank you for your suggestion. Actually, we had generated the null mutants by CRISPR/Cas9 system in ZH17 background. Phenotype investigation for the null mutants showed that the single mutant *cog1* or *osap2-39* increased grain weight and grain yield significantly, and

the double mutant *cog1/osap2-39* could further increase grain weight and grain yield compared to the single mutant *cog1* or *osap2-39*, while no obvious change in panicle number, panicle length, and branch number among the mutants and the controls (Supplementary Fig. 18 in the revised manuscript). These results further convinced that *COG1* and *OsAP2-39* promoted grain yield significantly with no obvious adverse effects on plants development and growth. We have added the new contents in line 254-262 in the revised manuscript as follows:

To further elucidate the genetic interaction between *COG1* and *OsAP2-39*, we generated the null single mutants *cog1* and *osap2-39*, and the double mutant *cog1/osap2-39* in the ZH17 background using the CRISPR/Cas9 system (Supplementary Fig. 18a). Phenotypic analysis showed that the single mutants *cog1* or *osap2-39* exhibited significantly increased individual grain weight and grain yield compared with the control (Supplementary Fig. 18b-o), while the double mutant *cog1/osap2-39* further increased grain weight and grain yield compared with the single mutants *cog1* or *osap2-39* (Supplementary Fig. 18b-o), indicating that *COG1* and *OsAP2-39* have additive effects on grain yield regulation.

R3Q4 (Reviewer #3's question 4): as mentioned above, grain yield is controlled by the three major components: the number of tillers per plant, the number of grains per panicle and 1,000-grain weight, and that are

typically negatively correlated with one another. In the current version of the manuscript, the authors only described the effects of the OsMADS1-COG1-OsAP2-39 module on the grain number and grain size, how about tiller number? The authors should add these additional information about the effects of the COG1, OsMADS1 and OsAP2-39 genes on tiller number.

Our response to R3Q4: Thanks for your suggestion. We had investigated the panicle number (tiller number) per plant for evaluating the effects generated by *COG1*, *OsMADS1* and *OsAP2-39*. Statistical analysis for panicle number in C418 and 8IL73 plants showed no significant difference (Supplementary Fig. 2d in the revised manuscript). All transgenic plants of *COG1*, *OsMADS1*, and *OsAP2-39* also showed no obvious change in panicle number compared to the controls. We have added the information about the effects of the *COG1*, *OsMADS1* and *OsAP2-39* genes on tiller number in the revised manuscript.

R3Q5 (Reviewer #3's question 5): the authors reported that a 65-bp deletion in the 5' untranslated region (5' UTR) was associated with the increased grain number and grain weight through decreasing mRNA translation efficiency. In Fig. 2r, the authors showed the accumulation of the COG1 protein in young panicles by analyzing the three different experimental comparisons. To compare the levels of the COG1 protein in isogenic lines and/or transgenic plants, the authors perform western blot

experiment with the same nitrocellulose or PVDF membrane. In addition, they should also quantify the transcript levels of the *COG1* gene using the same samples in the main figures.

Our response to R3Q5: Thanks for your comment and suggestion. We have analyzed the expression levels of *COG1* in C418 and different transgenic plants. It showed that the complement transgenic plants C418^{CTP} (complement transgenic plants harboring the *COG1* gene from 8IL73 in C418 background), C418^{CTP-CW} (the 2.5-kb *COG1* promoter from C418 fused with the *COG1* transcript region from 8IL73 in C418 background), and C418^{CTP-WC} (the 2.5-kb *COG1* promoter from 8IL73 fused with the *COG1* transcript region from C418 in C418 background) displayed no significant change of *COG1* expression levels compared to the C418 plants (Fig. 2e, Fig. 3d in the revised manuscript). In the 8IL73^{RNAi-COG1} and C418^{RNAi-COG1} plants, it exhibited significantly declined expression levels of *COG1* compared to 8IL73 and C418, respectively (Fig. 2l, Fig. 6e in the revised manuscript). We have added the new results in the revised manuscript.

R3Q6 (Reviewer #3's question 6): in Fig. 2s, the authors should represent the data of the accumulation of LUC protein.

Our response to R3Q6: Thanks for your suggestion. We have added the data of the accumulation of LUC protein according to your suggestion. The

dual-luciferase reporter system with the 5' UTR^{In65-bp} of *COG1* (an insertion of 65-bp) showed higher translation efficiency and richer accumulation of LUC protein than that with the 5' UTR^{De65-bp} (a deletion of 65-bp) in rice protoplast. We also found that the dual-luciferase reporter system with the full length of the 65-bp, the fragment F3 (22-bp in the end of 5' region in the 65-bp) and F5 (32-bp in the end of 3' region in the 65-bp) from the 65-bp enhanced translation efficiency and accumulated more LUC protein than that with fragment F1 (21-bp in the end of 3' region in the 65-bp), F2 (the 23rd base to the 44th base from 5' to 3' region in the 65-bp), and F4 (33-bp in the end of 3' region in the 65-bp). The new results were displayed in Fig. 2u in the revised manuscript.

R3Q7 (Reviewer #3's question 7): in Fig.S6, the authors showed the comparisons of transcriptional levels between C418 and its near-isogenic line 8IL73, it will be better to show the comparison of the accumulation of COG1 protein in the various tissues.

Our response to R3Q7: Thanks for your suggestion. We have added the comparison of the accumulation of COG1 protein in the root, tiller base, leaf sheath, leaf blade, node, inter node, pulvinus, and developing young panicles of C418 and 8IL73 plants. It consistently demonstrated that 8IL73 plants contained richer COG1 protein levels than C418 plants. The new data was presented in Supplementary Fig. 5b in the revised manuscript.

Reviewer #4's questions and our responses:

R4Q1 (Reviewer #4's question 1): Although the authors provided a large number of data supporting the genetic module of OsMADS1-COG1-OsAP2-39, there are some issues that need to be clarified.....

Our response to R4Q1: We are very appreciated for your positive evaluation for our manuscript.

R4Q2 (Reviewer #4's question 2): QTL analysis

The authors detected a QTL for grain number and weight on chr. 4. However, I do not know the traits are basically controlled by a single locus between the C418 and 8IL73.

Provide results of QTL analysis of markers covering other wild segments introgressed in C418 background. Also provide data about the contribution of the COG1 on phenotypic variance. This is important to clarify the difference of grain size and number between C418 and 8IL73 controlled by COG1 based on single Mendelian inheritance.

Our response to R4Q2: We had carried out QTL analysis using segregation population derived from the cross between C418 and the introgression line 8IL73 and polymorphic markers covering all wild rice chromosome segments introgressed in C418 background. One major QTL (*CONTROL OF GRAIN YIELD 1, COG1*) for grain number and grain weight located between the markers LT47 and LT43 on the long arm of

chromosome 4 was detected. The QTL explained 23% and 14% of phenotypic variance of grain number per main panicle and grain weight, respectively, with additive effects of -29.89 for grain number and -0.9 g for grain weight, indicating that the allele from wild rice decreased grain number and grain weight (Supplementary Table 1 in the revised manuscript). Based on only one major QTL *COG1* was detected in the segregation population derived from cross between C418 and the introgression line 8IL73, we speculated that the difference of grain size and number between C418 and 8IL73 mainly controlled by *COG1*, a single Mendelian genetic factor. We have added this data in Supplementary Table 1 in the revised manuscript. Thank you very much!

R4Q3 (Reviewer #4's question 3): Complementation test

In the first section of the text, authors showed detection of the QTL controlling grain size and number based on the segregating population between C418 and 8IL73, and further fine mapping analysis identified a candidate gene of *OsMADS17*. Then, complementation and RNAi experiments were carried out to confirm the causal gene. However, I am puzzled which allele is functional at this stage of gene identification. Authors carried out complementation test by introducing wild allele to C418 without any evidence. I felt this approach is a lack of logical flow. I suggest reconstructing the process for identification and confirmation of the causal mutation. Complementation test needs to be carried out after

clarifying the function of two alleles.

Our response to R4Q3: Thank you for your constructive suggestions. We have reconstructed these paragraphs in line 77-100 in the revised manuscript as follows:

Sequencing the transcript region of *LOC_Os04g49150* revealed that there were ten single-nucleotide polymorphisms (SNPs) and six insertions/deletions (indels) between 8IL73 and C418 (Supplementary Fig. 4). Among these variations, a synonymous mutation (SNP 1) and a 65-bp insertion/deletion (Indel 1) existed in the coding region and the 5' UTR, respectively (Supplementary Fig. 4). Reverse transcription quantitative PCR (RT-qPCR) analysis showed that the transcript levels of *LOC_Os04g49150* were not significantly different between 8IL73 and C418 in developing young panicles (Supplementary Fig. 5a).

As sequence comparisons and transcript level analysis did not provide enough evidence to estimate the functional allele of *LOC_Os04g49150*, we generated two reciprocal transgenic plants, C418^{CTP} and 8IL73^{CTP}. The C418^{CTP} transgenic plants, which were generated by introducing the *LOC_Os04g49150* gene from 8IL73 into C418 plants, showed a similar phenotype and *COG1* expression level as with 8IL73, but fewer grains per panicle, a lower grain weight and grain yield than that of the C418 control plants (Fig. 2b-h, Supplementary Fig. 6). On the other hand, the 8IL73^{CTP} transgenic plants, which were generated by introducing the

LOC_Os04g49150 gene from C418 into 8IL73 plants, displayed no significant difference in grain yield-related traits compared with 8IL73 (Supplementary Fig. 7). To verify the function of *LOC_Os04g49150*, we generated RNA interference (RNAi) transgenic plants (8IL73^{RNAi-COG1}) by interfering with expression of the *LOC_Os04g49150* gene in 8IL73 plants, and found that the 8IL73^{RNAi-COG1} plants exhibited a significant decrease in *COG1* expression levels and a significant increase in grain number, grain size, and grain yield compared with the controls (Fig. 2i-o, Supplementary Fig. 8). Taken together, these results demonstrated that *LOC_Os04g49150* was responsible for the *COG1* gene, which functions as a simultaneous co-regulator for grain number and individual grain weight in rice.

R4Q4 (Reviewer #4's question 4): Transgenic experiments

There are only a single transgenic lines presented in each experiment. As authors suggested that expression or translational control of the related genes are important for the grain size and number, the dosage in copy number of the transgenes is important matter in transgenic experiments. Especially, transgenic experiments were carried out using pCAMBIA, which can be a multi-copy vector, as long as I know. If the results are supported by multiple transgenic lines, I agree the authors conclusions, however the single transgenic line is insufficient generally. I also have a concern for RNAi experiments about their target specificity such as MADS

genes. The expression levels missing in the data can be added (i.e. RNAi plants).

Our response to R4Q4: Thank you for your valuable advice.

(1) For the transgenic plants of *COG1* and *OsAP2-39*, we used 3 independent transgenic lines for analyzing the relative expression levels and phenotype. We have showed all of them in the revised manuscript in Fig. 2e-h, 2l-o, Fig. 3d-g, 3k-n, Fig. 4i, 4m, Fig. 6e, 6g-i, Supplementary Fig.6, Supplementary Fig. 7, Supplementary Fig. 8, Supplementary Fig. 9, Supplementary Fig. 13, and Supplementary Fig. 19.

(2) The specificity of targeted gene sequence for RNAi experiments is important, especially for those with multiple homologous genes, such as MADS family and *AP2* family genes. Therefore, we carefully considered the specificity of targeted gene sequence when we generated the RNAi constructions for *COG1* and *OsAP2-39*. qRT-PCR analysis for 13 MADS family genes and 13 *AP2* family genes showed none of them exhibited significant alteration of expression levels in RNAi transgenic plants compared to the controls except for the targeted genes *COG1* and *OsAP2-39* (Response Fig. 1), indicating that the RNAi vector for *COG1* or *OsAP2-39* was specific to the targeted gene. We have added the contents in the part of methods in the revised manuscript in line 385-386.

Response Fig. 1 Expression levels of *MADS* (a) and *AP2* (b) family genes.

(3) Thank you for your reminding. We have added the missed data for expression levels in the revised manuscript. The complement transgenic plants C418^{CTP} (harboring the *COG1* gene from 8IL73 in C418 background), C418^{CTP-CW} (having the 2.5-kb *COG1* promoter from C418 fused with the *COG1* transcript region from 8IL73 in C418 background), and C418^{CTP-WC} (possessing the 2.5-kb *COG1* promoter from 8IL73 fused with the *COG1* transcript region from C418 in C418 background) displayed no significant change of *COG1* expression levels compared to

the C418 plants (Fig. 2e, Fig. 3d in the revised manuscript). The 8IL73^{RNAi-COG1} and C418^{RNAi-COG1} plants, exhibited significantly declined expression levels of *COG1* compared to 8IL73 and C418, respectively (Fig. 2l, Fig. 6e in the revised manuscript).

R4Q5 (Reviewer #4's question 5): Wild rice accessions

In supplementary Table 1 of 68 wild rice accessions, there are non-*O. rufipogon* species. Those are *O. officinalis*, *O. longistaminata*, *O. grandiglumis*. I do not understand why different wild rice species were included. Moreover, more than half of wild rice accessions used in this study are from China, which seems to be biased, therefore strongly recommended to analyse wild rice accessions broadly.

Our response to R4Q5: Thank you for your concern and suggestions.

(1) In original manuscript, we analyzed the sequence variation of *OsMADS1*, *COG1*, and *OsAP2-39* among 58 accessions of *O. rufipogon*, 7 accessions of *O. nivara*, 1 accession of *O. longistaminata*, 1 accession of *O. grandiglumis*, and 1 accession of *O. officinalis*. *O. rufipogon* and *O. nivara* belonged to AA genome and the relative species of cultivated rice. All *O. rufipogon* and *O. nivara* analyzed contained the genotype *OsMADS1*^{NS}, *COG1*^{In65-bp}, and *OsAP2-39*^C. We attempted to explore the variation of *OsMADS1*, *COG1*, and *OsAP2-39* in the non-AA genome species of wild rice, then we analyzed the genotype of *OsMADS1*, *COG1*,

and *OsAP2-39* among *O. longistaminata* (CC), *O. grandiglumis* (CCDD), and *O. officinalis* (CC). Interestingly, the wild rice with AA genome and non-AA genome harbored same genotypes *OsMADSI*^{NS}, *COG1*^{In65-bp}, and *OsAP2-39*^C.

(2) According to your suggestion, we analyzed additional 24 accessions of *O. rufipogon* from 8 different countries in South Asia and Southeast Asia, and found that all of them harbored the genotype *OsMADSI*^{NS}, *COG1*^{In65-bp}, and *OsAP2-39*^C as same as those from China. Detailed information for the 24 accessions of *O. rufipogon* was listed in Supplementary Table 2 in the revised manuscript.

R4Q6 (Reviewer #4's question 6): Selection of the loci in rice domestication.

L. 125 AV-65bp was originated from the mutation in cultivated rice, and then artificial selection were occurred in part of them.

Since the COG1AV-65bp allele was found in 10.1% and 44.2% of indica and japonica cultivars in the 3K rice genome, respectively. This results let me think the mutation was selected relatively recent, thus I do not agree that the locus was under selection in rice domestication. Without evidences (i.e. selective sweep at the locus), removal of “domestication” in the title and text is recommended. I request the authors rephrase this section of the manuscript to tone down their conclusions. COG1 can be improvement

gene.

Our response to R4Q6: Thank you for your valuable suggestions.

Generally, the process of crop evolution from wild species to cultivated species including two stages of domestication and diversification. In the early stage, it occurred the direct selection from wild ancestral species by the human resulting in remarkable changes of plant morphology, which was termed as “domestication”. After “domestication”, the “diversification” happened to further improve the yield, adaptation or quality of the domesticated crop species (Meyer et al., 2013).

There are two kinds of definition for domestication, both definitions are accepted by scientists in the research area. (i) Broad definition for domestication: the process of domestication included the above-mentioned two stages (“domestication” and “diversification”) (Li et al., 2017). (ii) Narrow definition for domestication: the process of domestication only included the above-mentioned first stage (“domestication”) excluding the second stage (“diversification”) (Meyer et al., 2013).

In our manuscript, we used broad definition for domestication. Therefore, we stated that *COG1* is a domestication-related gene. Previous studies identified several domestication-related genes, including *Sh4/SHAI* (Konishi et al., 2006; Li et al., 2006; Lin et al., 2007), *PROG1* (Jin et al., 2008; Tan et al., 2008), *Bh4* (Zhu et al., 2011), *Rc* (Sweeney et al., 2006),

OsLGI (Ishii et al., 2013; Zhu et al., 2013), *An-1* (Luo et al., 2013), *LABA1/An-2* (Gu et al., 2015; Hua et al., 2015), *RAE2/GAD1* (Bessho-Uehara et al., 2016; Jin et al., 2016;) and *FZP* (Bai et al., 2017; Huang et al., 2018). All these genes are considered to be domestication-related genes based on broad definition for domestication, while some of them (such as *Rc*, *Bh4*, *LABA1*) should be diversification-related genes according to narrow definition for domestication.

We are willing to use narrow definition for domestication to define the *COG1* gene. We have revised the title to “The genetic module *OsMADS1-COG1-OsAP2-39* synchronously increases grain number and grain weight in rice”. We also toned down our conclusions and used the word “improvement” for substituting the word “domestication” in the context in the revised manuscript according to your suggestion.

R4Q7 (Reviewer #4’s question 7): Identification of downstream and upstream genes.

Authors identified *OsAP2-39* and *OsMADS1*, as downstream and upstream factors. However, selections of these two genes are biased. Involvement of other factors are left behind. Involvement of other family genes can be addressed in the text.

Our response to R4Q7: Thanks for your constructive suggestion. In the RNA-seq results, we identified 355 differentially expressed genes (DEGs)

among C418, 8IL73, and 8IL73^{RNAi-CO₁} plants. Among the 355 DGEs, a number of genes, such as *OsAP2-39* (Yaish et al. 2010), *OsERF3* (Oh et al., 2009; Ramegowda et al., 2014), *OsCCAI* (Wang et al. 2020), *OsABCG18* (Zhao et al. 2019), *OsCEP6.1* (Sui et al. 2016), *GIF1* (Wang et al., 2008), *OsGSR1/GW6/OsGASR7* (Wang et al., 2009; Shi et al., 2020; Tang et al. 2021), and *OsMKKK70* (Liu et al., 2021), regulated grain number or/and grain yield regulation of rice. We are uncovering the relationships between *COG1* and these genes.

To identify the upstream genes of *COG1*, we analyzed the promoter region of *COG1* finding two CARG motifs that probably interacts with MADS family transcription factors. We validated that one of the binding sites could directly interact with OsMADS1, which was proved to control grain weight in rice. We think there are other family genes in the upstream and downstream of *COG1*, and we will further explore them in the next study. We have addressed these probably involved genes in the part of discussion in the revised manuscript in line 321-327.

R4Q8 (Reviewer #4's question 8): Use of “respectively” is grammatically wrong in many part of text. These must be corrected with English edits.

Our response to R4Q8: We are sorry for these errors. We have made corrections precisely in the whole text.

R4Q9 (Reviewer #4's question 9): Consider about use of “haplotype” in the text.

A haplotype is a group of alleles that are inherited together. Authors explain about a single locus in the work. I think “genotype” or “allele” may be appropriate.

Our response to R4Q9: We have substituted the word “haplotype” by “genotype” according to your suggestion. Thank you!

R4Q10 (Reviewer #4's question 10): L. 23. I do not understand what “Development process of human social” means.

Our response to R4Q10: We apologize for this confusion. We have corrected this sentence to “Crop domestication and improvement are outstanding events in human history” in line 28 in the revised manuscript.

R4Q11 (Reviewer #4's question 11): Fig. 2a. Provide genotype information for black and white bars as well as result of statistical analysis for “a” and “b”. Also, the coding region of COG1 (OsMADS17) is about 5.9-Kb region according to the Supplementary Figure 7. The display of the gene in Fig. 2 looks quite large in a mapped region of 19.6kb. I recommend to adjust it.

Our response to R4Q11: Thank you for your suggestions.

(1) We have added the genotype information and the results of statistical analysis in Fig. 2a.

(2) In Fig. 2a in the original manuscript, the area marked with double arrows represented the interval of 19.6-kb, and the schematic for the *COGI* gene didn't reflect the real length of the gene sequence. We have adjusted this region and added scale bars in Fig. 2a and Supplementary Fig. 4 in the revised manuscript.

R4Q12 (Reviewer #4's question 12): Fig. 2r. Gel blot panels for Mut-6 and pCPL1/2 can be swapped to follow the text.

Our response to R4Q12: We have swapped the figures according to your suggestion. Thank you very much!

R4Q13 (Reviewer #4's question 13): Fig. 2t. Some cultivars are removed from rice 3K (i.e. $n=1760+819=2579$)

Our response to R4Q13: In this study, we intend to understand the distribution of the genotype of *OsMADSI*^{NS/AS}, *COGI*^{In65-bp/De65-bp}, and *OsAP2-39*^{T/C} in *indica* and *japonica* rice. Therefore, we only selected 2,643 cultivars which were clearly classified into *indica* and *japonica* rice in the 3K RG (RFGB v2.0 database, <https://www.rmbreeding.cn/index>) for analysis. Among them, a total of 2,579 cultivars including 1,760 *indica* and 819 *japonica* had the genotype information for *OsMADSI*^{NS/AS}, *COGI*^{In65-bp/De65-bp}, and *OsAP2-39*^{T/C}. Therefore, we only used the 2,579 cultivars for analysis. Thank you very much!

R4Q14 (Reviewer #4's question 14): Fig. 4k. I could not find yellow circles in the map.

Our response to R4Q14: We apologize for this confusion. In the original manuscript, the yellow circles in Fig. 4k represented for *indica* rice with the genotype *OsMADSI*^{AS}. However, the genotype *OsMADSI*^{AS} appeared merely 0.17% in *indica* rice (Fig. 5h in the revised manuscript), and only 3 cultivars were found with genotype *OsMADSI*^{AS} in 1,760 *indica* rice, then the yellow circle representing *OsMADSI*^{AS} in *indica* was difficult to be found in the map. When the map is enlarged, the yellow circle is appeared in the left. To avoid confusion, we didn't show the yellow circle and removed the icon of the yellow circle from the figure legend in Fig. 5j in the revised manuscript. The revision will not affect the conclusion that the genotype *OsMADSI*^{AS} was originated from *japonica* rice in our study.

R4Q15 (Reviewer #4's question 15): Supplementary Table 1. Clarify about 'NS' in the table. Need a space between *Oryza* (O.) and *rufipogon*.

Our response to R4Q15: Thanks for your suggestion.

(1) 'NS' represented the type of the native splicing of *OsMADSI* which was accordant with that in *japonica* rice variety Nipponbare. We have added this information in the table legend of Supplementary Table 2 in line 972-977 in the revised manuscript.

(2) We have made these corrections according to your suggestion.

R4Q16 (Reviewer #4's question 16): Supplementary Figure 5. Expression of *COG1* in two transgenic lines are very different among lines. This can be due to copy numbers of the transgene or other factors. I do not know this is technical or biological replications. Please clarify these concerns.

Our response to R4Q16: In the original manuscript, we used three different lines as biological replications for each of pCPL1 and pCPL2 transgenic plants to detect expression levels of the *COG1* gene in Supplementary Figure 5. The differences of *COG1* expression levels among lines were probably aroused by different copy numbers of the transgene. In the revised manuscript (We renamed the complement transgenic plants pCPL1 and pCPL2 as C418^{CTP-WC} and C418^{CTP-CW}, respectively.), we detected the expression levels of *COG1* separately in three independent lines for each of C418^{CTP-WC} and C418^{CTP-CW} transgenic plants in Figure 3d. It also showed no significant difference of *COG1* expression levels among C418 and the transgenic plants. Thank you!

R4Q17 (Reviewer #4's question 17): Supplementary Figure 6. L.668. I afraid I do not understand what “pulvin” means. Is this a “node”? For in situ hybridization, provide plant information (C418 or 8IL73) for sense probe as negative control (b).

Our response to R4Q17: We apologize for the typing errors in the original manuscript. It should be “pulvinus”, and we have made the correction.

We used the young panicle of 8IL73 for sense probe as negative control, and we have added this information in line 795-796 in the revised manuscript.

R4Q18 (Reviewer #4's question 18): Supplementary Figure 11. Authors studied grain size and *OsAP2-39* expression level using 9 cultivars. The reason for the selection of these cultivars as well as their genotypic information for *COG1* needs to be provided. To meet up with authors idea, *COG1* genotype should be identical to discuss about the effect of *OsAP2-39*. In addition, 4542A3-49B-2B12 is a name of cultivar?

Our response to R4Q18: Thanks for your suggestions.

(1) We only found six cultivars with the *OsAP2-39^T* genotype from 169 cultivars. Among them, 4 cultivars with the *COG1^{In65-bp}* genotype and 2 cultivars with the *COG1^{De65-bp}* genotype at the *COG1* locus. To compare the grain size and expression levels of *OsAP2-39* of the genotype *OsAP2-39^T* and *OsAP2-39^C*, we selected 7 cultivars with the *OsAP2-39^C* genotype, in which 4 cultivars harbored the *COG1^{In65-bp}* genotype and 3 cultivars had the *COG1^{De65-bp}* genotype at the *COG1* locus for analysis. Therefore, we analyzed 13 cultivars to study grain size and *OsAP2-39* expression levels. In Supplementary Figure 11 in the original manuscript, we showed the results of 9 cultivars. Among the rest of 4 cultivars, three cultivars with the *OsAP2-39^T* genotype also had bigger grain size and lower expression levels

of *OsAP2-39*, one cultivar with the *OsAP2-39^C* genotype showed smaller grain size and higher expression levels of *OsAP2-39*. Further comparison of grain length and genotype at the *COG1* and *OsAP2-39* loci showed that the cultivars with the *OsAP2-39^T/COG1^{De65-bp}* genotype had longer grains than those with the *OsAP2-39^T/COG1^{In65-bp}* genotype. We have showed the results and detailed information for the 13 cultivars in Supplementary Figure 14 and Supplementary Data 5 in the revised manuscript. Thank you!

(2) 4542A3-49B-2B12 is a name of material kindly provided by Professor Susan R. McCouch of Cornell University.

R4Q19 (Reviewer #4's question 19): Use same style for plant materials. For example, C418RNAi-COG1 and RNAi-OsAP2-39-1.

Our response to R4Q19: According to your suggestion, the transgenic plants were renamed as the same style, such as C418^{CTP} (complement transgenic plants harboring the *COG1* gene from 8IL73 in C418 background), 8IL73^{CTP} (complement transgenic plants containing the *COG1* gene from C418 in 8IL73 background), C418^{CTP-WC} (complement transgenic plants having the 2.5-kb *COG1* promoter from 8IL73 fused with the *COG1* transcript region from C418 in C418 back ground), C418^{CTP-CW} (complement transgenic plants having the 2.5-kb *COG1* promoter from C418 fused with the *COG1* transcript region from 8IL73 in C418 background), C418^{RNAi-COG1} (RNA interference for *COG1* in C418 plants),

8IL73^{RNAi-COG1} (RNA interference for *COG1* in 8IL73 plants), C418^{RNAi-OsMADS1} (RNA interference for *OsMADS1* in C418 plants), C418^{OE-OsAP2-39} (overexpression of *OsAP2-39* in C418 plants), and C418^{RNAi-OsAP2-39} (RNA interference for *OsAP2-39* in C418 plants), in the revised manuscript. Thank you!

R4Q20 (Reviewer #4's question 20): In discussion section, how the *COG1* overcome the trade-off effect between grain and weight can be discussed. This can be an important point for the discovery of *COG1*.

Our response to R4Q20: Thanks for your suggestion.

We have added the contents for discussing the mechanism of *COG1* overcoming the trade-off effect between grain number and weight in line 313-329 in the revised manuscript as follows:

The trade-off effect between grain number and grain weight, in which, for example, increased grain number per panicle is associated with smaller grains, limits any increases in grain yield. To overcome this trade-off effect has been a tough task for rice improvement. Here, we found that a 65-bp deletion in the 5' UTR of the *COG1* gene, which decreased translation efficiency of *COG1* or downregulated expression of *COG1*, could increase grain number and individual grain weight simultaneously. As a transcription factor, *COG1* probably participated in the regulation of expression of multiple genes. For instance, the downstream gene *OsAP2-*

39, which was positively and directly regulated by COG1 also simultaneously controlled grain number and individual grain weight. Furthermore, the RNA-Seq results revealed that the genes *OsCCA147*, *OsABCG18*⁴⁸, *OsCEP6.1*⁴⁹, *GIF1*⁵⁰, *OsGSRI/GW6/OsGASR7*^{51–53}, and *OsMKKK70*⁵⁴, which regulated grain number and/or grain weight, were differentially expressed between C418 and 8IL73 (Supplementary Data 2). It is worth mentioning that *OsABCG18* regulates grain number by affecting shootward transport of root-derived cytokinins⁴⁸, and *OsGSRI/GW6/OsGASR7* regulates grain size by controlling brassinosteroid (BR) and gibberellin (GA) signaling^{51–53}. Therefore, we speculated that COG1 was probably involved in regulating the expression levels of these genes to overcome the trade-off effect between grain number and grain weight. However, the detailed molecular mechanisms still remain to be elucidated.

R4Q21 (Reviewer #4's question 21): L. 172. What “mutated cultivate rice” means?

Our response to R4Q21: We apologize for this confuse. We have corrected this sentence to “Taken together, *OsAP2-39*^T is a rare mutation in cultivated rice, and the varieties with the combination of *OsAP2-39*^T and *COG1*^{De-65bp} were selected for only in *indica* rice.” in line 216-218 in the revised manuscript.

Reviewers' Comments:

Reviewer #1:

Remarks to the Author:

The authors have addressed most of the comments and suggestions I raised in my initial review. I think the new data and corrections have improved the manuscript, in particular the clarifications regarding the number of transgenics events analyzed, quantification of COG1 protein levels and specificity of RNAi vectors.

Still, I have some concerns regarding the proposed genetic module OsMADS1-COG1-OsAP2-39:

I agree with the authors' conclusion that their evidence support that COG1 is a positive regulator of AP2-39, and that that regulation is likely involved in grain size and number regulation. However, I think that the genetic evidence does not support conclusions as stated in:

Title: The genetic module OsMADS1-COG1-OsAP2-39 simultaneously increases grain number and grain weight in rice.

Lines 21-22: "We found that the genetic module OsMADS1-COG1-OsAP2-39 increases both grain number and individual grain weight simultaneously".

Lines 253-254 "These results suggested that genes OsMADS1, COG1, and OsAP2-39 functioned together to regulate individual grain weight.

Lines 268-270: "these findings showed that OsMADS1, COG1, and OsAP2-39 co-regulated grain yield by constituting a genetic module sequentially from the upstream to the downstream (Fig. 5k)".

According to the new genetic results, cog1 osap2-39 double mutant showed additive effects relative the single mutants in the regulation of grain size and yield. This observation is not consistent with a single genetic module COG1-OsAP2-39, but more consistent with a model where COG1 (more likely) or AP2-39 regulate grain traits by controlling additional genes.

I think that all these 3 genes are part of large gene network controlling inflorescence, flower, and grain development. COG1 likely control multiple genes as indicated by the authors in line 176 "It showed that COG1 participated in multiple regulatory pathways" or in line 318 "As a transcription factor, COG1 probably participated in the regulation of expression of multiple genes". I think the authors did a good work validating the interaction of COG1 with one of those potential targets, AP2-39, but I think it is not correct to conclude that they have identify a genetic module controlling grain development. Instead, I think they have initiated the characterization of a network controlled by COG1 and I suggest the authors to modify the title, abstract and the manuscript to reflect that idea.

The relation with MADS1 is even more complicated since OsMADS1 controls the expression of multiple genes during inflorescence development and osmads1 plants have several pleiotropic effects. In addition, the authors did not find evidence that the natural variation in OsMADS1, associated with changes in grain size, affects COG1 or AP2-39 expression (supplementary fig 16). Also, I feel that the selection of MADS1 based on the presence of a CArG box in COG1 promoter is biased. In conclusion, I think that the current results related to MADS1 are not well supported, and are not relevant to the main story of the manuscript.

I have other minor suggestions:

Abstract.

I suggest including more details about COG1, for example that the 65-bp deletion in the 5' UTR in COG1 is presented in cultivated rice and is associated with enhanced grain weight. Also, that COG1 regulates multiple genes and that the interaction with one of them, AP2-39, has been characterized.

Introduction.

Lines 28-42. I found the two first paragraph of the introduction (lines 28-35 and lines 36 to 42) very redundant. I think they could be combined in a single improved paragraph.

Results.

Line 89. "...phenotype and COG1 expression level as with 8IL73". I think this should be corrected to "...phenotype and COG1 expression level as non-transformed C418".

Lines 93-97. The authors showed in their response letter that the RNAi vector against COG1 is specific, but they have not included that figure in the current manuscript. I think it would be good to include it when presenting the results of 8IL73RNAi-COG1. It could be mentioned as "To verify the function of LOC_Os04g49150, we generated a COG1 specific RNA interference (RNAi) vector (Supplementary FigX). Transgenic 8IL73 plants expressing the RNAi vector (8IL73RNAi-COG1) exhibited a significant decrease in COG1 expression levels..."

Line 119. I suggest changing "the expression" to "the transcript expression"

Line 121. The authors should mention that C418 contains lower protein levels in more tissues than young panicles. The plants having higher expression of COG1 look shorter (Figures 6a; Supplementary Figures 2a, 6a, 8a, 9b, 10c), so I am wondering whether the difference in COG1 protein levels in nodes (Supplementary Figure 5b) could be related to this phenotype.

Line 135. I suggest changing "F5 (containing the fragment F3)," to "F5 (32-bp in the end of 3' region in the 65-bp, including the deletion in the fragment F3)"

Lines 146-147. I think a better conclusion to this paragraph would be that "the De65-bp in the 5'UTR of the COG1 gene significantly attenuate TE and result in increased grain number, grain weight, and grain yield". Out of curiosity, did the author check for any conserve element in the 22-bp fragment?

Lines 165-167. I feel that the conclusion sentence fit better after paragraph between lines 148-157. Maybe a proper conclusion for the paragraph between lines 158-165 would be that 65-bp deletion is associated with larger grains in cultivated rice.

Line 192. The authors showed in the response letter that the RNAi vector against AP2-39 is specific, but they have not included the figure in the current manuscript. I suggest including that.

Line 203. Please indicate which tissue was used in the RT-qPCR.

Lines 217-218. Since only a few accessions contain both OsAP2-39T and COG1De65-bp alleles, I have doubts about the conclusion that the combination of both alleles was selected. How can the author exclude the possibility that observation is the result of random crossing?

Lines 252 and 253. COG1AS. Did the authors mean OsMADs1AS?

Lines 285-288. I think the authors could explain their results in more details. The authors have showed that only 24% of cultivated rice have the beneficial De65-bpCOG1 allele, so reducing COG1 expression in the 76% having the In65-bp allele would be very interesting. In addition, the authors showed that reducing COG1 expression in cultivars that have the COG1 beneficial allele, like C418, also result in more and larger grains. So further reduction of COG1 in cultivated rice with the beneficial allele still could be beneficial. I think the authors should improve the paragraph to clearly explain all these possibilities.

Methods

Line 416. Please indicate that in Fig 3 and 6 western blots were done with 3 replications. Also indicate the method use to quantify the western blot bands.

Reviewer #3:

Remarks to the Author:

In this revised version of the manuscript, the authors have added additional experiments and answered the questions which the reviewer #2 and I addressed. Most of my comments are relatively

minor, I think the authors should give attention to the following suggestions before it is accepted for publication:

1, both reviewer #1 and I pointed out the same suggestion, I still think that the authors should modify the title, and directly replace COG1 with OsMADS17.

2, in the new Supplementary Figure 16, the authors showed that allelic variation of the OsMADS1 gene affected the expression of OsMADS17/COG1. The key message of this manuscript is that 65-bp deletion in the 5'-UTR of OsMADS17 was associated with translation efficiency, improving grain yield. It is interesting to know if the allelic variation of OsMADS1 influence the accumulation of OsMADS17 at the protein level. And again, In Figure 5C, the authors should add both promoters of the OsMADS17 gene from C418 and 8IL73, respectively.

Reviewer #4:

Remarks to the Author:

On the whole, the authors have considered almost all of the concerns from my previous review and carefully explained them. I respect their responses and improvements for the present manuscript. The following two concerns can be considered.

In related to Supplemental Table 2 with a list of wild accessions, I pointed out non-rufipogon wild rice species, which were also given with "ruf" code. Also, I did not find non-rufipogon plants used in the experiment. To avoid confusion, I suggest not to use "ruf" code, or remove non-rufipogon wild rice accessions. I understand *O. nivara* can be left as an annual form of *O. rufipogon*. This suggestion is not essential, but from readers' view, it is confusing.

In related to Fig. 2t, I wonder aus cultivars can be missing as they are about 200.

We are appreciated for the reviewers' careful review and valuable suggestions. According to the reviewers' suggestions, we have revised the manuscript carefully. A detailed point-by-point response addressing the reviewers' comments and queries are described below.

Reviewer #1's questions and our responses:

R1Q1 (Reviewer #1's question 1): The authors have addressed most of the comments and suggestions I raised in my initial review. I think the new data and corrections have improved the manuscript, in particular the clarifications regarding the number of transgenics events analyzed, quantification of COG1 protein levels and specificity of RNAi vectors.

Our response to R1Q1: Thank you for your appreciation.

R1Q2 (Reviewer #1's question 2): Still, I have some concerns regarding the proposed genetic module OsMADS1-COG1-OsAP2-39:

I agree with the authors' conclusion that their evidence support that COG1 is a positive regulator of AP2-39, and that that regulation is likely involved in grain size and number regulation. However, I think that the genetic evidence does not support conclusions as stated in:

Title: The genetic module OsMADS1-COG1-OsAP2-39 simultaneously increases grain number and grain weight in rice.

Lines 21-22: "We found that the genetic module OsMADS1-COG1-OsAP2-39 increases both grain number and individual grain weight

simultaneously”.

Lines 253-254 “These results suggested that genes OsMADS1, COG1, and OsAP2-39 functioned together to regulate individual grain weight.

Lines 268-270: “these findings showed that OsMADS1, COG1, and OsAP2-39 co-regulated grain yield by constituting a genetic module sequentially from the upstream to the downstream (Fig. 5k)”.

According to the new genetic results, cog1 osap2-39 double mutant showed additive effects relative the single mutants in the regulation of grain size and yield. This observation is not consistent with a single genetic module COG1-OsAP2-39, but more consistent with a model where COG1 (more likely) or AP2-39 regulate grain traits by controlling additional genes.

I think that all these 3 genes are part of large gene network controlling inflorescence, flower, and grain development. COG1 likely control multiple genes as indicated by the authors in line 176 “It showed that COG1 participated in multiple regulatory pathways” or in line 318 “As a transcription factor, COG1 probably participated in the regulation of expression of multiple genes”. I think the authors did a good work validating the interaction of COG1 with one of those potential targets, AP2-39, but I think it is not correct to conclude that they have identify a genetic module controlling grain development. Instead, I think they have initiated the characterization of a network controlled by COG1 and I suggest the authors to modify the title, abstract and the manuscript to reflect that idea.

The relation with MADS1 is even more complicated since OsMADS1 controls the expression of multiple genes during inflorescence development and *osmads1* plants have several pleiotropic effects. In addition, the authors did not find evidence that the natural variation in OsMADS1, associated with changes in grain size, affects COG1 or AP2-39 expression (supplementary fig 16). Also, I feel that the selection of MADS1 based on the presence of a CArG box in COG1 promoter is biased. In conclusion, I think that the current results related to MADS1 are not well supported, and are not relevant to the main story of the manuscript.

Our response to R1Q2: (1) Thank you for your approval of our conclusion that *COG1/OsMADS17* regulates grain number and grain size. According to your suggestion, we have modified the contents in the revised manuscript as follows:

Title: *OsMADS17* simultaneously increases grain number and grain weight in rice.

In line 20-23: It indicated that *OsMADS1-OsMADS17-OsAP2-39* participated in the regulatory network controlling grain yield, and down-regulation of *OsMADS17* or *OsAP2-39* expression could further improve grain yield by simultaneously increasing grain number and grain weight.

In line 255-257: These results suggested that all of the genes *OsMADS1*, *OsMADS17*, and *OsAP2-39* were involved into regulating individual grain weight.

In line 270-272: Taken together, these findings showed that

OsMADS1, *OsMADS17*, and *OsAP2-39* participated in the regulatory network controlling grain yield (Fig. 5k).

We also made revision in the other part of our revised manuscript as follows:

In line 52-55: *OsMADS17* combining with the upstream gene *OsMADS1* and the downstream gene *OsAP2-39* participates in the regulatory network regulating rice grain yield, and the favorable variants of *OsMADS1*, *OsMADS17*, and *OsAP2-39* could increase grain yield in rice improvement.

In line 273-274: To reveal the distribution of different genotypes for *OsMADS1*, *OsMADS17*, and *OsAP2-39* in rice populations.

In line 352-353: In this present study, we demonstrated that *OsMADS1-OsMADS17-OsAP2-39* participated in the regulatory network controlling grain number and grain weight in rice.

In line 731-732: *OsMADS1*, *OsMADS17*, and *OsAP2-39* participated in the regulatory network controlling rice grain yield.

(2) Thank you for your suggestion. *OsMADS1* does have pleiotropic effects on inflorescence development by regulating the expression of multiple genes. The previous studies revealed that the variation of alternative splicing (*OsMADS1^{AS}*) generated longer grains and affected the transcription level of the downstream genes, such as *OsMADS55*, *OsARF14*, *OsKANADI4* (Liu et al., 2018; Yu et al., 2018). Although several target genes were verified, *COG1/OsMADS17* has not been identified as the downstream gene of *OsMADS1*. In our manuscript, we found that

OsMADS1 could promote the expression of *OsMADS17* by binding to its promoter region (Fig. 5a-c in the revised manuscript), and decreasing the expression of *OsMADS1* by RNAi could down-regulate the transcript levels of *OsMADS17* and *OsAP2-39*, producing much longer grains (Fig. 5d and Supplementary Fig. 15d, j in the revised manuscript), which suggested that altered expression of *OsMADS1* had significant effect on grain length by regulating the expression of *OsMADS17* and *OsAP2-39*.

We also agree with you that the selection of *OsMADS1* based on the presence of a CArG box in *COG1* promoter is not comprehensive. In this manuscript, we only confirmed that OsMADS1 was indeed the upstream regulator of *OsMADS17*, participating in the rice grain yield regulatory network. If you think it should be removed, we are also willing to remove the results related to *OsMADS1*. Thank you!

Reference related to the discussion as following:

Liu et al. *Nat. Commun.* **9**, 852 (2018)

Yu et al. *Plant Biotechnol. J.* **16**, 1667-1678 (2018)

R1Q3 (Reviewer #1's question 3): Abstract.

I suggest including more details about COG1, for example that the 65-bp deletion in the 5' UTR in COG1 is presented in cultivated rice and is associated with enhanced grain weight. Also, that COG1 regulates multiple genes and that the interaction with one of them, AP2-39, has been characterized.

Our response to R1Q3: According to your suggestion, we have modified the abstract in line 13-25 in the revised manuscript as follows:

During the processes of rice domestication and improvement, a trade-off effect between grain number and grain weight was a major obstacle for increasing yield. Here, we identified a critical gene *COG1*, encoding the transcription factor OsMADS17, with a 65-bp deletion in the 5' untranslated region (5' UTR) presented in cultivated rice increasing grain number and grain weight simultaneously through decreasing mRNA translation efficiency. OsMADS17 controls grain yield by regulating multiple genes and that the interaction with one of them, *OsAP2-39*, has been characterized. Besides, the expression of *OsMADS17* was regulated by OsMADS1 directly. It indicated that *OsMADS1-OsMADS17-OsAP2-39* participated in the regulatory network controlling grain yield, and downregulation of *OsMADS17* or *OsAP2-39* expression could further improve grain yield by simultaneously increasing grain number and grain weight. Our findings provide new insights into understanding the molecular basis co-regulating rice yield-related traits, and also offer a novel strategy for breeding higher-yielding rice varieties.

Thank you!

R1Q4 (Reviewer #1's question 4): Introduction.

Lines 28-42. I found the two first paragraph of the introduction (lines 28-35 and lines 36 to 42) very redundant. I think they could be combined in a single improved paragraph.

Our response to R1Q4: Thanks for your suggestion. We have combined the two paragraphs in a single improved paragraph in line 27-39 in the revised manuscript as follows:

Crop domestication and improvement are outstanding events in human history. During the process of the wild rice (*Oryza rufipogon* Griff.) domesticated into cultivated rice (*Oryza sativa* L.), remarkable morphological transitions have occurred, such as superior plant architecture accompanied by decreased tiller number and tiller angle and higher grain yield resulted from increased both grain number and grain weight¹⁻¹⁴ (Fig. 1a, b). After the domestication, increasing grain yield remained to be the prime target in the process of rice improvement. Over the past decades, a number of key genes regulating grain number¹⁵⁻²⁵ and grain weight²⁶⁻⁴² had been characterized, which largely contributed to the significant achievements in genetic improvement of rice grain yield. However, most of these genes only regulated one of the traits, or even negatively regulated the other one, which were known as the trade-off effect between grain number and grain weight, and it restrained the greater achievements for improving rice grain yield. Therefore, how to overcome the trade-off effect between grain number and grain weight promoting them simultaneously is a challenging task left to be resolved in modern breeding programmes.

R1Q5 (Reviewer #1's question 5): Results.

Line 89. "...phenotype and COG1 expression level as with 8IL73". I think this should be corrected to "...phenotype and COG1 expression level as non-transformed C418".

Our response to R1Q5: Thanks for your suggestion. We have corrected this sentence in line 84-87 in the revised manuscript as follows:

The C418^{CTP} transgenic plants, which were generated by introducing the *LOC_Os04g49150* gene from 8IL73 into C418 plants, showed a similar *OsMADS17* expression level, but fewer grains per panicle, a lower grain weight and grain yield than that of the C418 control plants.

R1Q6 (Reviewer #1's question 6): Lines 93-97. The authors showed in their response letter that the RNAi vector against COG1 is specific, but they have not included that figure in the current manuscript. I think it would be good to include it when presenting the results of 8IL73RNAi-COG1. It could be mentioned as "To verify the function of *LOC_Os04g49150*, we generated a COG1 specific RNA interference (RNAi) vector (Supplementary FigX). Transgenic 8IL73 plants expressing the RNAi vector (8IL73RNAi-COG1) exhibited a significant decrease in COG1 expression levels..."

Our response to R1Q6: According to your suggestion, we have added the figure (Supplementary Fig. 8a in the revised manuscript) and revised the sentence in line 90-94 in the revised manuscript as follows:

To further verify the function of *LOC_Os04g49150*, we generated a *OsMADS17* specific RNA interference (RNAi) vector (Supplementary Fig. 8a). Transgenic 8IL73 plants expressing the RNAi vector (8IL73^{RNAi-OsMADS17}) exhibited a significant decrease in *OsMADS17* expression levels and a significant increase in grain number, grain size, and grain yield compared with the controls (Fig. 2i-o, Supplementary Fig. 8b-j).

Thank you!

R1Q7 (Reviewer #1's question 7): Line 119. I suggest changing “the expression” to “the transcript expression”

Our response to R1Q7: Thank you very much. We have changed “the expression” to “the transcript expression” in line 116 in the revised manuscript according to your suggestion.

R1Q8 (Reviewer #1's question 8): Line 121. The authors should mention that C418 contains lower protein levels in more tissues than young panicles. The plants having higher expression of COG1 look shorter (Figures 6a; Supplementary Figures 2a, 6a, 8a, 9b, 10c), so I am wondering whether the difference in COG1 protein levels in nodes (Supplementary Figure 5b) could be related to this phenotype.

Our response to R1Q8: Thank you for your suggestion.

(1) We have added the contents “C418 contained lower *OsMADS17*

protein levels than that of 8IL73 in young panicles and other tissues, such as root, tiller base, and node” in line 118-119 in the revised manuscript.

(2) We investigated the plant height for analysis. Compared to C418, 8IL73 has a reduction of plant height by 8.6% and contains higher OsMADS17 protein levels in nodes (Supplementary Fig. 5b in the revised manuscript). Although we only detected the protein levels of OsMADS17 in young panicles (Fig. 3i, j, l, Fig. 6f in the revised manuscript), it can be reasonably predicted that the protein level of OsMADS17 in the nodes of C418^{CTP} transgenic plants is higher than that of control plants, and the protein content of OsMADS17 in the nodes of 8IL73^{RNAi-OsMADS17} and C418^{RNAi-OsMADS17} transgenic plants is lower than that of non-transgenic plants, respectively. Correspondingly, compared to the control, the plant height of the C418^{CTP} transgenic plants decreased by 6.8%, while the plant height of 8IL73^{RNAi-OsMADS17} and C418^{RNAi-OsMADS17} increased by 4.4% and 3.2%, respectively. These results suggested that the plants containing higher OsMADS17 protein accumulation exhibited declined plant height and those with lower OsMADS17 protein accumulation increased plant height. Therefore, we think that the change in OsMADS17 protein levels among various plants may be related to the difference of plant height.

R1Q9 (Reviewer #1’s question 9): Line 135. I suggest changing “F5 (containing the fragment F3),” to “F5 (32-bp in the end of 3’ region in the 65-bp, including the deletion in the fragment F3)”

Our response to R1Q9: We have revised it as “F5 (32-bp in the end of 5’ region in the 65-bp, including the fragment F3)” in line 132-133 in the revised manuscript. Thank you!

R1Q10 (Reviewer #1’s question 10): Lines 146-147. I think a better conclusion to this paragraph would be that “the De65-bp in the 5’UTR of the COG1 gene significantly attenuate TE and result in increased grain number, grain weight, and grain yield”. Out of curiosity, did the author check for any conserve element in the 22-bp fragment?

Our response to R1Q10: (1) We have revised it according to your suggestion in line 144-145 in the revised manuscript. Thank you!

(2) We had checked for the 22-bp fragment and found that the sequence “AGAAGAAGCAG” within it was highly similar to the features of N^1 -methyladenosine (m^1A , methylation at the N^1 position of adenosine) sequence motif as it reported (Dominissini et al., 2016; Li et al., 2016; Yang et al., 2020). m^1A occurs on the Watson-Crick interface, which carries a positive charge, dramatically altering RNA secondary structures and protein-RNA interactions (Roundtree et al., 2017). In mammals, m^1A is preferentially located within 5’ UTR near start codons and the first splice site, and the mRNA with m^1A tend to generate higher protein levels than those with non- m^1A (Dominissini et al., 2016; Li et al., 2016). In plants, such as petunia (*Petunia hybrida*), m^1A in mRNA, playing multiple roles for protein degradation, transcription or translation regulation, and so on,

regulates plant growth and development, which also has been identified (Yang et al., 2020). Therefore, we speculated that the 22-bp fragment within the In65-bp might result in m¹A for the mRNA of *OsMADS17*, which lead to the change of translation efficiency, and we will make further exploration in the next study to prove the speculation.

Discussion based on the following references:

Dominissini et al. *Nature* **530**, 441-446 (2016).

Li et al. *Nat. Chem. Biol.* **12**, 311-316 (2016).

Roundtree et al. *Cell* **169**, 1187-1200 (2017).

Yang et al. *Plant Physiol.* **183**, 1710-1724 (2020).

R1Q11 (Reviewer #1's question 11): Lines 165-167. I feel that the conclusion sentence fit better after paragraph between lines 148-157. Maybe a proper conclusion for the paragraph between lines 158-165 would be that 65-bp deletion is associated with larger grains in cultivated rice.

Our response to R1Q11: Thanks for your suggestions.

(1) We have moved the sentence “These results confirmed that the 65-bp deletion in the 5' UTR of the *OsMADS17* gene originated from the mutation in cultivated rice” to line 156-158 in the revised manuscript.

(2) We have added the conclusion “These results indicated that the 65-bp deletion was associated with larger grains in cultivated rice” in line 167 in the revised manuscript.

R1Q12 (Reviewer #1's question 12): Line 192. The authors showed in the response letter that the RNAi vector against AP2-39 is specific, but they have not included the figure in the current manuscript. I suggest including that.

Our response to R1Q12: According to your suggestion, we have added the figure (Supplementary Fig. 13a in the revised manuscript) and revised the sentence in line 191-194 in the revised manuscript as follows:

we constructed the *OsAP2-39* overexpressed (OE-*OsAP2-39*) and *OsAP2-39* specific RNA interference (RNAi-*OsAP2-39*) vectors (Supplementary Fig. 13a) to generate C418^{OE-*OsAP2-39*} and C418^{RNAi-*OsAP2-39*} transgenic plants, respectively.

Thank you!

R1Q13 (Reviewer #1's question 13): Line 203. Please indicate which tissue was used in the RT-qPCR.

Our response to R1Q13: We used the young panicles with the length of 0-0.5 cm for RT-qPCR analysis, and we have added this information in line 204 in the revised manuscript. Thank you!

R1Q14 (Reviewer #1's question 14): Lines 217-218. Since only a few accessions contain both *OsAP2-39*^T and *COG1*^{De65-bp} alleles, I have doubts about the conclusion that the combination of both alleles was selected.

How can the author exclude the possibility that observation is the result of random crossing?

Our response to R1Q14: Thanks for your valuable suggestion. Based on the current data, we really cannot exclude the possibility that observation is the result of random crossing. In order to express our conclusion more accurately, we have modified the contents in the revised manuscript as following:

In line 208-210: To gain an insight into the distribution of the *OsAP2-39^{T/C}* genotype in different rice populations, we screened the 82 accessions of wild rice, finding all of them contained the genotype *OsAP2-39^C* (Supplementary Table 2).

In line 217-219: Taken together, *OsAP2-39^T* is a rare mutation in cultivated rice, and the favorable variation of *OsAP2-39^T* and *OsMADS17^{De-65bp}* combined together only existed in *indica* rice.

In line 273-274: To reveal the distribution of different genotypes for *OsMADS1*, *OsMADS17*, and *OsAP2-39* in rice populations.

In line 283-285: Taken together, these findings indicated that *OsMADI^{AS}* and *OsMADS17^{De65-bp}* were distributed independently in cultivated rice, and that *OsAP2-39^T* and *OsMADI^{AS}* appeared simultaneously merely in *japonica* rice.

R1Q15 (Reviewer #1's question 15): Lines 252 and 253. *COG1^{AS}*. Did the authors mean *OsMADs1^{AS}*?

Our response to R1Q15: We apologize for the typing error in the previous manuscript. It should be *OsMADS17*^{De-65bp} in our manuscript, and we have made the corrections in line 254 and line 255 in the revised manuscript. Thank you!

R1Q16 (Reviewer #1's question 16): Lines 285-288. I think the authors could explain their results in more details. The authors have showed that only 24% of cultivated rice have the beneficial De65-bpCOG1 allele, so reducing COG1 expression in the 76% having the In65-bp allele would be very interesting. In addition, the authors showed that reducing COG1 expression in cultivars that have the COG1 beneficial allele, like C418, also result in more and larger grains. So further reduction of COG1 in cultivated rice with the beneficial allele still could be beneficial. I think the authors should improve the paragraph to clearly explain all these possibilities.

Our response to R1Q16: Thanks for your valuable suggestion. We have rewritten this paragraph in line 287-301 in the revised manuscript as follows:

As *OsMADS17* is a negative regulator controlling grain yield in rice, reducing its expression level perhaps could contribute to increasing rice grain yield. Above results had demonstrated that down-regulated expression levels of *OsMADS17* with In65-bp in 8IL73 plants could

increase grain yield (Fig. 2i-o, Supplementary Fig. 8b-j), which meant great prospect for improving grain yield of the 75.9% cultivars harboring the 65-bp insertion of *OsMADS17* (Fig. 4q). To further explore the potential of *OsMADS17* increasing grain yield in the remaining of 24.1% cultivated rice with the beneficial 65-bp deletion (Fig. 4q), we generated the transgenic plants C418^{RNAi-OsMADS17} with downregulated *OsMADS17* expression levels, which showed lower levels of OsMADS17 protein accumulation, higher grain number, bigger grain size, and higher grain yield compared with the controls (Fig. 6a-i, Supplementary Fig. 19). Field trials showed that the C418^{RNAi-OsMADS17} transgenic plants increased grain yield by 6.5% in Beijing (40.1°N, 116.2°E), northern China, and 12.4% in Hainan Province (18.3°N, 109.2°E), southern China, compared with the control plants (Fig. 6j, k). It indicated that reducing the expression levels of *OsMADS17* in cultivated rice was desirable to improve grain yield.

R1Q17 (Reviewer #1's question 17): Methods

Line 416. Please indicate that in Fig 3 and 6 western blots were done with 3 replications. Also indicate the method use to quantify the western blot bands.

Our response to R1Q17: Thanks for your suggestions.

(1) We have added the contents “(n = three replications)” in both line 686 and line 743-744 in the revised manuscript.

(2) We have added the contents “HSP82 and OsMADS17 bands were

analyzed with the Image J program (<https://imagej.net/ij/>) to calculate the protein levels according to the software instruction manual” in the part of methods in line 431-432 in the revised manuscript.

Reviewer #3’s questions and our responses:

R3Q1 (Reviewer #3’s question 1): In this revised version of the manuscript, the authors have added additional experiments and answered the questions which the reviewer #2 and I addressed. Most of my comments are relatively minor, I think the authors should give attention to the following suggestions before it is accepted for publication:

both reviewer #1 and I pointed out the same suggestion, I still think that the authors should modify the title, and directly replace COG1 with OsMADS17.

Our response to R3Q1: Thank you very much for your suggestion.

According to your suggestion, we have modified the title as “*OsMADS17* simultaneously increases grain number and grain weight in rice”, and we also have replaced COG1 with OsMADS17 in the text.

R3Q2 (Reviewer #3’s question 2): in the new Supplementary Figure 16, the authors showed that allelic variation of the OsMADS1 gene affected the expression of OsMADS17/COG1. The key message of this manuscript is that 65-bp deletion in the 5’-UTR of OsMADS17 was associated with translation efficiency, improving grain yield. It is interesting to know if the

allelic variation of *OsMADS1* influence the accumulation of *OsMADS17* at the protein level. And again, In Figure 5C, the authors should add both promoters of the *OsMADS17* gene from C418 and 8IL73, respectively.

Our response to R3Q2: Thanks for your suggestions.

(1) According to your suggestion, we have detected the protein levels of *OsMADS17* between NIL-*OsMADS1*^{NS} and NIL-*OsMADS1*^{AS} plants, and it also showed no significant change between them (Supplementary Fig. 16b, c in the revised manuscript). It meant that allelic variation of *OsMADS1*^{NS/AS} did not affect the transcription level of the *OsMADS17* gene, nor did it change the *OsMADS17* protein level. We have added the results in Supplementary Fig. 16b, c, and modified the contents in line 240-242 in the revised manuscript as following:

RT-qPCR and western blot analysis showed no significant difference in *OsMADS1*, *OsMADS17*, and *OsAP2-39* transcript levels, as well as *OsMADS17* protein levels between NIL-*OsMADS1*^{AS} and NIL-*OsMADS1*^{NS} plants (Supplementary Fig. 16).

(2) According to your suggestion, we have added the results of the interaction between *OsMADS1* and the promoter region of *OsMADS17* in C418 or 8IL73 by dual-*LUC* reporter assays. The results showed that *OsMADS1* promoted the activation of the promoter of *OsMADS17* in both C418 and 8IL73 (Fig. 5c in the revised manuscript). Thank you!

Reviewer #4's questions and our responses:

R4Q1 (Reviewer #4's question 1): On the whole, the authors have considered almost all of the concerns from my previous review and carefully explained them. I respect their responses and improvements for the present manuscript. The following two concerns can be considered.

Our response to R4Q1: Thank you for your appreciation.

R4Q2 (Reviewer #4's question 2): In related to Supplemental Table 2 with a list of wild accessions, I pointed out non-rufipogon wild rice spices, which were also given with “ruf” code. Also, I did not find non-rufipogon plants used in the experiment. To avoid confusion, I suggest not to use “ruf” code, or remove non-rufipogon wild rice accessions. I understand *O. nivara* can be left as an annual form of *O. rufipogon*. This suggestion is not essential, but from readers' view, it is confusing.

Our response to R4Q2: Thank you for your suggestion. To avoid confusing, we have removed the non-*rufipogon* wild rice accessions in the Supplementary Table 2.

R4Q3 (Reviewer #4's question 3): In related to Fig. 2t, I wonder aus cultivars can be missing as they are about 200.

Our response to R4Q3: Thank you for your suggestion. we have added

the results of the frequency of favorable genotype of *OsMADS17*^{De65-bp} (Fig. 3w), *OsAP2-39*^T (Fig. 4p), and *OsMADS1*^{AS} (Fig. 5h) in *aus* cultivars. The distribution study showed that the frequency in *indica*, *japonica*, and *aus* cultivars was 44.2%, 14.8%, and 26.0%, for the genotype *OsMADS17*^{De65-bp} and 6.6%, 11.6%, and 4.1% for *OsAP2-39*^T, and 1.7%, 56.0%, and 0.5% for *OsMADS1*^{AS}, respectively. These results showed that *OsMADS17*^{De65-bp} and *OsMADS1*^{AS} were both mainly appeared in *japonica* rice, and *OsAP2-39*^T was a rare mutation from cultivated rice. We have added the contents in the revised manuscript as follows:

In line 152-154: The distribution study showed that the frequency of cultivars with the genotype *OsMADS17*^{De65bp} in *japonica*, *indica*, and *aus* rice was 44.2%, 14.8%, and 26.0%, respectively (Fig. 3w).

In line 210-212: In the 2,775 cultivated rice accessions from rice 3K RG, the frequency of cultivars with *OsAP2-39*^T in *japonica*, *indica*, and *aus* rice was 6.6%, 11.6%, and 4.1%, respectively (Fig. 4p).

Reviewers' Comments:

Reviewer #1:

Remarks to the Author:

The authors have addressed all my comments and suggestions.
Congratulations for the great work.

Reviewer #3:

Remarks to the Author:

In this revised version of the

Reviewer #4:

Remarks to the Author:

The authors have addressed the comments from my previous review in a rigorous and systematic way. Now, I think the manuscript has been greatly improved. And I believe that the current work will be important for future study.